# Decision-Aware Actor-Critic with Function Approximation and Theoretical Guarantees

**Sharan Vaswani**
Simon Fraser University
vaswani.sharan@gmail.com

**Amirreza Kazemi**
Simon Fraser University
aka208@sfu.ca

**Reza Babanezhad**
Samsung - SAIT AI Lab, Montreal
babanezhad@gmail.com

**Nicolas Le Roux**
Microsoft Research, Mila
nicolas@le-roux.name

## Abstract

Actor-critic (AC) methods are widely used in reinforcement learning (RL), and benefit from the flexibility of using any policy gradient method as the actor and value-based method as the critic. The critic is usually trained by minimizing the TD error, an objective that is potentially decorrelated with the true goal of achieving a high reward with the actor. We address this mismatch by designing a joint objective for training the actor and critic in a *decision-aware* fashion. We use the proposed objective to design a generic, AC algorithm that can easily handle any function approximation. We explicitly characterize the conditions under which the resulting algorithm guarantees monotonic policy improvement, regardless of the choice of the policy and critic parameterization. Instantiating the generic algorithm results in an actor that involves maximizing a sequence of surrogate functions (similar to TRPO, PPO), and a critic that involves minimizing a closely connected objective. Using simple bandit examples, we provably establish the benefit of the proposed critic objective over the standard squared error. Finally, we empirically demonstrate the benefit of our decision-aware actor-critic framework on simple RL problems.

## 1   Introduction

Reinforcement learning (RL) is a framework for solving problems involving sequential decision-making under uncertainty, and has found applications in games [38, 50], robot manipulation tasks [55, 64] and clinical trials [45]. RL algorithms aim to learn a policy that maximizes the long-term return by interacting with the environment. Policy gradient (PG) methods [59, 54, 29, 25, 47] are an important class of algorithms that can easily handle function approximation and structured state-action spaces, making them widely used in practice. PG methods assume a differentiable parameterization of the policy and directly optimize the return with respect to the policy parameters. Typically, a policy's return is estimated by using Monte-Carlo samples obtained via environment interactions [59]. Since the environment is stochastic, this approach results in high variance in the estimated return, leading to higher sample-complexity (number of environment interactions required to learn a good policy). Actor-critic (AC) methods [29, 43, 5] alleviate this issue by using value-based approaches [52, 58] in conjunction with PG methods, and have been empirically successful [20, 23]. In AC algorithms, a value-based method ("critic") is used to approximate a policy's estimated value, and a PG method ("actor") uses this estimate to improve the policy towards obtaining higher returns.

Though AC methods have the flexibility of using any method to independently train the actor and critic, it is unclear how to train the two components *jointly* in order to learn good policies. For example, the critic is typically trained via temporal difference (TD) learning and its objective is to minimize the value estimation error across all states and actions. For large real-world Markov decision processes (MDPs), it is intractable to estimate the values across all states and actions, and

algorithms resort to function approximation schemes. In this setting, the critic should focus its limited model capacity to correctly estimate the state-action values that have the largest impact on improving the actor's policy. This idea of explicitly training each component of the RL system to help the agent take actions that result in higher returns is referred to as *decision-aware RL*. Decision-aware RL [17, 16, 1, 10, 13, 14, 32] has mainly focused on model-based approaches that aim to learn a model of the environment, for example, the rewards and transition dynamics in an MDP. In this setting, decision-aware RL aims to model relevant parts of the world that are important for inferring a good policy. This is achieved by (i) designing objectives that are aware of the current policy [1, 14] or its value [17, 16], (ii) differentiating through the transition dynamics to learn models that result in good action-value functions [13] or (iii) simultaneously learning value functions and models that are consistent [51, 40, 35]. In the model-free setting, decision-aware RL aims to train the actor and critic cooperatively in order to optimize the same objective that results in near-optimal policies. In particular, Dai et al. [10] use the linear programming formulation of MDPs and define a joint saddle-point objective (minimization w.r.t. the critic and maximization w.r.t. the actor). The use of function approximation makes the resulting optimization problem non-convex non-concave leading to training instabilities and necessitating the use of heuristics. Recently, Dong et al. [11] used stochastic gradient descent-ascent to optimize this saddle-point objective and, under certain assumptions on the problem, proved that the resulting policy converges to a stationary point of the value function. Similar to Dong et al. [11], we study a decision-aware AC method with function approximation and equipped with theoretical guarantees on its performance. In particular, we make the following contributions.

**Joint objective for training the actor and critic**: Following Vaswani et al. [57], we distinguish between a policy's *functional representation* (sufficient statistics that define a policy) and its *parameterization* (the specific model used to realize these sufficient statistics in practice). For example, a policy can be represented by its state-action occupancy measure, and we can use a neural network parameterization to model this measure in practice (refer to Sec. 2 for more examples). In Sec. 3.2, we exploit a smoothness property of the return and design a lower-bound (Prop. 1) on the return of an arbitrary policy. Importantly, the lower bound depends on both the actor and critic, and immediately implies a joint objective for training the two components (minimization w.r.t the critic and maximization w.r.t the actor). Unlike Dai et al. [10], Dong et al. [11], the proposed objective works for **any** policy representation – the policy could be represented as conditional distributions over actions for each state or a deterministic mapping from states to actions [21]. Another advantage of working in the functional space is that our lower bound does not depend on the parameterization of either the actor or the critic. Moreover, unlike Dai et al. [10], Dong et al. [11], our framework does not need to model the distribution over states, and hence results in a more efficient algorithm. We note that our framework can be used for other applications where gradient computation is expensive or has large variance [39], and hence requires a model of the gradient (e.g., variational inference).

**Generic actor-critic algorithm**: In Sec. 3.2, we use our joint objective to design a generic decision-aware AC algorithm. The resulting algorithm (Algorithm 1) can be instantiated with any functional representation of the policy, and can handle any policy or critic parameterization. Similar to Vaswani et al. [57], the actor update involves optimizing a *surrogate function* that depends on the current policy, and consequently supports *off-policy updates*, i.e. similar to common PG methods such as TRPO [46], PPO [48], the algorithm can update the policy without requiring additional interactions with the environment. This property coupled with the use of a critic makes the resulting algorithm sample-efficient in practice. In contrast with TRPO/PPO, both the off-policy actor updates and critic updates in Algorithm 1 are designed to maximize the same lower bound on the policy return.

**Theoretical guarantees**: In Sec. 4.1, we analyze the necessary and sufficient conditions in order to guarantee monotonic policy improvement, and hence convergence to a stationary point. We emphasize that these improvement guarantees hold regardless of the policy parameterization and the quality of the critic (up to a certain threshold that we explicitly characterize). This is in contrast to existing theoretical results that focus on the tabular or linear function approximation settings or rely on highly expressive critics to minimize the critic error and achieve good performance for the actor. By exploiting the connection to inexact mirror descent (MD), we prove that Algorithm 1 is guaranteed to converge to the neighbourhood of a stationary point where the neighbourhood term depends on the decision-aware critic loss (Sec. 4.2). Along the way, we improve the theoretical guarantees for MD on general smooth, non-convex functions [15, 12]. As an additional contribution, we demonstrate a way to use the framework of Vaswani et al. [57] to "lift" the existing convergence rates [61, 37, 24] for the tabular setting to use off-policy updates and function approximation (Appendix D.2 and D.3). This gives rise to a simple, black-box proof technique that might be of independent interest.

**Instantiating the general AC framework**: We instantiate the framework for two policy representations – in Sec. 5.1, we represent the policy by the set of conditional distributions over actions ("direct" representation), whereas in Sec. 5.2, we represent the policy by using the logits corresponding to a softmax representation of these conditional distributions ("softmax" representation). In both cases, we instantiate the generic lower-bound (Propositions 4, 6), completely specifying the actor and critic objectives in Algorithm 1. Importantly, unlike the standard critic objective that depends on the squared difference of the value functions, the proposed decision-aware critic loss (i) depends on the policy representation – it involves the state-action value functions for the direct representation and depends on the advantage functions for the softmax representation, and (ii) penalizes the under-estimation and over-estimation of these quantities in an asymmetric manner. For both representations, we consider simple bandit examples (Propositions 5, 7) which show that minimizing the decision-aware critic loss results in convergence to the optimal policy, whereas minimizing variants of the squared loss do not. In App. B, we consider a third policy representation involving stochastic value gradients [21] for continuous control, and instantiate our decision-aware actor-critic framework in this case.

**Experimental evaluation**: Finally, in Sec. 6, we consider simple RL environments and benchmark Algorithm 1 for both the direct and softmax representations with a linear policy and critic parameterization. We compare the actor performance when using the squared critic loss vs the proposed critic loss, and demonstrate the empirical benefit of our decision-aware actor-critic framework.

## 2 Problem Formulation

We consider an infinite-horizon discounted Markov decision process (MDP) [44] defined by the tuple $\langle \mathcal{S}, \mathcal{A}, \mathcal{P}, r, \rho, \gamma \rangle$ where $\mathcal{S}$ is the set of states, $\mathcal{A}$ is the action set, $\mathcal{P} : \mathcal{S} \times \mathcal{A} \to \Delta_{\mathcal{S}}$ is the transition probability function, $\rho \in \Delta_{\mathcal{S}}$ is the initial distribution of states, $r : \mathcal{S} \times \mathcal{A} \to [0, 1]$ is the reward function and $\gamma \in [0, 1)$ is the discount factor. For state $s \in \mathcal{S}$, a policy $\pi$ induces a distribution $p^\pi(\cdot|s)$ over actions. It also induces a measure $d^\pi$ over states such that $d^\pi(s) = (1 - \gamma) \sum_{\tau=0}^{\infty} \gamma^\tau \mathcal{P}(s_\tau = s \mid s_0 \sim \rho, a_\tau \sim p^\pi(\cdot|s_\tau))$. Similarly, we define $\mu^\pi$ as the measure over state-action pairs induced by policy $\pi$, implying that $\mu^\pi(s, a) = d^\pi(s) \, p^\pi(a|s)$ and $d^\pi(s) = \sum_a \mu^\pi(s, a)$. The *action-value function* corresponding to policy $\pi$ is denoted by $Q^\pi : \mathcal{S} \times \mathcal{A} \to \mathbb{R}$ such that $Q^\pi(s, a) := \mathbb{E}[\sum_{\tau=0}^{\infty} \gamma^\tau r(s_\tau, a_\tau)]$ where $s_0 = s, a_0 = a$ and for $\tau \geq 0$, $s_{\tau+1} \sim \mathcal{P}(\cdot|s_\tau, a_\tau)$ and $a_{\tau+1} \sim p^\pi(\cdot|s_{\tau+1})$. The *value function* of a stationary policy $\pi$ for the start state equal to $s$ is defined as $J_s(\pi) := \mathbb{E}_{a \sim p^\pi(\cdot|s)}[Q^\pi(s, a)]$ and we define $J(\pi) := \mathbb{E}_{s \sim \rho}[J_s(\pi)]$. For a state-action pair $(s, a)$, the *advantage function* corresponding to policy $\pi$ is given by $A^\pi(s, a) := Q^\pi(s, a) - J_s(\pi)$. Given a set of feasible policies $\Pi$, the objective is to compute the policy that maximizes $J(\pi)$.

**Functional representation vs Policy Parameterization:** Similar to the policy optimization framework of Vaswani et al. [57], we differentiate between a policy's functional representation and its parameterization. The *functional representation* of a policy $\pi$ defines its sufficient statistics, for example, we may represent a policy via the set of distributions $p^\pi(\cdot|s) \in \Delta_A$ for state $s \in \mathcal{S}$. We will refer to this as the *direct representation*. The same policy can have multiple functional representations, for example, since $p^\pi(\cdot|s)$ is a probability distribution, one can write $p^\pi(a|s) = \exp(z^\pi(s,a)) / \sum_{a'} \exp(z^\pi(s,a'))$, and represent $\pi$ by the set of logits $z^\pi(s, a)$ for each $(s, a)$ pair. We will refer to this as the *softmax representation*. On the other hand, the *policy parameterization* is determined by a *model* (with parameters $\theta$) that realizes these statistics. For example, we could use a neural-network to parameterize the logits corresponding to the policy's softmax representation, rewriting $z^\pi(s, a) = z^\pi(s, a|\theta)$ where the model is implicit in the $z^\pi(s, a|\theta)$ notation. As another example, the *tabular parameterization* corresponds to having a parameter for each state-action pair [61, 37]. The policy parameterization thus defines the set $\Pi$ of realizable policies that can be expressed with the parametric model at hand. Note that the policy parameterization can be chosen independently of its functional representation. Next, we recap the framework in Vaswani et al. [57] and generalize it to the actor-critic setting.

## 3 Methodology

We describe functional mirror ascent in Sec. 3.1, and use it to design a general decision-aware actor-critic framework and corresponding algorithm in Sec. 3.2.

### 3.1 Functional Mirror Ascent for Policy Gradient (FMAPG) framework

For a given functional representation, Vaswani et al. [57] update the policy by *functional mirror ascent* and project the updated policy onto the set $\Pi$ determined by the policy parameterization.

Functional mirror ascent is an iterative algorithm whose update at iteration $t \in \{0, 1, \ldots, T-1\}$ is given as: $\pi_{t+1} = \arg\max_{\pi \in \Pi} \left[ \langle \pi, \nabla_\pi J(\pi_t) \rangle - \frac{1}{\eta} D_\Phi(\pi, \pi_t) \right]$ where $\pi_t$ is the policy (expressed as its functional representation) at iteration $t$, $\eta$ is the step-size in the functional space and $D_\Phi$ is the Bregman divergence (induced by the mirror map $\Phi$) between the representation of policies $\pi$ and $\pi_t$. The FMAPG framework casts the projection step onto $\Pi$ as an unconstrained optimization w.r.t the parameters $\theta \in \mathbb{R}^n$ of a *surrogate function*: $\theta_{t+1} = \arg\max \ell_t(\theta) := \langle \pi(\theta), \nabla_\pi J(\pi(\theta_t)) \rangle - \frac{1}{\eta} D_\Phi(\pi(\theta), \pi(\theta_t))$. Here, $\pi(\theta)$ refers to the parametric form of the policy where the choice of the parametric model is implicit in the $\pi(\theta)$ notation. The policy at iteration $t$ is thus expressed as $\pi(\theta_t)$, whereas the updated policy is given by $\pi_{t+1} = \pi(\theta_{t+1})$. The surrogate function is non-concave in general and can be approximately maximized using a gradient-based method, resulting in a nested loop algorithm. Importantly, the inner-loop (optimization of $\ell_t(\theta)$) updates the policy parameters (and hence the policy), but does not involve recomputing $\nabla_\pi J(\pi)$. Consequently, these policy updates do not require interacting with the environment and are thus *off-policy*. This is a desirable trait for designing sample-efficient PG algorithms and is shared by methods such as TRPO [46] and PPO [48].

With the appropriate choice of $\Phi$ and $\eta$, the FMAPG framework guarantees monotonic policy improvement for any number of inner-loops and policy parameterization. A shortcoming of this framework is that it requires access to the exact gradient $\nabla_\pi J(\pi)$. When using the direct or softmax representations, computing $\nabla_\pi J(\pi)$ involves computing either the action-value $Q^\pi$ or the advantage $A^\pi$ function respectively. In complex real-world environments where the rewards and/or the transition dynamics are unknown, these quantities can only be estimated. For example, $Q^\pi$ can be estimated using Monte-Carlo sampling by rolling out trajectories using policy $\pi$ resulting in large variance, and consequently higher sample complexity. Moreover, for large MDPs, function approximation is typically used to estimate the $Q$ function, and the resulting aliasing makes it impossible to compute it exactly in practice. This makes the FMAPG framework impractical in real-world scenarios. Next, we generalize FMAPG to handle inexact gradients and subsequently design an actor-critic framework.

### 3.2 Generalizing FMAPG to Actor-Critic

To generalize the FMAPG framework, we first prove the following proposition in App. C.

**Proposition 1.** For any policy representations $\pi$ and $\pi'$, any strictly convex mirror map $\Phi$, and any gradient estimator $\hat{g}$, for $c > 0$ and $\eta$ such that $J + \frac{1}{\eta}\Phi$ is convex in $\pi$,

$$J(\pi) \geq J(\pi') + \langle \hat{g}(\pi'), \pi - \pi' \rangle - \left( \frac{1}{\eta} + \frac{1}{c} \right) D_\Phi(\pi, \pi') - \frac{1}{c} D_{\Phi^*}\left( \nabla\Phi(\pi') - c[\nabla J(\pi') - \hat{g}(\pi')], \nabla\Phi(\pi') \right)$$

where $\Phi^*$ is the Fenchel conjugate of $\Phi$ and $D_{\Phi^*}$ is the Bregman divergence induced by $\Phi^*$.

The above proposition is a statement about the relative smoothness [34] of $J$ (w.r.t $D_\Phi$) in the functional space. Here, the brown term is the linearization of $J$ around $\pi'$, but involves $\hat{g}(\pi')$ which can be **any** estimate of the gradient at $\pi'$. The red term quantifies the distance between the representations of policies $\pi$ and $\pi'$ in terms of $D_\Phi(\pi, \pi')$, whereas the blue term characterizes the penalty for an inaccurate estimate of $\nabla_\pi J(\pi')$ and depends on $\Phi$. We emphasize that Prop. 1 can be used for **any** continuous optimization problem that requires a model of the gradient, e.g., in variational inference which uses an approximate posterior in lieu of the true one.

For policy optimization with FMAPG, $\nabla_\pi J(\pi)$ involves the action-value or advantage function for the direct or softmax functional representations respectively (see Sec. 5 for details), and the gradient estimation error is equal to the error in these functions. Since these quantities are estimated by the critic, we refer to the blue term as the *critic error*. In order to use Prop. 1, at iteration $t$ of FMAPG, we set $\pi' = \pi_t$ and include the policy parameterization, resulting in **inequality (I)**:

$$J(\pi) - J(\pi_t) \geq \langle \hat{g}_t, \pi(\theta) - \pi_t \rangle - \left( \frac{1}{\eta} + \frac{1}{c} \right) D_\Phi(\pi(\theta), \pi_t) - \frac{1}{c} D_{\Phi^*}\left( \nabla\Phi(\pi_t) - c[\nabla J(\pi_t) - \hat{g}_t], \nabla\Phi(\pi_t) \right),$$

where $\hat{g}_t := \hat{g}(\pi_t)$. We see that in order to obtain a policy $\pi$ that maximizes the policy improvement $J(\pi) - J(\pi_t)$ and hence the LHS, we should maximize the RHS i.e. (i) learn $\hat{g}_t$ to minimize the blue term (equal to the critic objective) and (ii) compute $\pi \in \Pi$ that maximizes the green term (equal to the functional mirror ascent update at iteration $t$). Using a second-order Taylor series expansion of $D_{\Phi^*}$ (Prop. 14), we see that as $c$ decreases, the critic error decreases, whereas the $\left( \frac{1}{\eta} + \frac{1}{c} \right) D_\Phi(\pi, \pi_t)$ term increases. Consequently, we interpret the scalar $c$ as a trade-off parameter that relates the critic error to the permissible movement in the functional mirror ascent update.

Hence, *both the actor and critic objectives are coupled through Prop. 1 and both components of the RL system should be jointly trained in order to maximize policy improvement.* We refer to the resulting framework as *decision-aware actor-critic* and present its pseudo-code in Algorithm 1.

---

**Algorithm 1:** Generic actor-critic algorithm

---

1 **Input**: $\pi$ (choice of functional representation), $\theta_0$ (initial policy parameters), $\omega_{(-1)}$ (initial critic parameters), $T$ (AC iterations), $m_a$ (actor inner-loops), $m_c$ (critic inner-loops), $\eta$ (functional step-size for actor), $c$ (trade-off parameter), $\alpha_a$ (parametric step-size for actor), $\alpha_c$ (parametric step-size for critic)

2 **Initialization**: $\pi_0 = \pi(\theta_0)$

3 **for** $t \leftarrow 0$ **to** $T - 1$ **do**

4 $\quad$ Estimate $\widehat{\nabla}_\pi J(\pi_t)$ and form $\mathcal{L}_t(\omega) := \frac{1}{c} D_{\Phi^*}\left(\nabla\Phi(\pi_t) - c\left[\widehat{\nabla}_\pi J(\pi_t) - \hat{g}_t(\omega)\right], \nabla\Phi(\pi_t)\right)$

5 $\quad$ Initialize inner-loop: $\upsilon_0 = \omega_{t-1}$

6 $\quad$ **for** $k \leftarrow 0$ **to** $m_c - 1$ **do**

7 $\quad\quad$ $\upsilon_{k+1} = \upsilon_k - \alpha_c \nabla_\upsilon \mathcal{L}_t(\upsilon_k)$ /* Critic Updates */

8 $\quad$ $\omega_t = \upsilon_{m_c}$ ; $\quad \hat{g}_t = \hat{g}_t(\omega_t)$

9 $\quad$ Form $\ell_t(\theta) := \langle \hat{g}_t, \pi(\theta) - \pi_t \rangle - \left(\frac{1}{\eta} + \frac{1}{c}\right) D_\Phi(\pi(\theta), \pi_t)$

10 $\quad$ Initialize inner-loop: $\nu_0 = \theta_t$

11 $\quad$ **for** $k \leftarrow 0$ **to** $m_a - 1$ **do**

12 $\quad\quad$ $\nu_{k+1} = \nu_k + \alpha_a \nabla_\nu \ell_t(\nu_k)$ /* Off-policy actor updates */

13 $\quad$ $\theta_{t+1} = \nu_{m_a}$ ; $\quad \pi_{t+1} = \pi(\theta_{t+1})$

14 Return $\pi_T = \pi(\theta_T)$

---

Unlike Wu et al. [60], Konda and Tsitsiklis [29], Algorithm 1 does not update the actor and critic in a two time-scale setting (one environment interaction and update to the critic followed by an actor update), but rather performs multiple steps to update the critic, then uses the critic to perform multiple steps to update the actor [2, 61]. At iteration $t$ of Algorithm 1, $\hat{g}_t$ (the gradient estimate at $\pi_t$) is parameterized by $\omega$ and the parametric model for the critic is implicit in the $\hat{g}_t(\omega)$ notation. The algorithm interacts with the environment, uses these interactions to form the estimate $\widehat{\nabla}_\pi J(\pi_t)$ and construct the critic loss function $\mathcal{L}_t(\omega)$. For the direct or softmax representations, $\widehat{\nabla}_\pi J(\pi_t)$ corresponds to the empirical estimates of the action-value or advantage functions respectively. In practice, these quantities can be estimated using Monte-Carlo rollouts or bootstrapping. Given these estimates, the critic is trained (using $m_c$ inner-loops) to minimize $\mathcal{L}_t(\omega)$ and obtain $\hat{g}_t$ (Lines 5-8). Line 9 uses $\hat{g}_t$ to construct the surrogate function $\ell_t(\theta)$ for the actor and depends on the policy parameterization. The inner-loop (Lines 10 - 13) involves maximizing $\ell_t(\theta)$ and corresponds to $m_a$ off-policy updates. Next, we establish theoretical guarantees on the performance of Algorithm 1.

## 4 Theoretical Guarantees

We first establish the necessary and sufficient conditions to guarantee monotonic policy improvement in the presence of critic error (Sec. 4.1). In Sec. 4.2, we prove that Algorithm 1 is guaranteed to converge to the neighbourhood (that depends on the critic error) of a stationary point.

### 4.1 Conditions for monotonic policy improvement

According to **inequality (I)**, to guarantee monotonic policy improvement at iteration $t$, one must find a $(\theta, c)$ pair to guarantee that the RHS of **(I)** is positive. In Prop. 2 (proved in App. D), we derive the conditions on the critic error to ensure that it possible to find such an $(\theta, c)$ pair.

**Proposition 2.** For any policy representation and any policy or critic parameterization, there exists a $(\theta, c)$ pair that makes the RHS of **inequality (I)** strictly positive, and hence guarantees monotonic policy improvement ($J(\pi_{t+1}) > J(\pi_t)$), if and only if

$$\langle b_t, \tilde{H}_t^\dagger b_t \rangle > \langle [\nabla J(\pi_t) - \hat{g}_t], [\nabla_\pi^2 \Phi(\pi_t)]^{-1} [\nabla J(\pi_t) - \hat{g}_t] \rangle,$$

where $b_t \in \mathbb{R}^n := \sum_{s \in \mathcal{S}} \sum_{a \in \mathcal{A}} [\hat{g}_t]_{s,a} \nabla_\theta [\pi(\theta_t)]_{s,a}$, $\tilde{H}_t \in \mathbb{R}^{n \times n} := \nabla_\theta \pi(\theta_t)^\intercal \nabla_\pi^2 \Phi(\pi_t) \nabla_\theta \pi(\theta_t)$ and $\tilde{H}_t^\dagger$ denotes the pseudo-inverse of $\tilde{H}_t$. For the special case of the tabular policy parameterization, the above condition becomes equal to,

$$\langle \hat{g}_t, [\nabla_\pi^2 \Phi(\pi_t)]^{-1} \hat{g}_t \rangle > \langle [\nabla J(\pi_t) - \hat{g}_t], [\nabla_\pi^2 \Phi(\pi_t)]^{-1} [\nabla J(\pi_t) - \hat{g}_t] \rangle.$$

For the Euclidean mirror map with the tabular policy parameterization, this condition becomes equal to $\|\hat{g}_t\|_2^2 > \|\nabla J(\pi_t) - \hat{g}_t\|_2^2$ meaning that the relative error in estimating $\nabla J(\pi_t)$ needs to be less than 1. For a general mirror map, the relative error is measured in a different norm induced by the mirror map. The above proposition also quantifies the scenario when the critic error is too large to guarantee policy improvement. In this case, the algorithm should either improve the critic by better optimization or by using a more expressive model, or resort to using sufficiently many (high-variance) Monte-Carlo samples as in REINFORCE [59]. Finally, we see that the impact of a smaller function class for the actor is a potentially lower value for $\langle b_t, \tilde{H}_t^\dagger b_t \rangle$, making it more difficult to satisfy the condition. *The improvement guarantee in Prop. 2 holds regardless of the policy representation and parameterization of the policy or critic.* This is in contrast to existing theoretical results [41, 28, 18] that focus on either the tabular or linear function approximation setting for the policy and/or critic, or rely on using expressive models to minimize the critic error and achieve good performance for the actor. Moreover, this result only depends on the magnitude of the critic loss (after updates), irrespective of the optimizer, step-size or other factors influencing the critic optimization. The actor and critic are coupled via the threshold (on the critic loss) required to guarantee policy improvement.

### 4.2 Convergence of Algorithm 1

Prop. 2 holds when the critic error is small. We now analyze the convergence of Algorithm 1 for an arbitrary critic error. Define $\bar{\theta}_{t+1} := \arg\max_\theta \ell_t(\theta)$, $\bar{\pi}_{t+1} = \pi(\bar{\theta}_{t+1}) = \arg\max_{\pi \in \Pi} \left\{ \langle \hat{g}_t, \pi - \pi_t \rangle - \left( \frac{1}{\eta} + \frac{1}{c} \right) D_\Phi(\pi, \pi_t) \right\}$. Note that $\bar{\pi}_{t+1}$ is the iterate obtained by using the inexact mirror ascent (MA) update (because it does not use the true gradient $\nabla_\pi J(\pi_t)$) starting from $\pi_t$, and that the inner-loop (Lines 10-13) of Algorithm 1 approximates this update. This connection allows us to prove the following guarantee (see App. D.1 for details) for Algorithm 1.

---

**Proposition 3.** For any policy representation and mirror map $\Phi$ such that (i) $J + \frac{1}{\eta}\Phi$ is convex in $\pi$, any policy parameterization such that (ii) $\ell_t(\theta)$ is smooth w.r.t $\theta$ and satisfies the Polyak-Lojasiewicz (PL) condition, for $c > 0$, after $T$ iterations of Algorithm 1 we have that,

$$\mathbb{E}\left[ \frac{D_\Phi(\bar{\pi}_{\mathcal{R}+1}, \pi_\mathcal{R})}{\zeta^2} \right] \leq \frac{1}{\zeta T} \left[ \underbrace{J(\pi^*) - J(\pi_0)}_{\text{Term (i)}} + \sum_{t=0}^{T-1} \left( \underbrace{\frac{1}{c}\mathbb{E}D_{\Phi^*}\left( \nabla\Phi(\pi_t) - c\,\delta_t, \nabla\Phi(\pi_t) \right)}_{\text{Term (ii)}} + \underbrace{\mathbb{E}[e_t]}_{\text{Term (iii)}} \right) \right]$$

where $\delta_t := \nabla J(\pi_t) - \hat{g}_t$, $\frac{1}{\zeta} := \frac{1}{\eta} + \frac{1}{c}$, $\mathcal{R}$ is a random variable chosen uniformly from $\{0, 1, 2, \ldots T-1\}$ and $e_t \in \mathcal{O}(\exp(-m_a))$ is the projection error (onto $\Pi$) at iteration $t$.

---

Prop. 3 shows that Algorithm 1 converges to the neighbourhood of a stationary point of $J$ for an arbitrary critic error. The LHS of the above expression is a measure of sub-optimality similar to the one used in the analysis of stochastic mirror descent [65]. For the Euclidean mirror map, the LHS becomes equal to $\|\nabla_\pi J(\pi_R)\|_2^2$, the standard characterization of a stationary point. Term (i) on the RHS is the initial sub-optimality, whereas Term (ii) is equal to the critic error and can be further decomposed into variance and bias terms. The variance decreases as the number of samples used to train the critic (Line 4 in Algorithm 1) increases. The bias can be decomposed into an optimization error (that decreases as $m_c$ increases) and a function approximation error (that decreases as we use more expressive models for the critic). Finally, Term (iii) is the projection (onto $\Pi$) error, is equal to zero for the tabular policy parameterization, and decreases as $m_a$ increases. Hence, the performance of Algorithm 1 improves as we increase both $m_a$ and $m_c$.

In Sec. 5, we specify the step-size $\eta$ such that Assumption (i) is satisfied for both the direct and softmax representations. Assumption (ii) is satisfied when using a linear and, in some cases, a neural network policy parameterization [33]. For the above proposition to hold, we require that step-sizes $\alpha_c$ and $\alpha_a$ in Algorithm 1 be set according to the smoothness of critic ($\mathcal{L}_t(\omega)$) and actor ($\ell_t(\theta)$) objectives respectively. These choice of step-sizes guarantee ascent for the actor objective, and descent for the critic objective (refer to the proof of Prop. 3 in App. D for details). In practice, we set both step-sizes using an Armijo line-search, and refer the reader to App. F for details. Since Algorithm 1 does not update the actor and critic in a two time-scale setting, unlike [29, 60], the relative scales of the step-sizes and the number of inner iterations ($m_a, m_c$) do not affect the algorithm's performance.

In contrast to Prop. 3, Dong et al. [11] prove that their proposed algorithm results in an $O(1/T)$ convergence to the stationary point (not the neighbourhood). However, they make a strong unjustified

assumption that the minimization problem w.r.t the parameters modeling the policy and distribution over states is jointly PL. Compared to [2, 61, 36] that focus on proving convergence to the neighbourhood of the optimal value function, but bound the critic error in the $\ell_2$ or $\ell_\infty$ norm, we focus on proving convergence to the (neighbourhood) of a stationary point, but define the critic loss in a decision-aware manner that depends on $D_{\Phi^*}$. Since Algorithm 1 is not a two time-scale algorithm, unlike Konda and Tsitsiklis [29], Wu et al. [60], the proof of Prop. 3 does not require analyzing coupled recursions between the actor and critic. Furthermore, the guarantees in Prop. 3 are independent of how the critic loss is minimized. We could use any policy evaluation method in order to estimate the value function. Hence, unlike the standard two time-scale analyses, we do not make assumptions about the mixing time of the underlying Markov chain.

Compared to the existing theoretical work on general (not decision-aware) AC methods [62, 60, 8, 28, 22, 30, 18, 41, 9] that prove convergence for the tabular or linear function approximation settings, (a) our theoretical results require fewer assumptions on the function approximation. For instance, Prop. 2 holds for *any* actor or critic parameterization (including complex neural networks), while the guarantees in Prop. 3 hold for any critic parameterization, but require that $\ell_t$, the surrogate function for the actor satisfy smoothness and gradient domination properties. (b) On the other hand, since our analysis does not explicitly model how the critic error is minimized, we can only converge to the neighbourhood of a stationary point. This is in contrast to the existing two time-scale analyses that jointly analyze the actor and critic, and show convergence to a stationary point [60]. (c) Finally, we note that the proposed algorithm supports off-policy updates i.e. the actor can re-use the value estimates from the critic to update the policy multiple times (corresponding to Lines 10-13 in Algorithm 1). This is in contrast to existing theoretically principled actor-critic methods that require interacting with the environment and gathering new data after each policy update. Hence, compared to the existing literature on AC methods, Algorithm 1 is more practical, has weaker theoretical guarantees but requires fewer assumptions on the function approximation.

## 5  Instantiating the generic actor-critic framework

We now instantiate Algorithm 1 for the direct (Sec. 5.1) and softmax (Sec. 5.2) representation.

### 5.1  Direct representation

Recall that for the direct functional representation, policy $\pi$ is represented by the set of distributions $p^\pi(\cdot|s)$ over actions for each state $s \in \mathcal{S}$. Using the policy gradient theorem [53], $[\nabla_\pi J(\pi)]_{s,a} = d^\pi(s) Q^\pi(s,a)$. Similar to [57, 61], we use a weighted (across states) negative entropy mirror map implying that $D_\Phi(p^\pi, p^{\pi'}) = \sum_{s \in \mathcal{S}} d^{\pi_t}(s) D_\phi(p^\pi(\cdot|s), p^{\pi'}(\cdot|s))$ where $\phi(p^\pi(\cdot|s)) = -\sum_a p^\pi(a|s) \log(p^\pi(a|s))$ and hence, $D_\phi(p^\pi(\cdot|s), p^{\pi'}(\cdot|s)) = \text{KL}(p^\pi(\cdot|s)||p^{\pi'}(\cdot|s))$. We now instantiate **inequality (I)** in Sec. 3.2 in the proposition below (see App. E for the derivation).

**Proposition 4.** For the direct representation and negative entropy mirror map, $c > 0$, $\eta \leq \frac{(1-\gamma)^3}{2\gamma |A|}$,

$$J(\pi) - J(\pi_t) \geq C + \mathbb{E}_{s \sim d^{\pi_t}} \left[ \mathbb{E}_{a \sim p^{\pi_t}(\cdot|s)} \left[ \frac{p^\pi(a|s)}{p^{\pi_t}(a|s)} \left( \hat{Q}^{\pi_t}(s,a) - \left( \frac{1}{\eta} + \frac{1}{c} \right) \log \left( \frac{p^\pi(a|s)}{p^{\pi_t}(a|s)} \right) \right) \right] \right]$$

$$- \mathbb{E}_{s \sim d^{\pi_t}} \left[ \mathbb{E}_{a \sim p^{\pi_t}(\cdot|s)} [Q^{\pi_t}(s,a) - \hat{Q}^{\pi_t}(s,a)] + \frac{1}{c} \log \left( \mathbb{E}_{a \sim p^{\pi_t}(\cdot|s)} \left[ \exp \left( -c \left[ Q^{\pi_t}(s,a) - \hat{Q}^{\pi_t}(s,a) \right] \right) \right] \right) \right]$$

where $C$ is a constant and $\hat{Q}^{\pi_t}$ is the estimate of the action-value function for policy $\pi_t$.

For incorporating policy (with parameters $\theta$) and critic (with parameters $\omega$) parameterization, we note that $p^\pi(\cdot|s) = p^\pi(\cdot|s, \theta)$ and $\hat{Q}^\pi(s,a) = Q^\pi(s,a|\omega)$ where the model is implicit in the notation. Using the reasoning in Sec. 3.2 with Prop. 4 immediately gives us the actor and critic objectives ($\ell_t(\theta)$ and $L_t(\omega)$ respectively) at iteration $t$ and completely instantiates Algorithm 1. Observe that the critic error is asymmetric and penalizes the under/over-estimation of the $Q^\pi$ function differently. This is different from the standard squared critic loss: $E_{s \sim d^{\pi_t}} \mathbb{E}_{a \sim p^{\pi_t}(\cdot|s)} [Q^{\pi_t}(s,a) - Q^{\pi_t}(s,a|\omega)]^2$ that does not take into account the sign of the misestimation.

To demonstrate the effectiveness of the proposed critic loss, we consider a two-armed bandit example in Prop. 5 (see App. E for details) with deterministic rewards (there is no variance due to sampling), use the direct representation and tabular parameterization for the policy, linear function approximation

for the critic and compare minimizing the standard squared loss vs the decision-aware loss in Prop. 4.

**Proposition 5.** Consider a two-armed bandit example with deterministic rewards where arm 1 is optimal and has a reward $r_1 = Q_1 = 2$ whereas arm 2 has reward $r_2 = Q_2 = 1$. Consider using linear function approximation to estimate the $Q$ function i.e. $\hat{Q} = x\,\omega$ where $\omega$ is the parameter to be learned and $x$ is the feature of the corresponding arm. Let $x_1 = -2$ and $x_2 = 1$ implying that $\hat{Q}_1(\omega) = -2\omega$ and $\hat{Q}_2(\omega) = \omega$. Let $p_t$ be the probability of pulling the optimal arm at iteration $t$ and consider minimizing two alternative objectives to estimate $\omega$:

(1) Squared loss: $\omega_t^{(1)} := \arg\min \left\{ \frac{p_t}{2} \, [\hat{Q}_1(\omega) - Q_1]^2 + \frac{1-p_t}{2} \, [\hat{Q}_2(\omega) - Q_2]^2 \right\}$.

(2) Decision-aware critic loss: $\omega_t^{(2)} = \arg\min \mathcal{L}_t(\omega) := p_t \, [Q_1 - \hat{Q}_1(\omega)] + (1 - p_t) \, [Q_2 - \hat{Q}_2(\omega)] + \frac{1}{c} \log \left( p_t \exp\left( -c \, [Q_1 - \hat{Q}_1(\omega)] + (1 - p_t) \exp\left( -c \, [Q_2 - \hat{Q}_2(\omega)] \right) \right) \right)$.

For $p_0 < \frac{2}{5}$, minimizing the squared loss results in convergence to the sub-optimal action, while minimizing the decision-aware loss (for $c, p_0 > 0$) results in convergence to the optimal action.

Hence, minimizing the decision-aware critic loss results in a better, more well-informed estimate of $\omega$ which when coupled with the actor update results in convergence to the optimal arm. For this simple example, at every iteration $t$, $\mathcal{L}_t(\omega_t^{(2)}) = 0$, while the standard squared loss is non-zero at $\omega_t^{(1)}$, though we use the same linear function approximation model in both cases. In Prop. 16, we prove that for a 2-arm bandit with deterministic rewards and linear critic parameterization, minimizing the decision-aware critic loss will always result in convergence to the optimal arm.

### 5.2 Softmax representation

Recall that for the softmax functional representation, policy $\pi$ is represented by the logits $z^\pi(s, a)$ for each $s \in \mathcal{S}$ and $a \in \mathcal{A}$ such that $p^\pi(a|s) = \frac{\exp(z^\pi(s,a))}{\sum_{a'} \exp(z^\pi(s,a'))}$. Using the policy gradient theorem, $[\nabla_\pi J(\pi)]_{s,a} = d^\pi(s) \, A^\pi(s,a) \, p^\pi(a|s)$ where $A^\pi$ is the advantage function. Similar to Vaswani et al. [57], we use a weighted (across states) log-sum-exp mirror map implying that $D_\Phi(z, z') = \sum_{s \in \mathcal{S}} d^{\pi_t}(s) \, D_\phi(z(s, \cdot), z'(s, \cdot))$ where $\phi(z(s, \cdot)) = \log(\sum_a \exp(z(s, a)))$ and hence, $D_\phi(z(s, \cdot), z'(s, \cdot)) = \mathrm{KL}(p^{\pi'}(\cdot|s), p^\pi(\cdot|s))$ (see Lemma 11 for a derivation). We now instantiate **inequality (I)** in Sec. 3.2 in the proposition below (see App. E for the derivation).

**Proposition 6.** For the softmax representation and log-sum-exp mirror map, $c > 0, \eta \leq 1 - \gamma$,

$$J(\pi) - J(\pi_t) \geq \mathbb{E}_{s \sim d^{\pi_t}} \mathbb{E}_{a \sim p^{\pi_t}(\cdot|s)} \left[ \left( \hat{A}^{\pi_t}(s, a) + \frac{1}{\eta} + \frac{1}{c} \right) \log\left( \frac{p^\pi(a|s)}{p^{\pi_t}(a|s)} \right) \right]$$
$$- \frac{1}{c} \mathbb{E}_{s \sim d^{\pi_t}} \mathbb{E}_{a \sim p^{\pi_t}(\cdot|s)} \left[ \left( 1 - c \, [A^{\pi_t}(s, a) - \hat{A}^{\pi_t}(s, a)] \right) \log\left( 1 - c \, [A^{\pi_t}(s, a) - \hat{A}^{\pi_t}(s, a)] \right) \right]$$

where $\hat{A}^{\pi_t}$ is the estimate of the advantage function for policy $\pi_t$.

For incorporating policy (with parameters $\theta$) and critic (with parameters $\omega$) parameterization, we note that $p^\pi(a|s) = \frac{\exp(z^\pi(s,a|\theta))}{\sum_{a'} \exp(z^\pi(s,a'|\theta))}$ and $\hat{A}^\pi(s, a) = A^\pi(s, a|\omega)$ where the model is implicit in the notation. Using the reasoning in Sec. 3.2 with Prop. 6 immediately gives us the actor and critic objectives ($\ell_t(\theta)$ and $L_t(\omega)$ respectively) at iteration $t$ and completely instantiates Algorithm 1. Similar to the direct representation, observe that $\mathcal{L}_t$ is asymmetric and penalizes the under/over-estimation of the advantage function differently.

To demonstrate the effectiveness of the proposed critic loss, we construct a two-armed bandit example in Prop. 7 below (see App. E for details), use the softmax representation and tabular parameterization for the policy and consider a discrete hypothesis class (with two hypotheses) as the model for the critic. We compare minimizing the squared loss on the advantage: $E_{s \sim d^{\pi_t}} \mathbb{E}_{a \sim p^{\pi_t}(\cdot|s)} [A^{\pi_t}(s, a) - A^{\pi_t}(s, a|\omega)]^2$ with minimizing the decision-aware loss. We see that minimizing the decision-aware critic loss can distinguish between the two hypotheses and choose the correct hypothesis resulting in convergence to the optimal action.

**Proposition 7.** Consider a two-armed bandit example and define $p \in [0, 1]$ as the probability of pulling arm 1. Given $p$, let the advantage of arm 1 be equal to $A_1 := \frac{1}{2} > 0$, while that of arm 2 is $A_2 := -\frac{p}{2(1-p)} < 0$ implying that arm 1 is optimal. For $\varepsilon \in \left( \frac{1}{2}, 1 \right)$, consider approximating the advantage of the two arms using a function approximation model with two hypotheses that

depend on $p$: $\mathcal{H}_0 : \hat{A}_1 = \frac{1}{2} + \varepsilon$, $\hat{A}_2 = -\frac{p}{1-p}\left(\frac{1}{2} + \varepsilon\right)$ and $\mathcal{H}_1 : \hat{A}_1 = \frac{1}{2} - \varepsilon\,\mathrm{sgn}\left(\frac{1}{2} - p\right)$, $\hat{A}_2 = -\frac{p}{1-p}\left(\frac{1}{2} - \varepsilon\,\mathrm{sgn}\left(\frac{1}{2} - p\right)\right)$ where sgn is the signum function. If $p_t$ is the probability of pulling arm 1 at iteration $t$, consider minimizing two alternative loss functions to choose the hypothesis $\mathcal{H}_t$:

(1) Squared loss: $\mathcal{H}_t = \arg\min_{\{\mathcal{H}_0,\mathcal{H}_1\}} \left\{ \frac{p_t}{2}\left[A_1 - \hat{A}_1\right]^2 + \frac{1-p_t}{2}\left[A_2 - \hat{A}_2\right]^2 \right\}$.

(2) Decision-aware critic loss with $c = 1$: $\mathcal{H}_t = \arg\min_{\{\mathcal{H}_0,\mathcal{H}_1\}}$

$\left\{ p_t\left(1 - [A_1 - \hat{A}_1]\right)\log(1 - [A_1 - \hat{A}_1]) + (1 - p_t)\left(1 - [A_2 - \hat{A}_2]\right)\log(1 - [A_2 - \hat{A}_2]) \right\}$.

For $p_0 \leq \frac{1}{2}$, the squared loss cannot distinguish between $\mathcal{H}_0$ and $\mathcal{H}_1$, and depending on how ties are broken, minimizing it can result in convergence to the sub-optimal action. On the other hand, minimizing the divergence loss (for any $p_0 > 0$) results in convergence to the optimal arm.

In Prop. 13 in App. E, we study the softmax representation with the Euclidean mirror map and instantiate **inequality (I)** for this case. Finally, in App. B, we instantiate our actor-critic framework to handle stochastic value gradients used for learning continuous control policies [21]. In the next section, we consider simple RL environments to empirically benchmark Algorithm 1.

## 6 Experiments

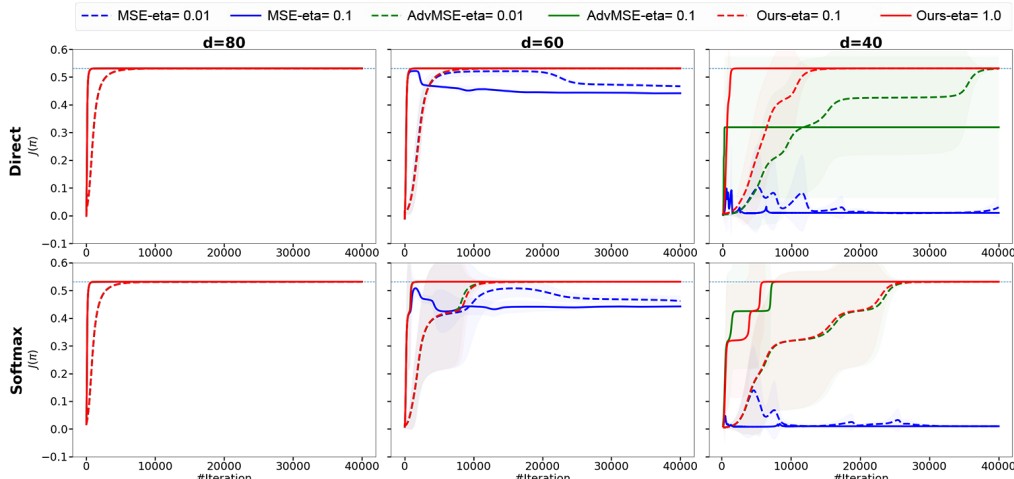

Figure 1: Comparison of decision-aware, `Adv-MSE` and `MSE` loss functions using a linear actor and linear (with three different dimensions) critic in the Cliff World environment for direct and softmax policy representations. For $d = 80$ (corresponding to an expressive critic), all algorithms have the same performance. For $d = 40$ and $d = 60$, `MSE` does not have monotonic improvement and converges to a sub-optimal policy. `Adv-MSE` almost always reaches the optimal policy. Compared to the `Adv-MSE` and `MSE`, minimizing the decision-aware loss always results in convergence to the optimal policy at a faster rate, especially when using a less expressive critic ($d = 40$).

We demonstrate the benefit of the decision-aware framework over the standard AC algorithm where the critic is trained by minimizing the squared error. We instantiate Algorithm 1 for the direct and softmax representations, and evaluate the performance on two grid-world environments, namely Cliff World [53] and Frozen Lake [6] (see App. F for details). We compare the performance of three AC algorithms that have the same actor, but differ in the objective function used to train the critic.

**Critic Optimization**: For the direct and softmax representations, the critic's objective is to estimate the action-value ($Q$) and advantage ($A$) functions respectively. We use a linear parameterization for the $Q$ function implying that for each policy $\pi$, $Q^\pi(s, a|\omega) = \langle \omega, \mathbf{X}(s, a)\rangle$, where $\mathbf{X}(s, a) \in \mathbb{R}^d$ are features obtained via tile-coding [53, Ch. 9]. We vary the dimension $d \in \{80, 60, 40\}$ of the tile-coding features to vary the expressivity of the critic. Given the knowledge of $p^\pi$ and the estimate $Q^\pi(s, a|\omega)$, the estimated advantage can be obtained as: $A^\pi(s, a|\omega) = Q^\pi(s, a|\omega) - \sum_a p^\pi(a|s)\, Q^\pi(s, a|\omega)$. We consider two ways to estimate the $Q$ function for training the critic: (a) using the known MDP to exactly compute the $Q$ values and (b) estimating the $Q$ function using Monte-Carlo (MC) rollouts. There are three sources of error for an insufficient critic – the bias due to limited model capacity, the optimization error due to an insufficient number of inner iterations ($m_c$) and the variance due to Monte-Carlo sampling. We use a large value of $m_c$ and sufficiently large

Monte-Carlo rollouts to control the optimization error and variance respectively (see App. F). This ensures that the bias dominates, and enables us to isolate the effect of the form of the critic loss.

We evaluate the performance of the decision-aware loss defined for the direct (Prop. 4) and softmax representations (Prop. 6). For both representations, we minimize the corresponding objective at each iteration $t$ (Lines 6-8 in Algorithm 1) using gradient descent with the step-size $\alpha_c$ determined by the Armijo line-search [4]. We use a grid-search to tune the trade-off parameter $c$, and propose an alternative albeit conservative method to estimate $c$ in App. F. We compare against two baselines (see App. F for implementation details) – (i) the standard squared loss on the $Q$ functions (referred to as `MSE` in the plots) defined in Prop. 5 and (ii) squared loss on $A$ function (referred to as `Adv-MSE` in the plots) defined in Prop. 7. We note that the `Adv-MSE` loss corresponds to a second-order Taylor series expansion of the decision-aware loss (see Prop. 14 for details), and is similar to the loss in Pan et al. [42]. Recall that the critic error consists of the variance when using MC samples (equal to zero when we exactly compute the $Q$ function) and the bias because of the critic optimization error (controlled since the critic objective is convex) and error due to the limited expressivity of the linear function approximation (decreases as $d$ increases). Since our objective is to study the effect of the critic loss and its interaction with function approximation, we do not use bootstrapping to estimate the $Q^\pi$ since it would result in a confounding bias term.

**Actor Optimization**: For all algorithms, we use the same actor objective defined for the direct (Prop. 4) and softmax representations (Prop. 6). We consider both the tabular and linear policy paramterization for the actor. For the linear function approximation, we use the same tile-coded features and set $n = 60$ for both environments. We update the policy parameters at each iteration $t$ in the off-policy inner-loop (Lines 11-13 in Algorithm 1) using Armijo line-search to set $\alpha_a$. For details about the derivatives and closed-form solutions for the actor objective, refer to [57] and App. F. We use a grid-search to tune $\eta$, and compare different values. Our experimental setup [1] enables us to isolate the effect of the critic loss, without non-convexity or optimization issues acting as confounders.

**Results**: For each environment, we conduct four experiments that depend on (a) whether we use MC samples or the true dynamics to estimate the $Q$ function, and (b) on the policy parameterization. We only show the plot corresponding to using the true dynamics for estimating the $Q$ function and linear policy parameterization, and defer the remaining plots to App. G. For all experiments, we report the mean and 95% confidence interval of $J(\pi)$ averaged across 5 runs. In the main paper, we only include 2 values of $\eta \in \{0.01, 0.1\}$ and vary $d \in \{40, 60, 80\}$, and defer the complete figure with a broader range of $\eta$ and $d$ to App. G. For this experiment, $c$ is tuned to 0.01 and we include a sensitivity (of $J(\pi)$ to $c$) plot in App. G. From Fig. 1, we see that (i) with a sufficiently expressive critic ($d = 80$), all algorithms reach the optimal policy at nearly the same rate. (ii) as we decrease the critic capacity, minimizing the `MSE` loss does not result in monotonic improvement and converges to a sub-optimal policy, (iii) minimizing the `Adv-MSE` usually results in convergence to the optimal policy, whereas (iv) minimizing the decision-aware loss results in convergence to better policies at a faster rate, and is more beneficial when using a less-expressive critic (corresponding to $d = 40$). We obtain similar results for the tabular policy parameterization or when using sampling to estimate the $Q$ function (see App. G for additional results).

## 7  Discussion

We designed a generic decision-aware actor-critic framework where the actor and critic are trained cooperatively to optimize a joint objective. Our framework can be used with any policy representation and easily handle general policy and critic parameterization, while preserving theoretical guarantees. Instantiating the framework resulted in an actor that supports off-policy updates, and a corresponding critic loss that can be minimized using first-order optimization. We demonstrated the benefit of our framework both theoretically and empirically. We note that Algorithm 1 can be directly used with any complex actor/critic parameterization in order to generalize across states/actions. The theoretical guarantees of Prop. 2 would still hold. From a practical perspective, w.r.t tuning hyper-parameters, $\eta$ does not depend on the actor/critic parameterization. On the other hand, $\alpha_a$ and $\alpha_c$ are set adaptively using an Armijo line-search that only requires the smoothness of actor/critic objectives, and does not depend on their convexity. However, Algorithm 1 does require tuning hyper-parameter $c$, and we will aim to investigate automatic adaptive ways to set it. In the future, we aim to benchmark Algorithm 1 for complex deep RL environments. Finally, we aim to broaden the scope our framework to applications such as variational inference.

---

[1]Code to reproduce the experiments is available at https://github.com/amirrezakazemi/ACPG

## Acknowledgements

We would like to thank Michael Lu for feedback on the paper. This research was partially supported by the Canada CIFAR AI Chair program, the Natural Sciences and Engineering Research Council of Canada (NSERC) Discovery Grant RGPIN-2022-04816.

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

# Supplementary material

## Organization of the Appendix

## A   Definitions

- **[Solution set]**. We define the solution set $\mathcal{X}^*$ for a function $f$ as $\mathcal{X}^* := \{x^* | x^* \in \arg\min_{x \in \text{dom}(f)} f(x)\}$.

- **[Convexity]**. A differentiable function $f$ is convex iff for all $v$ and $w$ in $\text{dom}(f)$

$$f(v) \geq f(w) + \langle \nabla f(w),\, v - w \rangle. \tag{Convexity}$$

- **[Lipschitz continuity]**. A differentiable function $f$ is $G$-Lipschitz continuous, meaning that for all $v$ and $w$ and constant $G >$,

$$|f(v) - f(w)| \leq G \|v - w\| \implies \|\nabla f(v)\| \leq G. \tag{Lipschitz Continuity}$$

- **[Smoothness]**. A differentiable function $f$ is $L$-smooth, meaning that for all $v$ and $w$ and some constant $L > 0$

$$f(v) \leq f(w) + \langle \nabla f(w),\, v - w \rangle + \frac{L}{2} \|v - w\|_2^2. \tag{Smoothness}$$

- **[Polyak-Lojasiewicz inequality]**. A differentiable function $f$ satisfies the Polyak-Lojasiewicz (PL) inequality if there exists a constant $\mu_p > 0$ s.t. for all $v$,

$$\mu_p(f(v) - f^*) \leq \frac{1}{2} \|\nabla f(v)\|_2^2, \tag{PL}$$

where $f^*$ is the optimal function value i.e. $f^* := f(x^*)$ for $x^* \in \mathcal{X}^*$.

- **[Restricted Secant Inequality]**. A differentiable function $f$ satisfies the Restricted Secant Inequality (RSI) inequality if there exists a constant $\mu_r > 0$ that for all $v$

$$\langle \nabla f(v),\, v - v_p \rangle \geq \mu_r \|v - v_p\|_2^2, \tag{RSI}$$

where $v_p$ is the projection of $v$ onto $\mathcal{X}^*$.

- **[Bregman divergence]**. For a strictly-convex, differentiable function $\Phi$, we define the Bregman divergence induced by $\Phi$ (known as the mirror map) as:

$$D_\Phi(w, v) := \Phi(w) - \Phi(v) - \langle \nabla \Phi(v),\, w - v \rangle. \tag{Bregman divergence}$$

- **[Relative smoothness]**. A function $f$ is $\rho$-relatively smooth w.r.t. $D_\Phi$ iff $f + \rho\Phi$ is convex. Furthermore, if $f$ is $\rho$-relatively smooth w.r.t. $\Phi$, then, $|f(w) - f(v) - \langle \nabla f(v),\, w - v \rangle| \leq \rho\, D_\Phi(w, v)$.

- **[Mirror Ascent]**. Optimizing $\max_{x \in \mathcal{X}} f(x)$ using mirror ascent (MA), if $x_t$ is the current iterate, then the update at iteration $t \in \{0, 1, \ldots, T - 1\}$ with a step-size $\eta_t$ and mirror map $\Phi$ is given as:

$$x_{t+1} := \arg\max_{x \in \mathcal{X}} \left\{ \langle \nabla f(x_t),\, x \rangle - \frac{1}{\eta_t} D_\Phi(x, x_t) \right\}, \tag{MD update}$$

The above update can be formulated into two steps Bubeck [7, Chapter 4] as follows:

$$y_{t+1} := (\nabla\Phi)^{-1}\left(\nabla\Phi(x_t) + \eta_t \nabla f(x_t)\right) \qquad \text{(Move in dual space)}$$
$$x_{t+1} := \arg\min_{x\in\mathcal{X}}\left\{D_\Phi(x, y_{t+1})\right\} \qquad \text{(Projection step)}$$

## B  Extension to stochastic value gradients

In Sec. 5, we have seen alternative ways to represent a policy's conditional distributions over actions $p^\pi(\cdot|s)$ for each state $s \in \mathcal{S}$. On the other hand, stochastic value gradients [21] represent a policy by a set of actions. Formally, if $\varepsilon$ are random variables drawn from a fixed distribution $\chi$, then policy $\pi$ is a deterministic map from $\mathcal{S} \times \chi \to \mathcal{A}$. This corresponds to the functional representation of the policy, and is particularly helpful for continuous control, i.e. when the action-space is continuous. The action $a$ chosen by $\pi$ in state $s$, when fixing the random variable $\varepsilon = \epsilon$, is represented as $\pi(s, \epsilon)$, and the value function for policy $\pi$ is given as:

$$J(\pi) = \sum_s d^\pi(s) \int_{\varepsilon\sim\chi} r(s, \pi(s, \varepsilon))\, d\varepsilon \qquad (1)$$

and Silver et al. [49] showed that $\frac{\partial J(\pi)}{\partial \pi(s, \epsilon)} = d^\pi(s)\nabla_a Q^\pi(s, a)\big|_{a=\pi(s,\epsilon)}$. In order to characterize the dependence on the policy parameterization, we note that $\pi(s, \epsilon) = \pi(s, \epsilon, \theta)$ where $\theta$ are the model parameters. For a fixed $\epsilon$, we will use a Euclidean mirror map implying that $D_\Phi(\pi, \pi') = \sum_{s\in\mathcal{S}} d^{\pi_t}(s)D_\phi(\pi(s, \epsilon), \pi'(s, \epsilon))$ and choose $\phi(\pi(s, \epsilon)) = \frac{1}{2}\|\pi'(s, \epsilon)\|_2^2$ implying that $D_\phi(\pi(s, \epsilon), \pi'(s, \epsilon)) = \frac{1}{2}\left[\pi(s, \epsilon) - \pi'(s, \epsilon)\right]^2$. In order to instantiate the generic lower bound in Prop. 1 at iteration $t$, we prove the following proposition in App. E.

**Proposition 8.** For the stochastic value gradient representation and Euclidean mirror map, $c > 0$, $\eta$ such that $J + \frac{1}{\eta}\Phi$ is convex in $\pi$.

$$J(\pi) - J(\pi_t) \geq C + \mathbb{E}_{s\sim d^{\pi_t}}\mathbb{E}_{\varepsilon\sim\chi}\left[\widehat{\nabla_a Q^{\pi_t}}(s, a)\big|_{a=\pi_t(s,\varepsilon)} \pi(s, \varepsilon) - \frac{1}{2}\left(\frac{1}{\eta} + \frac{1}{c}\right)\left[\pi_t(s, \epsilon) - \pi(s, \epsilon)\right]^2\right]$$

$$- \frac{c}{2}\mathbb{E}_{s\sim d^{\pi_t}}\mathbb{E}_{\varepsilon\sim\chi}\left[\nabla_a Q^{\pi_t}(s, a)\big|_{a=\pi_t(s,\varepsilon)} - \widehat{\nabla_a Q^{\pi_t}}(s, a)\big|_{a=\pi_t(s,\varepsilon)}\right]^2$$

where $C$ is a constant and $\widehat{\nabla_a Q^{\pi_t}}(s, a)\big|_{a=\pi_t(s,\varepsilon)}$ is the estimate of the action-value gradients for policy $\pi$ at state $s$ and $a = \pi_t(s, \epsilon)$.

For incorporating policy (with parameters $\theta$) and critic (with parameters $\omega$) parameterization, we note that $\pi(s, \varepsilon) = \pi(s, \varepsilon|\theta)$ and $\widehat{\nabla_a Q^{\pi_t}}(s, a)_{a=\pi_t(s,\varepsilon)} = \nabla_a Q^{\pi_t}(s, a|\omega)_{a=\pi_t(s,\varepsilon,\theta_t)}$ where the model is implicit in the notation. Using the reasoning in Sec. 3.2 with Prop. 8 immediately gives us the actor and critic objectives ($\ell_t(\theta)$ and $L_t(\omega)$ respectively) at iteration $t$ and completely instantiates Algorithm 1. The actor objective is similar to Eq (15) of Silver et al. [49], with the easier to compute $Q^{\pi_t}$ instead of $Q^\pi$, whereas the critic objective is similar to the one used in existing work on policy-aware model-based RL for continuous control [13].

# C   Proofs for Sec. 3

**Proposition 1.** For any policy representations $\pi$ and $\pi'$, any strictly convex mirror map $\Phi$, and any gradient estimator $\hat{g}$, for $c > 0$ and $\eta$ such that $J + \frac{1}{\eta}\Phi$ is convex in $\pi$,

$$J(\pi) \geq J(\pi') + \langle \hat{g}(\pi'), \pi - \pi' \rangle - \left( \frac{1}{\eta} + \frac{1}{c} \right) D_\Phi(\pi, \pi') - \frac{1}{c} D_{\Phi^*}\left( \nabla\Phi(\pi') - c[\nabla J(\pi') - \hat{g}(\pi')], \nabla\Phi(\pi') \right)$$

where $\Phi^*$ is the Fenchel conjugate of $\Phi$ and $D_{\Phi^*}$ is the Bregman divergence induced by $\Phi^*$.

*Proof.* For any $\eta$ such that $J + \frac{1}{\eta}\Phi$ is convex, we use Lemma 2 to form the following lower-bound,

$$J(\pi) \geq J(\pi') + \langle \nabla J(\pi'), (\pi - \pi') \rangle - \frac{1}{\eta} D_\phi(\pi, \pi')$$

$$= J(\pi') + \langle \hat{g}(\pi'), (\pi - \pi') \rangle + \langle \nabla J(\pi') - \hat{g}(\pi'), (\pi - \pi') \rangle - \frac{1}{\eta} D_\phi(\pi, \pi')$$

Defining $\delta := \nabla J(\pi') - \hat{g}(\pi')$, and assuming that $c\,\delta$ is small enough to satisfy the requirement for Lemma 1, we use Lemma 1 with $x = \delta$, $y = \pi$ and $y' = \pi'$.

$$= J(\pi') + \langle \hat{g}(\pi'), (\pi - \pi') \rangle - \frac{1}{\eta} D_\phi(\pi, \pi') - \frac{1}{c} \left[ D_\phi(\pi, \pi') + D_{\phi^*}\left( \nabla\phi(\pi') - c\delta, \nabla\phi(\pi') \right) \right]$$

$$\implies J(\pi) \geq J(\pi') + \hat{g}(\pi')^\top (\pi - \pi') - \left( \frac{1}{\eta} + \frac{1}{c} \right) D_\Phi(\pi, \pi') - \frac{1}{c} D_{\phi^*}\left( \nabla\phi(\pi') - c[\nabla J(\pi') - \hat{g}(\pi')], \nabla\phi(\pi') \right)$$

$\square$

**Lemma 1** (Bregman Fenchel-Young). *Let $x \in \mathcal{Y}^*$, $y \in \mathcal{Y}$, $y' \in \mathcal{Y}$. Then, for sufficiently small $c > 0$ and $x$ s.t. $(\nabla\phi)^{-1}[\nabla\phi(y') - c\,x] \in \mathcal{Y}$, we have*

$$\langle y - y', x \rangle \geq -\frac{1}{c} \left[ D_\phi(y, y') + D_{\phi^*}(\nabla\phi(y') - c\,x, \nabla\phi(y')) \right]. \tag{2}$$

*For a fixed $y'$, this inequality is tight for $y = \arg\min_v \left\{ \langle x, v - y' \rangle + \frac{1}{c} D_\Phi(v, y) \right\}$.*

*Proof.* Define $f(y) := \langle x, y - y' \rangle + \frac{1}{c} D_\Phi(y, y')$. If $y^* = \arg\min f(y)$, then,

$$\nabla f(y^*) = 0 \implies \nabla\phi(y^*) = \nabla\phi(y') - cx$$

$$y^* = (\nabla\phi)^{-1}[\nabla\phi(y') - cx] \implies y^* = \nabla\phi^*[\nabla\phi(y') - cx]$$

Note that according to our assumption, $y^* \in \mathcal{Y}$. For any $y$,

$$f(y) \geq f(y^*) = \langle x, y^* - y' \rangle + \frac{1}{c} D_\Phi(y^*, y') \tag{3}$$

In order to simplify $D_\Phi(y^*, y')$, we will use the definition of $\phi^*(z)$. In particular, for any $y$,

$$\phi(y) = \max_z \left[ \langle z, y \rangle - \phi^*(z) \right] \quad ; \quad z^* = \arg\max_z \left[ \langle z, y \rangle - \phi^*(z) \right] \implies y = \nabla\phi^*(z^*) \implies z^* = \nabla\phi(y)$$

$$\implies \phi(y) = \langle \nabla\phi(y), y \rangle - \phi^*(\nabla\phi(y)) \tag{4}$$

$$D_\Phi(y^*, y') = \phi(y^*) - \phi(y') - \langle \nabla\phi(y'), y^* - y' \rangle$$

$$= [\langle \nabla\phi(y^*), y^* \rangle - \phi^*(\nabla\phi(y^*))] - \phi(y') - \langle \nabla\phi(y'), y^* - y' \rangle$$

(using Eq. (4) to simplify the first term)

Let us focus on the first term and simplify it,

$$\langle \nabla\phi(y^*), y^* \rangle - \phi^*(\nabla\phi(y^*)) = \langle \nabla\phi \left( \nabla\phi^*[\nabla\phi(y') - cx] \right), \nabla\phi^*[\nabla\phi(y') - cx] \rangle - \phi^*\left( \nabla\phi \left( \nabla\phi^*[\nabla\phi(y') - cx] \right) \right)$$

$$= \langle [\nabla\phi(y') - cx], \nabla\phi^*[\nabla\phi(y') - cx] \rangle - \phi^*([\nabla\phi(y') - cx]) \quad \text{(For any } z, \nabla\phi(\nabla\phi^*(z)) = z)$$

Using the above relations,

$$D_\Phi(y^*, y') = \langle [\nabla\phi(y') - cx], \nabla\phi^*[\nabla\phi(y') - cx]\rangle - \phi^*([\nabla\phi(y') - cx]) - \phi(y')$$
$$- \langle \nabla\phi(y'), \nabla\phi^*[\nabla\phi(y') - cx] - y'\rangle$$
$$= \langle \nabla\phi(y'), \nabla\phi^*[\nabla\phi(y') - cx]\rangle - c\langle x, \nabla\phi^*[\nabla\phi(y') - cx]\rangle$$
$$- \phi^*([\nabla\phi(y') - cx]) - \phi(y') - \langle \nabla\phi(y'), \nabla\phi^*[\nabla\phi(y') - cx] - y'\rangle$$
$$\implies D_\Phi(y^*, y') = -c\langle x, \nabla\phi^*[\nabla\phi(y') - cx]\rangle - \phi^*([\nabla\phi(y') - cx]) - \phi(y') + \langle \nabla\phi(y'), y'\rangle$$

Using the above simplification with Eq. (3),

$$f(y) \geq \langle x, y^* - y'\rangle + \frac{1}{c}\left[ -c\langle x, \nabla\phi^*[\nabla\phi(y') - cx]\rangle - \phi^*([\nabla\phi(y') - cx]) - \phi(y') + \langle \nabla\phi(y'), y'\rangle \right]$$

$$= \langle x, y^* - y'\rangle - \langle x, \nabla\phi^*[\nabla\phi(y') - cx]\rangle - \frac{1}{c}\left[ \phi^*([\nabla\phi(y') - cx]) + \phi(y') - \langle \phi(y'), y\rangle \right]$$

$$= -\langle x, y'\rangle + \langle x, \nabla\phi^*[\nabla\phi(y') - cx]\rangle - \langle x, \nabla\phi^*[\nabla\phi(y') - cx]\rangle - \frac{1}{c}\left[ \phi^*([\nabla\phi(y') - cx]) + \phi(y') - \langle \nabla\phi(y'), y'\rangle \right]$$

$$= -\langle x, y'\rangle - \frac{1}{c}\left[ \phi^*([\nabla\phi(y') - cx]) + \phi(y') - \langle \nabla\phi(y'), y'\rangle \right]$$

Using Eq. (4), $\phi(y') = \langle \nabla\phi(y'), y'\rangle - \phi^*(\nabla\phi(y')) \implies \phi(y') - \langle \nabla\phi(y'), y'\rangle = -\phi^*(\nabla\phi(y'))$,

$$\implies f(y) \geq -\langle x, y'\rangle - \frac{1}{c}\left[ \phi^*([\nabla\phi(y') - cx]) - \phi^*(\nabla\phi(y')) \right] = -\frac{1}{c}\left[ c\langle x, y\rangle + \phi^*([\nabla\phi(y') - cx]) - \phi^*(\nabla\phi(y')) \right]$$

$$= -\frac{1}{c}\left[ c\langle x, y'\rangle + [\phi^*([\nabla\phi(y') - cx]) - \phi^*(\nabla\phi(y')) - \langle \nabla\phi^*(\nabla\phi(y')), \nabla\phi(y') - cx - \nabla\phi(y')\rangle] \right.$$

$$\left. + \langle \nabla\phi^*(\nabla\phi(y')), \nabla\phi(y') - cx - \nabla\phi(y')\rangle \right]$$

$$\implies f(y) \geq -\frac{1}{c}\left[ c\langle x, y'\rangle + D_\Phi^*(\nabla\phi(y') - cx, \nabla\phi(y')) + \langle y', -cx\rangle \right] = -\frac{1}{c}D_\Phi^*(\nabla\phi(y') - cx, \nabla\phi(y'))$$

Using the definition of $f(y)$,

$$\langle x, y - y'\rangle + \frac{1}{c}D_\Phi(y, y') \geq -\frac{1}{c}D_\Phi^*(\nabla\phi(y') - cx, \nabla\phi(y'))$$

$$\implies \langle x, y - y'\rangle \geq -\frac{1}{c}\left[ D_\Phi(y, y') + D_\Phi^*(\nabla\phi(y') - cx, \nabla\phi(y')) \right]$$

$\square$

**Lemma 2.** *If $J + \frac{1}{\eta}\Phi$ is convex, then, $J(\pi)$ is $\frac{1}{\eta}$-relatively smooth w.r.t to $D_\Phi$, and satisfies the following inequality,*

$$J(\pi) \geq J(\pi') + \langle \nabla_\pi J(\pi'), \pi - \pi'\rangle - \frac{1}{\eta}D_\Phi(\pi, \pi')$$

*Proof.* If $J + \frac{1}{\eta}\Phi$ is convex,

$$\left( J + \frac{1}{\eta}\phi \right)(\pi) \geq \left( J + \frac{1}{\eta}\phi \right)(\pi') + \left\langle \pi - \pi', \nabla_\pi \left( J + \frac{1}{\eta}\phi \right)(\pi') \right\rangle$$

$$\implies J(\pi) \geq J(\pi') + \langle \pi - \pi', \nabla_\pi J(\pi')\rangle - \frac{1}{\eta}\left[ \phi(\pi) - \phi(\pi') - \langle \nabla_\pi\phi(\pi'), \pi - \pi'\rangle \right]$$

$$\implies J(\pi) \geq J(\pi') + \langle \pi - \pi', \nabla_\pi J(\pi')\rangle - \frac{1}{\eta}D_\Phi(\pi, \pi')$$

$\square$

# D  Proofs for Sec. 4

**Proposition 2.** For any policy representation and any policy or critic parameterization, there exists a $(\theta, c)$ pair that makes the RHS of **inequality (I)** strictly positive, and hence guarantees monotonic policy improvement $(J(\pi_{t+1}) > J(\pi_t))$, if and only if

$$\langle b_t, \tilde{H}_t^\dagger b_t \rangle > \langle [\nabla J(\pi_t) - \hat{g}_t], [\nabla_\pi^2 \Phi(\pi_t)]^{-1} [\nabla J(\pi_t) - \hat{g}_t] \rangle,$$

where $b_t \in \mathbb{R}^n := \sum_{s \in \mathcal{S}} \sum_{a \in \mathcal{A}} [\hat{g}_t]_{s,a} \nabla_\theta [\pi(\theta_t)]_{s,a}$, $\tilde{H}_t \in \mathbb{R}^{n \times n} := \nabla_\theta \pi(\theta_t)^\top \nabla_\pi^2 \Phi(\pi_t) \nabla_\theta \pi(\theta_t)$ and $\tilde{H}_t^\dagger$ denotes the pseudo-inverse of $\tilde{H}_t$. For the special case of the tabular policy parameterization, the above condition becomes equal to,

$$\langle \hat{g}_t, [\nabla_\pi^2 \Phi(\pi_t)]^{-1} \hat{g}_t \rangle > \langle [\nabla J(\pi_t) - \hat{g}_t], [\nabla_\pi^2 \Phi(\pi_t)]^{-1} [\nabla J(\pi_t) - \hat{g}_t] \rangle.$$

*Proof.* As a warmup, let us first consider the tabular parameterization where $\pi(\theta) = \theta \in \mathbb{R}^{SA}$. In this case, the lower-bound in Prop. 1 is equal to,

$$J(\pi) - J(\pi_t) \geq \langle \hat{g}_t, \theta - \theta_t \rangle - \left( \frac{1}{\eta} + \frac{1}{c} \right) D_\Phi(\theta, \theta_t) - \frac{1}{c} D_{\Phi^*} \left( \nabla \Phi(\theta_t) - c[\nabla J(\theta_t) - \hat{g}_t], \nabla \Phi(\theta_t) \right)$$

We shall do a second-order Taylor expansion of the critic objective (blue term) in $c$ around 0 and a second-order Taylor expansion of the actor objective (green term) around $\theta = \theta_t$. Defining $\delta := \nabla J(\theta_t) - \hat{g}_t$,

$$\text{RHS} = \langle \hat{g}_t, \theta - \theta_t \rangle - \frac{1}{2} \left( \frac{1}{\eta} + \frac{1}{c} \right) (\theta - \theta_t)^\top [\nabla^2 \Phi(\theta_t)] (\theta - \theta_t) - \frac{c}{2} \langle \delta, \nabla^2 \phi^*(\nabla \phi(\pi_t)) \delta \rangle + o(c) + o(\|\theta - \theta_t\|_2^2),$$

$$\text{(Using Prop. 14)}$$

where $o(c)$ and $o(\|\theta - \theta_t\|_2^2)$ consist of the higher order terms in the Taylor series expansion. A necessary and sufficient condition for monotonic improvement is equivalent to finding a $(\theta, c)$ such that RHS is positive. As $c$ tends to 0, the $\theta$ maximizing the RHS is

$$\theta^* = \theta_t + \frac{c\eta}{c + \eta} \left[ \nabla^2 \Phi(\theta_t) \right]^\dagger \hat{g}_t$$

With this choice,

$$\text{RHS} = \frac{1}{2} \frac{1}{\frac{1}{\eta} + \frac{1}{c}} \langle \hat{g}_t, \left[ \nabla^2 \Phi(\theta_t) \right]^{-1} \hat{g}_t \rangle - \frac{c}{2} \langle \delta, \nabla^2 \phi^*(\nabla \phi(\pi_t)) \delta \rangle + o(c) \qquad (o(\|\theta - \theta_t\|_2^2) \text{ is subsumed by } o(c))$$

$$= \frac{c}{2} \langle \hat{g}_t, \left[ \nabla^2 \Phi(\theta_t) \right]^{-1} \hat{g}_t \rangle - \frac{c}{2} \langle \delta, \nabla^2 \phi^*(\nabla \phi(\pi_t)) \delta \rangle + \underbrace{\frac{1}{2} \left( \frac{1}{\frac{1}{\eta} + \frac{1}{c}} - c \right) \langle \hat{g}_t, \left[ \nabla^2 \Phi(\theta_t) \right]^{-1} \hat{g}_t \rangle}_{o(c) \text{ term}} + o(c)$$

$$= \frac{c}{2} \langle \hat{g}_t, \left[ \nabla^2 \Phi(\theta_t) \right]^{-1} \hat{g}_t \rangle - \frac{c}{2} \langle \delta, \nabla^2 \phi^*(\nabla \phi(\pi_t)) \delta \rangle + o(c) \qquad \text{(Subsuming the additional } o(c) \text{ term)}$$

If $\langle \hat{g}_t, \left[ \nabla^2 \Phi(\theta_t) \right]^{-1} \hat{g}_t \rangle > \langle \delta, \nabla^2 \phi^*(\nabla \phi(\pi_t)) \delta \rangle$, i.e. there exists an $\epsilon > 0$ s.t. $\langle \hat{g}_t, \left[ \nabla^2 \Phi(\theta_t) \right]^{-1} \hat{g}_t \rangle = \langle \delta, \nabla^2 \phi^*(\nabla \phi(\pi_t)) \delta \rangle + \epsilon$, then, RHS $= \frac{c\epsilon}{2} + o(c)$. For any fixed $\kappa > 0$, since $o(c)/c \to 0$ as $c \to 0$, there exists a neighbourhood $(0, c_\kappa)$ around zero such that for all $c$ in this neighbourhood, $o(c)/c > -\kappa$ and hence $o(c) > -\kappa c$. Setting $\kappa = \frac{\varepsilon}{4}$, there is a $c$ such that

$$\text{RHS} > \frac{c\varepsilon}{4} > 0$$

Hence, there exists a $c \in (0, \min\{\eta, c_\kappa\})$ such that the RHS is positive, and is hence sufficient to guarantee monotonic policy improvement.

On the other hand, if $\langle \hat{g}_t, \left[ \nabla^2 \Phi(\theta_t) \right]^{-1} \hat{g}_t \rangle < \langle \delta, \nabla^2 \phi^*(\nabla \phi(\pi_t)) \delta \rangle$, i.e. there exists an $\epsilon > 0$ s.t. $\langle \hat{g}_t, \left[ \nabla^2 \Phi(\theta_t) \right]^{-1} \hat{g}_t \rangle = \langle \delta, \nabla^2 \phi^*(\nabla \phi(\pi_t)) \delta \rangle - \epsilon$, then, RHS $= \frac{-c\epsilon}{2} + o(c)$ which can be negative and hence monotonic improvement can not be guaranteed. Hence, $\langle \hat{g}_t, \left[ \nabla^2 \Phi(\theta_t) \right]^{-1} \hat{g}_t \rangle > \langle \delta, \nabla^2 \phi^*(\nabla \phi(\pi_t)) \delta \rangle$ is a necessary and sufficient condition for improvement.

Let us now consider the more general case, and define $m = SA$, $\hat{g}_t \in \mathbb{R}^{m \times 1}$, $\pi(\theta) \in \mathbb{R}^{m \times 1}$ is a function of $\theta \in \mathbb{R}^{n \times 1}$. Rewriting Prop. 1,

$$J(\pi) - J(\pi_t) \geq \langle \hat{g}_t, \pi(\theta) - \pi(\theta_t) \rangle - \left( \frac{1}{\eta} + \frac{1}{c} \right) D_\Phi(\pi(\theta), \pi(\theta_t)) - \frac{1}{c} D_{\Phi^*} \left( \nabla \Phi(\pi_t) - c[\nabla J(\pi_t) - \hat{g}_t], \nabla \Phi(\pi_t) \right)$$

As before, we shall do a second-order Taylor expansion of the critic objective (blue term) in $c$ around 0 and a second-order Taylor expansion of the actor objective (green term) around $\theta = \theta_t$. Defining $\delta := \nabla J(\theta_t) - \hat{g}_t$. From Prop. 14, we know that,

$$\frac{1}{c} D_{\Phi^*}\left(\nabla\Phi(\pi_t) - c[\nabla J(\pi_t) - \hat{g}_t], \nabla\Phi(\pi_t)\right) = \frac{c}{2}\langle\delta, \nabla^2\phi^*(\nabla\phi(\pi_t))\,\delta\rangle + o(c)$$

In order to calculate the second-order Taylor series expansion of the actor objective, we define $\nabla_\theta\pi(\theta_t) \in \mathbb{R}^{m\times n}$ as the Jacobian of the $\theta :\to \pi$ map, and use $\nabla_\theta[\pi(\theta_t)]_i \in \mathbb{R}^{1\times n}$ for $i \in [m]$ to refer to row $i$

$$\langle\hat{g}_t, \pi(\theta) - \pi(\theta_t)\rangle = \sum_{i=1}^m \underbrace{[\hat{g}_t]_i}_{1\times 1} \underbrace{\nabla_\theta[\pi(\theta_t)]_i}_{1\times n} \underbrace{(\theta - \theta_t)}_{n\times 1} + \frac{1}{2}\underbrace{(\theta - \theta_t)}_{1\times n}\left[\sum_{i=1}^m \underbrace{[\hat{g}_t]_i}_{1\times 1} \underbrace{\nabla_\theta^2[\pi(\theta_t)]_i}_{n\times n}\right]\underbrace{(\theta - \theta_t)}_{n\times 1} + o(\|\theta - \theta_t\|_2^2)$$

where $o(\|\theta - \theta_t\|_2^2)$ consist of the higher order terms in the Taylor series expansion. For expanding the divergence term, note that $D_\Phi(\pi(\theta), \pi(\theta_t)) = \phi(\pi(\theta)) - \phi(\pi(\theta_t)) - \langle\nabla\phi(\pi(\theta_t)), \pi(\theta) - \pi(\theta_t)\rangle$

$$\phi(\pi(\theta)) - \phi(\pi(\theta_t)) = \underbrace{\nabla_\pi\phi(\pi_t)^\top}_{1\times m} \underbrace{\nabla_\theta\pi(\theta_t)}_{m\times n}\underbrace{(\theta - \theta_t)}_{n\times 1}$$

$$+ \frac{1}{2}\underbrace{(\theta - \theta_t)^\top}_{1\times n}\left[\underbrace{\nabla_\theta\pi(\theta_t)^\top}_{n\times m}\underbrace{\nabla_\pi^2\phi(\pi_t)}_{m\times m}\underbrace{\nabla_\theta\pi(\theta_t)}_{m\times n} + \sum_{i=1}^m\underbrace{[\nabla_\pi\phi(\pi_t)]_i}_{1\times 1}\underbrace{\nabla_\theta^2[\pi(\theta_t)]_i}_{n\times n}\right]\underbrace{(\theta - \theta_t)}_{n\times 1} + o(\|\theta - \theta_t\|_2^2)$$

$$\langle\nabla\phi(\pi(\theta_t)), \pi(\theta) - \pi(\theta_t)\rangle = \sum_{i=1}^m\underbrace{[\nabla\phi(\pi_t)]_i}_{1\times 1}\underbrace{[\nabla_\theta\pi(\theta_t)]_i}_{1\times n}\underbrace{(\theta - \theta_t)}_{n\times 1} + \frac{1}{2}\underbrace{(\theta - \theta_t)}_{1\times n}\left[\sum_{i=1}^m\underbrace{[\nabla\phi(\pi_t)]_i}_{1\times 1}\underbrace{\nabla_\theta^2[\pi(\theta_t)]_i}_{n\times n}\right]\underbrace{(\theta - \theta_t)}_{n\times 1} + o(\|\theta - \theta_t\|_2^2)$$

Putting everything together,

$$\text{RHS} = \left(\sum_{i=1}^m\underbrace{[\hat{g}_t]_i}_{1\times 1}\underbrace{\nabla_\theta[\pi(\theta_t)]_i}_{1\times n}\right)\underbrace{(\theta - \theta_t)}_{n\times 1}$$

$$+ \frac{1}{2}\underbrace{(\theta - \theta_t)^\top}_{1\times n}\left[\sum_{i=1}^m\left(\underbrace{[\hat{g}_t]_i}_{1\times 1}\underbrace{\nabla_\theta^2[\pi(\theta_t)]_i}_{n\times n}\right) - \left(\frac{1}{\eta} + \frac{1}{c}\right)\underbrace{\nabla_\theta\pi(\theta_t)^\top}_{n\times m}\underbrace{\nabla_\pi^2\phi(\pi_t)}_{m\times m}\underbrace{\nabla_\theta\pi(\theta_t)}_{m\times n}\right]\underbrace{(\theta - \theta_t)}_{n\times 1}$$

$$- \frac{c}{2}\langle\delta, \nabla^2\phi^*(\nabla\phi(\pi_t))\,\delta\rangle + o(\|\theta - \theta_t\|_2^2) + o(c)$$

Defining $b_t := \sum_{i=1}^m [\hat{g}_t]_i \nabla_\theta[\pi(\theta_t)]_i$ and $H_t := \nabla_\theta\pi(\theta_t)^\top \nabla_\pi^2\phi(\pi_t)\nabla_\theta\pi(\theta_t) - \frac{1}{(\frac{1}{\eta} + \frac{1}{c})}\sum_{i=1}^m\left([\hat{g}_t]_i \nabla_\theta^2[\pi(\theta_t)]_i\right)$

$$\text{RHS} = \langle b_t, \theta - \theta_t\rangle + \frac{1}{2}\left(\frac{1}{\eta} + \frac{1}{c}\right)\langle(\theta - \theta_t), H_t(\theta - \theta_t)\rangle - \frac{c}{2}\langle\delta, \nabla^2\phi^*(\nabla\phi(\pi_t))\,\delta\rangle + o(\|\theta - \theta_t\|_2^2) + o(c)$$

As a sanity check, it can be verified that if $\pi(\theta) = \theta$, $H_t = \left(\frac{1}{\eta} + \frac{1}{c}\right)\nabla^2\Phi(\theta_t)$ and $b_t = \hat{g}_t$, and we recover the tabular result above. Notice that $\left\langle(\theta - \theta_t), \frac{1}{(\frac{1}{\eta} + \frac{1}{c})}\sum_{i=1}^m\left([\hat{g}_t]_i \nabla_\theta^2[\pi(\theta_t)]_i\right)(\theta - \theta_t)\right\rangle$ is $o(c)$ as $c$ goes to zero. Subsuming this term in $o(c)$,

$$\text{RHS} = \langle b_t, \theta - \theta_t\rangle + \frac{1}{2}\left(\frac{1}{\eta} + \frac{1}{c}\right)\langle(\theta - \theta_t), \tilde{H}_t(\theta - \theta_t)\rangle - \frac{c}{2}\langle\delta, \nabla^2\phi^*(\nabla\phi(\pi_t))\,\delta\rangle + o(\|\theta - \theta_t\|_2^2) + o(c)$$

where $\tilde{H}_t := \nabla_\theta\pi(\theta_t)^\top \nabla_\pi^2\phi(\pi_t)\nabla_\theta\pi(\theta_t)$. As before, a necessary and sufficient condition for monotonic improvement is equivalent to finding a $(\theta, c)$ such that RHS is positive. As $c$ tends to 0, the $\theta$ maximizing the RHS is

$$\theta^* = \theta_t + \frac{c\,\eta}{(c + \eta)}\left[\tilde{H}_t\right]^\dagger b_t$$

With this choice,

$$\text{RHS} = \frac{1}{2}\frac{1}{\frac{1}{\eta} + \frac{1}{c}}\langle b_t, \left[\tilde{H}_t\right]^\dagger b_t\rangle - \frac{c}{2}\langle\delta, \nabla^2\phi^*(\nabla\phi(\pi_t))\,\delta\rangle + o(c) \qquad (o(\|\theta - \theta_t\|_2^2) \text{ is subsumed in } o(c))$$

As in the tabular case, since $\frac{1}{\frac{1}{\eta}+\frac{1}{c}}$ is $o(c)$, we can subsume it, and we get that,

$$\text{RHS} = \frac{c}{2}\langle b_t, \left[\tilde{H}_t\right]^{\dagger} b_t\rangle - \frac{c}{2}\langle \delta, \nabla^2\phi^*(\nabla\phi(\pi_t))\,\delta\rangle + o(c)$$

Using the same reasoning as in the tabular case above, we can prove that

$$\langle b_t, \left[\tilde{H}_t\right]^{\dagger} b_t\rangle > \langle \delta, \nabla^2\phi^*(\nabla\phi(\pi_t))\,\delta\rangle$$

is a necessary and sufficient condition for monotonic policy improvement. Finally, we use Gorni [19, Eq (1.5)] which shows that $\nabla^2\phi^*(\nabla\phi(\pi_t)) = [\nabla_\pi^2 \Phi(\pi_t)]^{-1}$ and complete the proof.

□

### D.1 Proof of Prop. 3

The following proposition shows the convergence of inexact mirror ascent in the functional space.

**Proposition 9.** Assuming that (i) $J + \frac{1}{\eta}\Phi$ is convex in $\pi$, for a constant $c > 0$, after $T$ iterations of mirror ascent with $\frac{1}{\eta'} = \frac{2}{\eta} + \frac{2}{c}$ we have

$$\mathbb{E}\frac{D_\Phi(\bar{\pi}_{\mathcal{R}+1}, \bar{\pi}_{\mathcal{R}})}{\zeta^2} \leq \frac{1}{\zeta T}\left[[J(\pi^*) - J(\pi_0)] + \frac{1}{c}\sum_{t=0}^{T-1}\mathbb{E}D_{\phi^*}\Big(\nabla\phi(\bar{\pi}_t) - c[\nabla J(\bar{\pi}_t) - \hat{g}(\bar{\pi}_t)], \nabla\phi(\bar{\pi}_t)\Big)\right]$$

where $\zeta = \eta'/2$ and $\mathcal{R}$ is picked uniformly random from $\{0, 1, 2, \ldots T - 1\}$.

*Proof.* We divide the mirror ascent (MA) update into two steps:

$$\nabla\phi(\tilde{\pi}_{t+1}) = \nabla\phi(\bar{\pi}_t) + \eta_t'\hat{g}(\bar{\pi}_t) \implies \hat{g}(\bar{\pi}_t) = \frac{1}{\eta_t'}[\nabla\phi(\tilde{\pi}_{t+1}) - \nabla\phi(\bar{\pi}_t)]$$

$$\bar{\pi}_{t+1} = \arg\min_{\pi\in\Pi} D_\Phi(\pi, \tilde{\pi}_{t+1}).$$

We denote the above update as $\bar{\pi}_{t+1} = \text{MA}(\bar{\pi}_t)$. Using Prop. 1 with $\pi = \bar{\pi}_{t+1}, \pi' = \bar{\pi}_t$,

$$J(\bar{\pi}_{t+1}) \geq J(\bar{\pi}_t) + \hat{g}(\bar{\pi}_t)^\top(\bar{\pi}_{t+1} - \bar{\pi}_t) - \left(\frac{1}{\eta} + \frac{1}{c}\right)D_\Phi(\bar{\pi}_{t+1}, \bar{\pi}_t) - \underbrace{\frac{1}{c}D_{\phi^*}\Big(\nabla\phi(\bar{\pi}_t) - c[\nabla J(\bar{\pi}_t) - \hat{g}(\bar{\pi}_t)], \nabla\phi(\bar{\pi}_t)\Big)}_{:=\epsilon_t^c}$$

$$\geq J(\bar{\pi}_t) + \frac{1}{\eta_t'}\langle\nabla\phi(\tilde{\pi}_{t+1}) - \nabla\phi(\bar{\pi}_t), \bar{\pi}_{t+1} - \bar{\pi}_t\rangle - \left(\frac{1}{\eta} + \frac{1}{c}\right)D_\Phi(\bar{\pi}_{t+1}, \bar{\pi}_t) - \epsilon_t^c \qquad \text{(Using the update)}$$

$$\geq J(\bar{\pi}_t) + \frac{1}{\eta_t'}\left\{D_\Phi(\bar{\pi}_{t+1}, \bar{\pi}_t) + D_\Phi(\bar{\pi}_t, \tilde{\pi}_{t+1}) - D_\Phi(\bar{\pi}_{t+1}, \tilde{\pi}_{t+1})\right\} - \left(\frac{1}{\eta} + \frac{1}{c}\right)D_\Phi(\bar{\pi}_{t+1}, \bar{\pi}_t) - \epsilon_t^c$$
$$\text{(using Lemma 4)}$$

$$= J(\bar{\pi}_t) + \frac{1}{\eta_t'}\underbrace{\left\{D_\Phi(\bar{\pi}_t, \tilde{\pi}_{t+1}) - D_\Phi(\bar{\pi}_{t+1}, \tilde{\pi}_{t+1})\right\}}_{:=A} + \left(\frac{1}{\eta_t'} - \frac{1}{\eta} - \frac{1}{c}\right)D_\Phi(\bar{\pi}_{t+1}, \bar{\pi}_t) - \epsilon_t^c$$

$$\geq J(\bar{\pi}_t) + \left(\frac{1}{\eta_t'} - \frac{1}{\eta} - \frac{1}{c}\right)D_\Phi(\bar{\pi}_{t+1}, \bar{\pi}_t) - \epsilon_t^c \qquad (A \geq 0 \text{ since } \bar{\pi}_{t+1} \text{ is the projection of } \tilde{\pi}_{t+1} \text{ onto } \Pi)$$

$$\geq J(\bar{\pi}_t) + \underbrace{\left(\frac{1}{\eta} + \frac{1}{c}\right)}_{:=\frac{1}{\zeta}}D_\Phi(\bar{\pi}_{t+1}, \bar{\pi}_t) - \epsilon_t^c \qquad (\text{Sinc } 1/\eta_t' = 2/\eta + 2/c)$$

Recursing for $T$ iterations and dividing by $1/\zeta$, picking $\mathcal{R}$ uniformly random from $\{0, 1, 2, \ldots T-1\}$ and taking expectation we get

$$
\begin{aligned}
\mathbb{E}\frac{D_\Phi(\bar{\pi}_{\mathcal{R}+1}, \bar{\pi}_{\mathcal{R}})}{\zeta^2} &= \frac{1}{\zeta^2 T}\sum_{t=0}^{T-1}\mathbb{E}D_\Phi(\bar{\pi}_{t+1}, \bar{\pi}_t) \\
&\leq \frac{\mathbb{E}[J(\bar{\pi}_T) - J(\pi_0)]}{\zeta T} + \frac{\sum_{t=0}^{T-1}\mathbb{E}\epsilon_t^c}{T} \\
&\leq \frac{[J(\pi^*) - J(\pi_0)]}{\zeta T} + \frac{\sum_{t=0}^{T-1}\mathbb{E}\epsilon_t^c}{T} \\
&= \frac{1}{\zeta T}\left[[J(\pi^*) - J(\pi_0)] + \frac{1}{c}\sum_{t=0}^{T-1}\mathbb{E}D_{\phi^*}\left(\nabla\phi(\bar{\pi}_t) - c[\nabla J(\bar{\pi}_t) - \hat{g}(\bar{\pi}_t)], \nabla\phi(\bar{\pi}_t)\right)\right]
\end{aligned}
$$

$\square$

Compared to Dragomir et al. [15], D'Orazio et al. [12] that analyze stochastic mirror ascent in the smooth, non-convex setting, our analysis ensures that the (i) sub-optimality gap (the LHS in the above proposition) is always positive, and (ii) uses a different notion of variance that depends on $D_{\Phi^*}$.

Similar to Vaswani et al. [57], we assume that each $\pi \in \Pi$ is parameterized by $\theta$. In Algorithm 1, we run gradient ascent (GA) on $\ell_t(\theta)$ to compute $\pi_{t+1} = \pi(\theta_{t+1})$ and interpret the inner loop of Algorithm 1 as an approximation to the projection step in the mirror ascent update. We note that $\ell_t(\theta)$ does not have any additional randomness and is a deterministic function w.r.t $\theta$. Note that $\bar{\pi}_{t+1} =$ where $\pi_t = \pi(\theta_t)$. Assuming that $\ell_t(\theta)$ is smooth, and satisfies the PL condition [27], we get the following convergence guarantee for Algorithm 1.

**Proposition 3.** For any policy representation and mirror map $\Phi$ such that (i) $J + \frac{1}{\eta}\Phi$ is convex in $\pi$, any policy parameterization such that (ii) $\ell_t(\theta)$ is smooth w.r.t $\theta$ and satisfies the Polyak-Lojasiewicz (PL) condition, for $c > 0$, after $T$ iterations of Algorithm 1 we have that,

$$
\mathbb{E}\left[\frac{D_\Phi(\bar{\pi}_{\mathcal{R}+1}, \pi_\mathcal{R})}{\zeta^2}\right] \leq \frac{1}{\zeta T}\left[\underbrace{J(\pi^*) - J(\pi_0)}_{\text{Term (i)}} + \sum_{t=0}^{T-1}\left(\underbrace{\frac{1}{c}\mathbb{E}D_{\Phi^*}\left(\nabla\Phi(\pi_t) - c\,\delta_t, \nabla\Phi(\pi_t)\right)}_{\text{Term (ii)}} + \underbrace{\mathbb{E}[e_t]}_{\text{Term (iii)}}\right)\right]
$$

where $\delta_t := \nabla J(\pi_t) - \hat{g}_t$, $\frac{1}{\zeta} := \frac{1}{\eta} + \frac{1}{c}$, $\mathcal{R}$ is a random variable chosen uniformly from $\{0, 1, 2, \ldots T-1\}$ and $e_t \in \mathcal{O}(\exp(-m_a))$ is the projection error (onto $\Pi$) at iteration $t$.

*Proof.* For this proof, we define the following notation:

$$
\begin{aligned}
\pi_t &:= \pi(\theta_t) \\
\ell_t(\theta) &:= J(\pi_t) + \hat{g}(\pi_t)^\top(\pi(\theta) - \pi_t) - \left(\frac{1}{\eta} + \frac{1}{c}\right)D_\Phi(\pi(\theta), \pi_t) \\
\bar{\theta}_{t+1} &:= \arg\max \ell_t(\theta) \\
\bar{\pi}_{t+1} &:= \pi(\bar{\theta}_{t+1}) = \arg\max_{\pi\in\Pi}\{\langle\hat{g}_t(\pi_t), \pi - \pi_t\rangle - \frac{1}{\eta'}D_\Phi(\pi, \pi_t)\} = \text{MA}(\pi_t) \\
&\qquad\qquad\text{(Iterate obtained after running 1 step of mirror ascent starting from $\pi_t$)} \\
\theta_{t+1} &:= \text{GradientAscent}(\ell_t(\theta), \theta_t, m_a) \\
\pi_{t+1} &:= \pi(\theta_{t+1}),
\end{aligned}
$$

where $\text{GradientAscent}(\ell_t(\theta), \theta_t, m_a)$ means running GradientAscent for $m_a$ iterattions on $\ell_t(\theta)$ with the initialization equal to $\theta_t$. Since we assume that $\ell_t$ satisfies the $PL$-condition w.r.t. $\theta$ for all $t$, based on the results from Karimi et al. [27], we get

$$
\ell_t(\bar{\theta}_{t+1}) - \ell_t(\theta_{t+1}) \leq \underbrace{c_1\exp(-c_2 m_a)\left(\ell_t(\bar{\theta}_{t+1}) - \ell_t(\theta_t)\right)}_{:=e_t}
$$

where $c_1$, $c_2$ are problem-dependent constants related to the smoothness and curvature of $\ell_t$, and $e_t$ is the approximation error diminishes as we increase the value of $m_a$. Following the same steps as before,

$$J(\pi_{t+1}) \geq J(\pi_t) + \hat{g}(\pi_t)^\top(\pi(\theta_{t+1}) - \pi_t) - \left(\frac{1}{\eta} + \frac{1}{c}\right)D_\Phi(\pi(\theta_{t+1}), \pi_t) - \epsilon_t \qquad \text{(Using Prop. 1)}$$

$$\geq J(\pi_t) + \hat{g}(\pi_t)^\top(\pi(\bar{\theta}_{t+1}) - \pi_t) - \left(\frac{1}{\eta} + \frac{1}{c}\right)D_\Phi(\pi(\bar{\theta}_{t+1}), \pi_t) - \epsilon_t \qquad \text{(Using the above bound for GA)}$$

$$= J(\pi_t) + \hat{g}(\pi_t)^\top(\bar{\pi}_{t+1} - \pi_t) - \left(\frac{1}{\eta} + \frac{1}{c}\right)D_\Phi(\bar{\pi}_{t+1}, \pi_t) - e_t - \epsilon_t$$

$$\geq J(\pi_t) + \frac{1}{\eta'_t}\langle\nabla\Phi(\tilde{\pi}_{t+1}) - \nabla\Phi(\pi_t), \bar{\pi}_{t+1} - \pi_t\rangle - \left(\frac{1}{\eta} + \frac{1}{c}\right)D_\Phi(\bar{\pi}_{t+1}, \pi_t) - \epsilon_t^c - e_t \quad \text{(Using the MA update)}$$

$$\geq J(\pi_t) + \frac{1}{\eta'_t}\left\{D_\Phi(\bar{\pi}_{t+1}, \pi_t) + D_\Phi(\pi_t, \tilde{\pi}_{t+1}) - D_\Phi(\bar{\pi}_{t+1}, \tilde{\pi}_{t+1})\right\} - \left(\frac{1}{\eta} + \frac{1}{c}\right)D_\Phi(\bar{\pi}_{t+1}, \pi_t) - \epsilon_t^c - e_t$$

$$\text{(using Lemma 4)}$$

$$= J(\pi_t) + \frac{1}{\eta'_t}\underbrace{\left\{D_\Phi(\pi_t, \tilde{\pi}_{t+1}) - D_\Phi(\bar{\pi}_{t+1}, \tilde{\pi}_{t+1})\right\}}_{:=A} + \left(\frac{1}{\eta'_t} - \frac{1}{\eta} - \frac{1}{c}\right)D_\Phi(\bar{\pi}_{t+1}, \pi_t) - \epsilon_t^c - e_t$$

$$\geq J(\pi_t) + \left(\frac{1}{\eta'_t} - \frac{1}{\eta} - \frac{1}{c}\right)D_\Phi(\bar{\pi}_{t+1}, \pi_t) - \epsilon_t^c - e_t \quad (A \geq 0 \text{ since } \bar{\pi}_{t+1} \text{ is the projection of } \tilde{\pi}_{t+1} \text{ into the simplex})$$

$$\geq J(\pi_t) + \underbrace{\left(\frac{1}{\eta} + \frac{1}{c}\right)}_{:=\frac{1}{\zeta}}D_\Phi(\bar{\pi}_{t+1}, \pi_t) - \epsilon_t^c - e_t \qquad (\text{ setting } \eta'_t \text{ s.t. } 1/\eta'_t \geq 2/\eta + 2/c)$$

Recusing for $T$ iterations and dividing by $1/\zeta$, picking $\mathcal{R}$ uniformly random from $\{0, 1, 2, \ldots T-1\}$ and taking expectation we get

$$\mathbb{E}\frac{D_\Phi(\bar{\pi}_{\mathcal{R}+1}, \pi_\mathcal{R})}{\zeta^2} = \frac{1}{\zeta^2 T}\sum_{t=0}^{T-1}\mathbb{E}D_\Phi(\bar{\pi}_{t+1}, \pi_t)$$

$$\leq \frac{\mathbb{E}[J(\pi_T) - J(\pi_0)]}{\zeta T} + \frac{\sum_{t=0}^{T-1}\mathbb{E}\epsilon_t^c}{\zeta T} + \frac{\sum_{t=0}^{T-1}\mathbb{E}e_t}{\zeta T}$$

$$\leq \frac{[J(\pi^*) - J(\pi_0)]}{\gamma T} + \frac{\sum_{t=0}^{T-1}\mathbb{E}\epsilon_t^c}{\gamma T} + \frac{\sum_{t=0}^{T-1}\mathbb{E}e_t}{\gamma T}$$

$$\implies \mathbb{E}\frac{D_\Phi(\bar{\pi}_{\mathcal{R}+1}, \pi_\mathcal{R})}{\zeta^2} \leq \frac{1}{\zeta T}\left[[J(\pi^*) - J(\pi_0)] + \frac{1}{c}\sum_{t=0}^{T-1}\mathbb{E}D_{\Phi^*}\left(\nabla\Phi(\pi_t) - c[\nabla J(\pi_t) - \hat{g}(\pi_t)], \nabla\Phi(\pi_t)\right) + \sum_{t=0}^{T-1}\mathbb{E}e_t\right]$$

$$\square$$

Note that it is possible to incorporate a sampling error (in the distribution $d^\pi$ across states) for the actor update in Algorithm 1. This corresponds to an additional error in calculating $D_\Phi$, and we can use the techniques from Lavington et al. [31] to characterize the convergence in this case.

## D.2  Exact setting with Lifting (Direct representation)

Recall the mirror ascent update in the functional space.

$$\bar{\pi}_{t+1} = \arg\max_{\pi \in \Pi} \left[ J(\pi_t) + \nabla J(\pi_t)^\top (\pi - \pi_t) - \frac{1}{\eta'_t} D_\Phi(\pi, \pi_t) \right],$$

For the direct representation, we define $\pi^s := p^\pi(\cdot|s)$, $\hat{g}(\pi_t)(s, \cdot) = d^{\pi_t}(s) Q^{\pi_t}(s, \cdot)$ and $D_\Phi(\pi, \pi_t) = \sum_s d^{\pi_t}(s) D_\phi(\pi^s, \pi_t{}^s)$. Rewriting the MA update,

$$\bar{\pi}_{t+1} = \arg\max_{\{\pi^s \in \Delta_A\}_{s \in \mathcal{S}}} \left[ J(\pi_t) + \sum_s d^{\pi_t}(s) \left[ \langle Q^{\pi_t}(s, \cdot), \pi^s - \pi_t{}^s \rangle - \frac{1}{\eta'_t} D_\phi(\pi^s, \pi_t{}^s) \right] \right]$$

$$\implies \bar{\pi}_{t+1}^s = \arg\max_{\pi^s \in \Delta_A} \left[ \langle Q^{\pi_t}(s, \cdot), \pi^s - \pi_t{}^s \rangle - \frac{1}{\eta'_t} D_\phi(\pi^s, \pi_t{}^s) \right] \qquad \text{(Can decompose across states since } d^{\pi_t}(s) \geq 0\text{)}$$

For each state $s$ and $Q^{\pi_t}(s, .)$ we define the set $\Pi_t^s = \{\pi^s : \pi^s \in \arg\max_{p^s \in \Delta_A} \langle Q^{\pi_t}(s, .), p^s \rangle\}$ i.e. a set of greedy policies w.r.t. $Q^{\pi_t}(s, .)$. Similar to Johnson et al. [24] we define $\eta'_t$ as follows

$$\eta'_t \geq \frac{1}{c_t} \max_s \left\{ \min_{\pi \in \Pi_t^s} D_\Phi(\pi, \pi_t) \right\} \tag{5}$$

where $c_t > 0$ is a constant. Now we consider the policy parameterization, $\pi = \pi(\theta)$. We assume that the mapping from $\theta \to \pi$ is $L_\pi$ Lipschitz continuous.

$$\ell_t(\theta) := J(\pi_t) + \sum_s d^{\pi_t}(s) \left[ \langle Q^{\pi_t}(s, \cdot), \pi(\theta)^s - \pi_t{}^s \rangle - \frac{1}{\eta'_t} D_\phi(\pi(\theta)^s, \pi_t{}^s) \right]$$

$$\tilde{\theta}_{t+1} := \arg\max_\theta \ell_t(\theta)$$

$$\theta_{t+1} := \text{GradientAscent}(\ell_t(\theta), \theta_t, m)$$

$$\tilde{\pi}_{t+1} := \pi(\tilde{\theta}_{t+1})$$

$$\pi_{t+1} := \pi(\theta_{t+1})$$

GradientAscent$(\ell_t(\theta), \theta_t, m)$ means that we run gradient ascent for $m$ iterations to maximize $\ell_t$ with $\theta_t$ as the initial value. We assume that $\ell_t$ satisfies Restricted Secant Inequality (RSI) and is smooth w.r.t. $\theta$. Based on the convergence property of Gradient Ascent for RSI and smooth functions [27], we have:

$$\left\| \tilde{\theta}_{t+1} - \theta_{t+1} \right\|_2^2 \leq \mathcal{O}(\exp(-m))$$

$$\implies \|\tilde{\pi}_{t+1} - \pi_{t+1}\|_2^2 = \left\| \pi(\tilde{\theta}_{t+1}) - \pi(\theta_{t+1}) \right\|_2^2 \leq L_\pi^2 \left\| \tilde{\theta}_{t+1} - \theta_{t+1} \right\|_2^2 \leq \underbrace{e_t^2 := \mathcal{O}(\exp(-m))}_{\text{approximation error}}$$

$$\text{( since } \pi(\theta) \text{ is Lipschitz continuous)}$$

Furthermore, we assume that $\|\bar{\pi}_{t+1} - \tilde{\pi}_{t+1}\|_2^2 \leq b_t^2$ for all $t$ which represents the bias because of the function approximation. Before stating the main proposition of this section, we restate Johnson et al. [24, Lemma 2].

**Lemma 3** (Lemma 2 of Johnson et al. [24]). *For all $(s, a) \in \mathcal{S} \times \mathcal{A}$ we have*

$$Q^{\bar{\pi}_{t+1}}(s, a) \geq Q^{\pi_t}(s, a).$$

Now we state the main proposition of this part.

**Proposition 10** (Convergence of tabular MDP with Lifting). Assume that (i) $\ell_t(\theta)$ is smooth and satisfies RSI condition, (ii) $\pi(\theta)$ is $L_\pi$-Lipschitz continuous, (iii) the bias is bounded for all $t$ i.e. $\|\bar{\pi}_{t+1} - \tilde{\pi}_{t+1}\|_2^2 \leq b_t^2$, (iv) $\|Q^\pi(s,\cdot)\| \leq q$ for all $\pi$ and $s$. By setting $\eta_t'$ as in Eq. (5) and running Gradient Ascent for $m$ iterations to maximize $\ell_t$ we have

$$\|J(\pi^*) - J(\pi_T)\|_\infty \leq \gamma^T \left( \|J(\pi^*) - J(\pi_0)\|_\infty + \sum_{t=1}^{T} \gamma^{-t} \left( c_t + \frac{q}{1-\gamma} [e_t + b_t] \right) \right)$$

where $\pi^*$ is the optimal policy, $\pi^{*s}$ refers to the optimal action in state $s$. Here, $e_t = O(\exp(-m))$ is the approximation error.

*Proof.* This proof is mainly based on the proof of Theorem 3 of Johnson et al. [24]. Using Lemma 3 and the fact that $\pi^s \geq 0$, we have $\langle Q^{\pi_t}(s,\cdot), \bar{\pi}_{t+1}^s \rangle \leq \langle Q^{\bar{\pi}_{t+1}}(s,\cdot), \bar{\pi}_{t+1}^s \rangle = J_s(\bar{\pi}_{t+1})$. Using this inequality we get,

$$
\begin{aligned}
\langle Q^{\pi_t}(s,\cdot), \pi^{*s} - \bar{\pi}_{t+1}^s \rangle &\geq \langle Q^{\pi_t}(s,\cdot), \pi^{*s} \rangle - J_s(\bar{\pi}_{t+1}) \\
&= \langle Q^{\pi_t}(s,\cdot) - Q^{\pi^*}(s,\cdot), \pi^{*s} \rangle + \langle Q^{\pi^*}(s,\cdot), \pi^{*s} \rangle - J_s(\bar{\pi}_{t+1}) \\
&\geq -\left\| Q^{\pi_t}(s,\cdot) - Q^{\pi^*}(s,\cdot) \right\|_\infty + J_s(\pi^*) - J_s(\bar{\pi}_{t+1}) \qquad \text{(Holder's inequality)} \\
&\geq -\gamma \|J(\pi_t) - J(\pi^*)\|_\infty + J_s(\pi^*) - J_s(\bar{\pi}_{t+1})
\end{aligned}
$$

The last inequality is from the definition of $Q$ and $J$ as follows. For any action $a$,

$$
\begin{aligned}
Q^{\pi_t}(s,a) - Q^{\pi^*}(s,a) &= \gamma \sum_{s'} P(s'|s,a) \left[ J_{s'}(\pi_t) - J_{s'}(\pi^*) \right] \\
&\leq \gamma \sum_{s'} P(s'|s,a) \|J(\pi_t) - J(\pi^*)\|_\infty \\
&\leq \gamma \|J(\pi_t) - J(\pi^*)\|_\infty
\end{aligned}
$$

From the above inequality,

$$
\begin{aligned}
-\gamma \|J(\pi_t) - J(\pi^*)\|_\infty + J_s(\pi^*) - J_s(\bar{\pi}_{t+1}) &\leq \langle Q^{\pi_t}(s,\cdot), \pi^{*s} - \bar{\pi}_{t+1}^s \rangle \\
&\leq \langle Q^{\pi_t}(s,\cdot), p_t^s - \bar{\pi}_{t+1}^s \rangle \qquad \text{(For any } p_t^s \in \Pi_t^s) \\
&\leq \frac{D_\phi(p_t^s, \pi_t^s) - D_\phi(p_t^s, \bar{\pi}_{t+1}^s) - D_\phi(\bar{\pi}_{t+1}^s, \pi_t^s)}{\eta_t'} \\
&\qquad \text{(Using Lemma 5 with } d = Q^{\pi_t}(s,\cdot),\ y = \bar{\pi}_{t+1}^s,\ x = p_t^s) \\
&\leq \frac{D_\phi(p_t^s, \pi_t^s)}{\eta_t'} \\
\implies -\gamma \|J(\pi_t) - J(\pi^*)\|_\infty + J_s(\pi^*) - J_s(\bar{\pi}_{t+1}) &\leq \min_{p_t^s \in \Pi_t^s} \frac{D_\phi(p_t^s, \pi_t^s)}{\eta_t'} \leq c_t \quad \text{(Based on the definition of } \eta' \text{ in Eq. (5))} \\
\implies -\gamma \|J(\pi_t) - J(\pi^*)\|_\infty + J_{s'}(\pi^*) - J_{s'}(\pi_{t+1}) &\leq c_t + J_{s'}(\bar{\pi}_{t+1}) - J_{s'}(\pi_{t+1}) \\
&\qquad \text{(Since } s \text{ is an arbitrary state, changing } s = s' \text{ for convenience)} \\
&= c_t + \frac{1}{1-\gamma} \sum_s d^{\bar{\pi}_{t+1}}(s) \langle Q^{\pi_{t+1}}(s,\cdot), \bar{\pi}_{t+1}^s - \pi_{t+1}^s \rangle \\
&\qquad \text{(Using performance difference lemma 6 with the starting state equal to } s') \\
&\leq c_t + \frac{1}{1-\gamma} \sum_s d^{\bar{\pi}_{t+1}}(s) \|Q^{\pi_{t+1}}(s,\cdot)\| \left\| \bar{\pi}_{t+1}^s - \pi_{t+1}^s \right\| \\
&\qquad \text{(Cauchy Schwartz)} \\
&\leq c_t + \frac{q}{1-\gamma} \sum_s d^{\bar{\pi}_{t+1}}(s) \left\| \bar{\pi}_{t+1}^s - \pi_{t+1}^s \right\| \\
&\leq c_t + \frac{q}{1-\gamma} \sum_s d^{\bar{\pi}_{t+1}}(s) \left[ \left\| \bar{\pi}_{t+1}^s - \tilde{\pi}_{t+1}^s \right\| + \left\| \tilde{\pi}_{t+1}^s - \pi_{t+1}^s \right\| \right] \\
&\leq c_t + \frac{q}{1-\gamma} (e_t + b_t)
\end{aligned}
$$

Since the above equation is true for all $s'$ we have:

$$\|J(\pi^*) - J(\pi_{t+1})\|_\infty \leq \gamma \|J(\pi_t) - J(\pi^*)\|_\infty + c_t + \frac{q}{1-\gamma}(e_t + b_t)$$

Recursing for $T$ iterations we get:

$$\|J(\pi^*) - J(\pi_T)\|_\infty \leq \gamma^T \left( \|J(\pi^*) - J(\pi_0)\|_\infty + \sum_{t=1}^T \gamma^{-t}(c_t + \frac{q}{1-\gamma}[e_t + b_t]) \right)$$

$\square$

We can control the approximation error $e_t$ by using a larger $m$. The bias term $b_t$ can be small if our function approximation model is expressive enough. $c_t$ is an arbitrary value and if we set $c_t = \gamma^t c$ for some constant $c > 0$, then $\sum_{t=1}^T \gamma^{-t}(c_t) = Tc$ and therefore $\gamma^T Tc$ can diminish linearly. The above analysis relied on the knowledge of the true $Q$ functions, but can be easily extended to using inexact estimates of $Q^\pi$ by using the techniques developed in [61, 24].

### D.3 Exact setting with lifting trick (Softmax representation)

In the softmax representation in the tabular MDP, we consider the case that $\pi$ is parameterized with parameter $\theta \in \mathcal{R}^n$. In this setting $\Phi$ is the Euclidean norm. Using Prop. 1, for $\eta$ such that $J + \frac{1}{\eta}\phi$ is convex we have for a given $\pi_t$,

$$J(\pi) \geq J(\pi_t) + \langle \nabla J(\pi_t), \pi - \pi_t \rangle - \frac{1}{\eta} D_\Phi(\pi, \pi_t)$$

$$= \underbrace{J(\pi_t) + \langle \nabla J(\pi_t), \pi - \pi_t \rangle - \frac{1}{2\eta} \|\pi - \pi_t\|_2^2}_{:=h(\pi)} \qquad \text{(Since } \phi(.) = \tfrac{1}{2}\|.\|_2^2)$$

If we maximize $h(\pi)$ w.r.t. $\pi$ we get

$$\bar{\pi}_{t+1} = \arg\max_\pi \{h(\pi)\} \implies \bar{\pi}_{t+1} = \pi_t + \eta \nabla_\pi J(\pi_t)$$

Mei et al. [37, Lemma 8] proves that $J(\pi)$ satisfies a gradient domination condition w.r.t the softmax representation. In particular, if $a^*(s)$ is the optimal action in state $s$ and $\mu := \min_\pi \frac{\min_s p^\pi(a^*(s)|s)}{\sqrt{S}\left\|\frac{d^{\pi^*}}{d^\pi}\right\|_\infty}$, they prove that for all $\pi$,

$$\|\nabla_\pi J(\pi)\| \geq \mu \left[ J(\pi^*) - J(\pi) \right]$$

Consider optimization in the parameter space where $\ell_t(\theta) := J(\pi_t) + \langle \nabla J(\pi(\theta_t)), \pi(\theta) - \pi(\theta_t) \rangle - \frac{1}{\eta} D_\Phi(\pi(\theta), \pi(\theta_t))$.

$$\tilde{\theta}_{t+1} := \arg\max_\theta \ell_t(\theta)$$

$$\tilde{\pi}_{t+1} = \pi(\tilde{\theta}_{t+1})$$
$$\theta_{t+1} := \text{GradientAscent}(\ell_t, \theta_t, m)$$
$$\pi_{t+1} = \pi(\theta_{t+1})$$

GradientAscent$(\ell_t(\theta), \theta_t, m)$ means that we run gradient ascent for $m$ iterations to maximize $\ell_t$ with $\theta_t$ as the initial value. Assuming that $\ell_t$ is Lipschitz smooth w.r.t. $\theta$ and satisfies the Polyak-Lojasiewicz (PL) condition, we use the gradient ascent property for PL functions [27] to obtain,

$$h(\tilde{\pi}_{t+1}) - h(\pi_{t+1}) = \ell_t(\tilde{\theta}_{t+1}) - \ell(\theta_{t+1}) \leq \underbrace{e_t := O(\exp(-m))}_{\text{approximation error}}$$

**Proposition 11** (Convergence of softmax+tabular setting with Lifting). Assume (i) $J + \frac{1}{\eta}\phi$ is convex, (ii) $J$ satisfies gradient domination property above with $\mu > 0$, (iii) $\ell_t(\theta)$ is Lipschitz smooth and satisfies PL condition, (iv) $|h(\bar{\pi}_{t+1}) - h(\tilde{\pi}_{t+1})| \le b_t$ for all $t$. Then after running Gradient Ascent for $m$ iterations to maximize $\ell_t$ we have

$$\min_{t \in [T-1]} [J(\pi^*) - J(\pi_t)] \le \sqrt{\frac{J(\pi^*) - J(\pi_0) + \sum_{t=0}^{T-1} [e_t + b_t]}{\alpha T}}$$

where $\alpha := \frac{\eta \mu^2}{2}$ and $e_t$ is the approximation error at iteration $t$ and $[T-1] := \{0, 1, 2, \ldots T-1\}$.

*Proof.* Since $J + \frac{1}{\eta}\phi$ is convex,

$$J(\pi_{t+1}) \ge h(\pi_{t+1}) = J(\pi_t) + \langle \nabla J(\pi_t), \pi_{t+1} - \pi_t \rangle - \frac{1}{2\eta} \|\pi_{t+1} - \pi_t\|_2^2$$

$$\ge h(\tilde{\pi}_{t+1}) - e_t \qquad \text{(Using the GA bound from above)}$$

$$\ge h(\bar{\pi}_{t+1}) - e_t - b_t = J(\pi_t) + \langle \nabla J(\pi_t), \bar{\pi}_{t+1} - \pi_t \rangle - \frac{1}{2\eta} \|\bar{\pi}_{t+1} - \pi_t\|_2^2 - e_t - b_t$$

$$\ge J(\pi_t) + \frac{\eta}{2} \|\nabla_\pi J(\pi_t)\|_2^2 - e_t - b_t \qquad \text{(Since } \bar{\pi}_{t+1} = \pi_t + \eta \nabla_\pi J(\pi_t))$$

$$\ge J(\pi_t) + \frac{\eta \mu^2}{2} [J(\pi^*) - J(\pi_t)]^2 - e_t - b_t \qquad \text{(Using gradient domination of } J)$$

$$\implies J(\pi^*) - J(\pi_{t+1}) \le \underbrace{J(\pi^*) - J(\pi_t)}_{:=\delta_t} - \underbrace{\frac{\eta \mu^2}{2}}_{:=\alpha} [J(\pi^*) - J(\pi_t)]^2 + e_t + b_t$$

$$\implies \delta_{t+1} \le \delta_t - \alpha \delta_t^2 + e_t + b_t$$

$$\implies \alpha \delta_t^2 \le \delta_t - \delta_{t+1} + e_t + b_t$$

Summing up for $T$ iterations and dividing both sides by $T$

$$\alpha \min_{t \in [T-1]} \delta_t^2 \le \frac{1}{T} \alpha \sum_{t=0}^{T-1} \delta_t^2$$

$$\le \frac{1}{T} [\delta_0 - \delta_{T+1}] + \frac{1}{T} \sum_{t=0}^{T-1} [e_t + b_t] \le \frac{1}{T} [\delta_0] + \frac{1}{T} \sum_{t=0}^{T-1} [e_t + b_t]$$

$$\implies \min_{t \in [T-1]} \delta_t \le \sqrt{\frac{\delta_0 + \sum_{t=0}^{T-1} [e_t + b_t]}{\alpha T}}$$

$\square$

The above analysis relied on the knowledge of the exact gradient $\nabla J(\pi)$, but can be easily extended to using inexact estimates of the gradient by using the techniques developed in [63].

### D.4 Helper Lemmas

**Lemma 4** (3-Point Bregman Property). *For $x, y, z \in \mathcal{X}$,*

$$\langle \nabla \phi(z) - \nabla \phi(y), z - x \rangle = D_\Phi(x, z) + D_\Phi(z, y) - D_\Phi(x, y)$$

**Lemma 5** (3-Point Descent Lemma for Mirror Ascent). *For any $z \in$ rint dom $\phi$, and a vector $d$, let*

$$y = \arg\max_{x \in \mathcal{X}} \{ \langle d, x \rangle - \frac{1}{\eta} D_\Phi(x, z) \}.$$

*Then $y \in$ rint dom $\phi$ and for any $x \in \mathcal{X}$*

$$\langle d, y - x \rangle \ge \frac{1}{\eta} [D_\Phi(y, z) + D_\Phi(x, y) - D_\Phi(x, z)]$$

**Lemma 6** (Performance Difference Lemma [26]). *For any $\pi, \pi' \in \Pi$,*

$$J(\pi) - J(\pi') = \frac{1}{1-\gamma} \mathbb{E}_{s \sim d^\pi} \left[ \langle Q^{\pi'}(s, \cdot), p^\pi(\cdot|s) - p^{\pi'}(\cdot|s) \rangle \right]$$

# E   Proofs for Sec. 5

**Proposition 12** (State-wise lower bound). For (i) any representation $\pi$ that is separable across states i.e. there exists $\pi^s \in R^A$ such that $\pi_{s,a} = [\pi^s]_a$, (ii) any strictly convex mirror map $\Phi$ that induces a Bregman divergence that is separable across states i.e. $D_\Phi(\pi, \pi') = \sum_s d^\pi(s) D_\phi(\pi^s, \pi'^s)$, (iii) any $\eta$ such that $J + \frac{1}{\eta}\Phi$ is convex, if (iv) $\nabla J(\pi)$ is separable across states i.e. $[\nabla J(\pi)]_{s,a} = d^\pi(s) [\nabla_{\pi^s} J(\pi)]_a$ where $\nabla_{\pi^s} J(\pi) \in \mathbb{R}^A$, then (v) for any separable (across states) gradient estimator $\hat{g}$ i.e. $[\hat{g}(\pi)]_{s,a} = d^\pi(s) [\hat{g}^s(\pi)]_a$ where $\hat{g}^s(\pi) \in \mathbb{R}^A$, and $c \in (0, \infty)^S$,

$$J(\pi) \geq J(\pi_t) + \langle \hat{g}(\pi_t), (\pi - \pi_t) \rangle - \sum_s d^{\pi_t}(s) \left( \frac{1}{\eta} + \frac{1}{c_s} \right) D_\phi(\pi^s, \pi_t^s) - \sum_s \frac{d^{\pi_t}(s) D_{\phi^*}(\nabla\phi(\pi_t^s) - c_s \delta_t^s, \nabla\phi(\pi_t^s))}{c_s}$$

*Proof.* Using condition (iii) of the proposition with Lemma 2,

$$J(\pi) \geq J(\pi_t) + \langle \nabla J(\pi_t), \pi - \pi_t \rangle - \frac{1}{\eta} D_\phi(\pi, \pi_t)$$

$$= J(\pi_t) + \langle \hat{g}(\pi_t), \pi - \pi_t \rangle + \langle \nabla J(\pi_t) - \hat{g}(\pi_t), \pi - \pi_t \rangle - \frac{1}{\eta} D_\phi(\pi, \pi_t)$$

Using conditions (iv) and (v), we know that $[\nabla J(\pi_t)]_{s,a} = d^{\pi_t}(s) [\nabla_{\pi^s} J(\pi_t)]_a$ and $[\hat{g}(\pi_t)]_{s,a} = d^{\pi_t}(s) [\hat{g}^s(\pi_t)]_a$. Defining $\delta_t^s := \nabla_{\pi^s} J(\pi_t) - \hat{g}^s(\pi_t) \in \mathbb{R}^A$. Using conditions (i) and (ii), we can rewrite the lower-bound as follows,

$$J(\pi) \geq J(\pi_t) + \langle \hat{g}(\pi_t), (\pi - \pi_t) \rangle + \sum_s d^{\pi_t}(s) \langle \delta_t^s, \pi^s - \pi_t^s \rangle - \frac{1}{\eta} \sum_s d^{\pi_t}(s) D_\phi(\pi^s, \pi_t^s)$$

$$= J(\pi_t) + \langle \hat{g}(\pi_t), \pi - \pi_t \rangle + \sum_s d^{\pi_t}(s) \left[ \langle \delta_t^s, \pi^s - \pi_t^s \rangle - \frac{1}{\eta} D_\phi(\pi^s, \pi_t^s) \right]$$

Using Lemma 1 with $x = \delta_t^s$, $y = \pi^s$ and $y' = \pi_t^s$,

$$\geq J(\pi_t) + \langle \hat{g}(\pi_t), \pi - \pi_t \rangle - \sum_s d^{\pi_t}(s) \left[ \frac{D_{\phi^*}(\nabla\phi(\pi_t^s) - c_s \delta_t^s, \nabla\phi(\pi_t^s))}{c_s} + \left( \frac{1}{\eta} + \frac{1}{c_s} \right) D_\phi(\pi^s, \pi_t^s) \right]$$

$$J(\pi) \geq J(\pi_t) + \langle \hat{g}(\pi_t), \pi - \pi_t \rangle - \sum_s d^{\pi_t}(s) \left( \frac{1}{\eta} + \frac{1}{c_s} \right) D_\phi(\pi^s, \pi_t^s) - \sum_s \frac{d^{\pi_t}(s) D_{\phi^*}(\nabla\phi(\pi_t^s) - c_s \delta_t^s, \nabla\phi(\pi_t^s))}{c_s}$$

$\square$

**Proposition 4.** For the direct representation and negative entropy mirror map, $c > 0$, $\eta \leq \frac{(1-\gamma)^3}{2\gamma |A|}$,

$$J(\pi) - J(\pi_t) \geq C + \mathbb{E}_{s \sim d^{\pi_t}} \left[ \mathbb{E}_{a \sim p^{\pi_t}(\cdot|s)} \left[ \frac{p^\pi(a|s)}{p^{\pi_t}(a|s)} \left( \hat{Q}^{\pi_t}(s, a) - \left( \frac{1}{\eta} + \frac{1}{c} \right) \log \left( \frac{p^\pi(a|s)}{p^{\pi_t}(a|s)} \right) \right) \right] \right]$$

$$- \mathbb{E}_{s \sim d^{\pi_t}} \left[ \mathbb{E}_{a \sim p^{\pi_t}(\cdot|s)} [Q^{\pi_t}(s, a) - \hat{Q}^{\pi_t}(s, a)] + \frac{1}{c} \log \left( \mathbb{E}_{a \sim p^{\pi_t}(\cdot|s)} \left[ \exp \left( -c [Q^{\pi_t}(s, a) - \hat{Q}^{\pi_t}(s, a)] \right) \right] \right) \right]$$

where $C$ is a constant and $\hat{Q}^{\pi_t}$ is the estimate of the action-value function for policy $\pi_t$.

*Proof.* For the direct representation, $\pi_{s,a} = p^\pi(a|s)$. Using the policy gradient theorem, $[\nabla_\pi J(\pi)]_{s,a} = d^\pi(s) Q^\pi(s, a)$. We choose $\hat{g}(\pi)$ such that $[\hat{g}(\pi)]_{s,a} = d^\pi(s) \hat{Q}^\pi(s, a)$ as the estimated gradient. Using Vaswani et al. [57, Proposition 2], $J + \frac{1}{\eta}\Phi$ is convex for $\eta \leq \frac{(1-\gamma)^3}{2\gamma |A|}$. Defining $\delta_t^s := \nabla_{\pi^s} J(\pi_t) - \hat{g}^s(\pi_t) = Q^{\pi_t}(s, \cdot) - \hat{Q}^{\pi_t}(s, \cdot) \in \mathbb{R}^A$, and using Prop. 12 with $c_s = c$ for all $s$,

$$J(\pi) \geq J(\pi_t) + \langle \hat{g}(\pi_t), \pi - \pi_t \rangle - \sum_s d^{\pi_t}(s) \left( \frac{1}{\eta} + \frac{1}{c} \right) D_\phi(\pi^s, \pi_t^s) - \sum_s \frac{d^{\pi_t}(s) D_{\phi^*}(\nabla\phi(\pi_t^s) - c \delta_t^s, \nabla\phi(\pi_t^s))}{c}$$

Since $\phi(\pi^s) = \phi(p^\pi(\cdot|s)) = \sum_a p^\pi(a|s) \log(p^\pi(a|s))$, using Lemma 10, $D_\phi(\pi^s, \pi_t{}^s) = \mathrm{KL}(p^\pi(\cdot|s)||p^{\pi_t}(\cdot|s))$. Hence,

$$J(\pi) \geq J(\pi_t) + \sum_s d^{\pi_t}(s) \sum_a \hat{Q}^{\pi_t}(s,a) \left[p^\pi(a|s) - p^{\pi_t}(a|s)\right] - \left(\frac{1}{\eta} + \frac{1}{c}\right) \sum_s d^{\pi_t}(s) \, \mathrm{KL}(p^\pi(\cdot|s)||p^{\pi_t}(\cdot|s))$$
$$- \sum_s \frac{d^{\pi_t}(s) \, D_{\phi^*}\left(\nabla\phi(\pi_t{}^s) - c\,\delta_t^s, \nabla\phi(\pi_t{}^s)\right)}{c}$$

Using Lemma 7 to simplify the last term,

$$\sum_s \frac{d^{\pi_t}(s) \, D_{\phi^*}\left(\nabla\phi(\pi_t{}^s) - c\,\delta_t^s, \nabla\phi(\pi_t{}^s)\right)}{c}$$
$$= \frac{1}{c}\left[\sum_s d^{\pi_t}(s) \left[c \langle p^{\pi_t}(\cdot|s), \delta_t^s\rangle + \log\left(\sum_a p^{\pi_t}(a|s) \, \exp(-c\,\delta_t^s[a])\right)\right]\right]$$
$$= \sum_s d^{\pi_t}(s) \left[\sum_a p^{\pi_t}(a|s)[Q^{\pi_t}(s,a) - \hat{Q}^{\pi_t}(s,a)] + \frac{1}{c}\log\left(\sum_a p^{\pi_t}(a|s) \, \exp\left(-c\,[Q^{\pi_t}(s,a) - \hat{Q}^{\pi_t}(s,a)]\right)\right)\right]$$

Putting everything together,

$$J(\pi) \geq J(\pi_t) + \sum_s d^{\pi_t}(s) \sum_a \hat{Q}^{\pi_t}(s,a)\left[p^\pi(a|s) - p^{\pi_t}(a|s)\right] - \left(\frac{1}{\eta} + \frac{1}{c}\right) \sum_s d^{\pi_t}(s) \, \mathrm{KL}(p^\pi(\cdot|s)||p^{\pi_t}(\cdot|s))$$
$$- \left[\sum_s d^{\pi_t}(s) \left[\sum_a p^{\pi_t}(a|s)[Q^{\pi_t}(s,a) - \hat{Q}^{\pi_t}(s,a)] + \frac{1}{c}\log\left(\sum_a p^{\pi_t}(a|s) \, \exp\left(-c[Q^{\pi_t}(s,a) - \hat{Q}^{\pi_t}(s,a)]\right)\right)\right]\right]$$
$$= J(\pi_t) - \underbrace{\mathbb{E}_{s\sim d^{\pi_t}}\left[\mathbb{E}_{a\sim p^{\pi_t}(\cdot|s)}[\hat{Q}^{\pi_t}(s,a)]\right]}_{:=-C} + \mathbb{E}_{s\sim d^{\pi_t}}\left[\mathbb{E}_{a\sim p^\pi(\cdot|s)}\left[\hat{Q}^{\pi_t}(s,a) - \left(\frac{1}{\eta} + \frac{1}{c}\right)\log\left(\frac{p^\pi(a|s)}{p^{\pi_t}(a|s)}\right)\right]\right]$$
$$- \left[\sum_s d^{\pi_t}(s) \left[\sum_a p^{\pi_t}(a|s)[Q^{\pi_t}(s,a) - \hat{Q}^{\pi_t}(s,a)] + \frac{1}{c}\log\left(\sum_a p^{\pi_t}(a|s) \, \exp\left(-c[Q^{\pi_t}(s,a) - \hat{Q}^{\pi_t}(s,a)]\right)\right)\right]\right]$$
$$J(\pi) \geq J(\pi_t) + C + \mathbb{E}_{s\sim d^{\pi_t}}\left[\mathbb{E}_{a\sim p^{\pi_t}(\cdot|s)}\left[\frac{p^\pi(a|s)}{p^{\pi_t}(a|s)}\left(\hat{Q}^{\pi_t}(s,a) - \left(\frac{1}{\eta} + \frac{1}{c}\right)\log\left(\frac{p^\pi(a|s)}{p^{\pi_t}(a|s)}\right)\right)\right]\right]$$
$$- \mathbb{E}_{s\sim d^{\pi_t}}\left[\mathbb{E}_{a\sim p^{\pi_t}(\cdot|s)}[Q^{\pi_t}(s,a) - \hat{Q}^{\pi_t}(s,a)] + \frac{1}{c}\log\left(\mathbb{E}_{a\sim p^{\pi_t}(\cdot|s)}\left[\exp\left(-c\,[Q^{\pi_t}(s,a) - \hat{Q}^{\pi_t}(s,a)]\right)\right]\right)\right]$$

$\square$

---

**Proposition 6.** For the softmax representation and log-sum-exp mirror map, $c > 0$, $\eta \leq 1 - \gamma$,

$$J(\pi) - J(\pi_t) \geq \mathbb{E}_{s\sim d^{\pi_t}} \mathbb{E}_{a\sim p^{\pi_t}(\cdot|s)}\left[\left(\hat{A}^{\pi_t}(s,a) + \frac{1}{\eta} + \frac{1}{c}\right)\log\left(\frac{p^\pi(a|s)}{p^{\pi_t}(a|s)}\right)\right]$$
$$- \frac{1}{c}\mathbb{E}_{s\sim d^{\pi_t}} \mathbb{E}_{a\sim p^{\pi_t}(\cdot|s)}\left[\left(1 - c\,[A^{\pi_t}(s,a) - \hat{A}^{\pi_t}(s,a)]\right)\log\left(1 - c\,[A^{\pi_t}(s,a) - \hat{A}^{\pi_t}(s,a)]\right)\right]$$

where $\hat{A}^{\pi_t}$ is the estimate of the advantage function for policy $\pi_t$.

---

*Proof.* For the softmax representation, $\pi_{s,a} = z(s,a)$ s.t. $p^\pi(a|s) = \frac{\exp(z(s,a))}{\sum_{a'} \exp(z(s,a'))}$. Using the policy gradient theorem, $[\nabla_\pi J(\pi)]_{s,a} = d^\pi(s) \, p^\pi(a|s) \, A^\pi(s,a)$. We choose $\hat{g}(\pi)$ such that $[\hat{g}(\pi)]_{s,a} = d^\pi(s) \, p^\pi(a|s) \, \hat{A}^\pi(s,a)$ as the estimated gradient. Using Vaswani et al. [57, Proposition 3], $J + \frac{1}{\eta}\Phi$ is convex for $\eta \leq 1 - \gamma$. Define $\delta_s \in \mathbb{R}^A$ such that $\delta_t^s[a] := \nabla_{\pi^s} J(\pi_t) - \hat{g}^s(\pi_t) = p^{\pi_t}(a|s)\,[A^{\pi_t}(s,a) - \hat{A}^{\pi_t}(s,a)]$. Using Prop. 12 with $c_s = c$ for all $s$,

$$J(\pi) \geq J(\pi_t) + \langle\hat{g}(\pi_t), \pi - \pi_t\rangle - \sum_s d^{\pi_t}(s)\left(\frac{1}{\eta} + \frac{1}{c}\right) D_\phi(\pi^s, \pi_t{}^s) - \sum_s \frac{d^{\pi_t}(s) \, D_{\phi^*}\left(\nabla\phi(\pi_t{}^s) - c\,\delta_t^s, \nabla\phi(\pi_t{}^s)\right)}{c}$$

Since $\phi(\pi^s) = \phi(z(s, \cdot)) = \log\left(\sum_a \exp(z(s, a))\right)$, using Lemma 11, $D_\phi(\pi^s, \pi_t^s) = \text{KL}(p^{\pi_t}(\cdot|s)||p^\pi(\cdot|s))$ where $p^\pi(a|s) = \frac{\exp(z(s,a))}{\sum_{a'} \exp(z(s,a'))}$ and $p^{\pi_t}(a|s) = \frac{\exp(z_t(s,a))}{\sum_{a'} \exp(z_t(s,a'))}$. Hence, the above bound can be simplified as,

$$J(\pi) \geq J(\pi_t) + \sum_s d^{\pi_t}(s) \sum_a \hat{A}^{\pi_t}(s, a) \, p^{\pi_t}(a|s) \, [z(s, a) - z_t(s, a)] - \left(\frac{1}{\eta} + \frac{1}{c}\right) \sum_s d^{\pi_t}(s) \, \text{KL}(p^{\pi_t}(\cdot|s)||p^\pi(\cdot|s))$$
$$- \sum_s \frac{d^{\pi_t}(s) \, D_{\phi^*}\left(\nabla\phi(\pi_t^s) - c\,\delta_t^s, \nabla\phi(\pi_t^s)\right)}{c}$$

Using Lemma 8 to simplify the last term,

$$\sum_s \frac{d^{\pi_t}(s) \, D_{\phi^*}\left(\nabla\phi(\pi_t^s) - c\,\delta_t^s, \nabla\phi(\pi_t^s)\right)}{c}$$

$$= \frac{1}{c}\left[\sum_s d^{\pi_t}(s) \left[\sum_a (p^{\pi_t}(a|s) - c\,\delta_t^s[a]) \, \log\left(\frac{p^{\pi_t}(a|s) - c\,\delta_t^s[a]}{p^{\pi_t}(a|s)}\right)\right]\right]$$

$$= \frac{1}{c}\left[\sum_s d^{\pi_t}(s) \left[\sum_a \left(p^{\pi_t}(a|s) - c\left[p^{\pi_t}(a|s)\,[A^{\pi_t}(s, a) - \hat{A}^{\pi_t}(s, a)]\right]\right) \log\left(\frac{p^{\pi_t}(a|s) - c\left[p^{\pi_t}(a|s)\,[A^{\pi_t}(s, a) - \hat{A}^{\pi_t}(s, a)]\right]}{p^{\pi_t}(a|s)}\right)\right]\right]$$

$$= \frac{1}{c}\left[\sum_s d^{\pi_t}(s) \left[\sum_a p^{\pi_t}(a|s) \left[\left(1 - c\,[A^{\pi_t}(s, a) - \hat{A}^{\pi_t}(s, a)]\right) \log\left(1 - c\,[A^{\pi_t}(s, a) - \hat{A}^{\pi_t}(s, a)]\right)\right]\right]\right]$$

Putting everything together,

$$J(\pi) \geq J(\pi_t) + \sum_s d^{\pi_t}(s) \sum_a \hat{A}^{\pi_t}(s, a) \, p^{\pi_t}(a|s) \, [z(s, a) - z_t(s, a)] - \left(\frac{1}{\eta} + \frac{1}{c}\right) \sum_s d^{\pi_t}(s) \, \text{KL}(p^{\pi_t}(\cdot|s)||p^\pi(\cdot|s))$$

$$- \frac{1}{c}\left[\sum_s d^{\pi_t}(s) \left[\sum_a p^{\pi_t}(a|s) \left[\left(1 - c\,[A^{\pi_t}(s, a) - \hat{A}^{\pi_t}(s, a)]\right) \log\left(1 - c\,[A^{\pi_t}(s, a) - \hat{A}^{\pi_t}(s, a)]\right)\right]\right]\right]$$

$$= J(\pi_t) + \sum_s d^{\pi_t}(s) \sum_a \left[p^{\pi_t}(a|s) \, \hat{A}^{\pi_t}(s, a) \, [z(s, a) - z_t(s, a)] - \left(\frac{1}{\eta} + \frac{1}{c}\right) p^{\pi_t}(a|s) \, \log\left(\frac{p^{\pi_t}(a|s)}{p^\pi(a|s)}\right)\right]$$

$$- \frac{1}{c}\mathbb{E}_{s\sim d^{\pi_t}} \mathbb{E}_{a\sim p^{\pi_t}(\cdot|s)} \left[\left(1 - c\,[A^{\pi_t}(s, a) - \hat{A}^{\pi_t}(s, a)]\right) \log\left(1 - c\,[A^{\pi_t}(s, a) - \hat{A}^{\pi_t}(s, a)]\right)\right]$$

Let us focus on simplifying $\sum_a \left[p^{\pi_t}(a|s) \, \hat{A}^{\pi_t}(s, a) \, [z(s, a) - z_t(s, a)]\right]$ for a fixed $s$. Note that $\sum_a p^{\pi_t}(a|s) \, \hat{A}^{\pi_t}(s, a) = 0 \implies \log\left(\sum_{a'} \exp(z(s, a'))\right) \sum_a p^{\pi_t}(a|s) \, \hat{A}^{\pi_t}(s, a) = 0$.

$$\sum_a \left[p^{\pi_t}(a|s) \, \hat{A}^{\pi_t}(s, a) \, z(s, a)\right] = \sum_a \left[p^{\pi_t}(a|s) \, \hat{A}^{\pi_t}(s, a) \left(z(s, a) - \log\left(\sum_{a'} \exp(z(s, a'))\right)\right)\right]$$

$$= \sum_a \left[p^{\pi_t}(a|s) \, \hat{A}^{\pi_t}(s, a) \left(\log(\exp(z(s, a))) - \log\left(\sum_{a'} \exp(z(s, a'))\right)\right)\right]$$

$$= \sum_a \left[p^{\pi_t}(a|s) \, \hat{A}^{\pi_t}(s, a) \, \log\left(\frac{\exp(z(s, a))}{\sum_{a'} \exp(z(s, a'))}\right)\right] = \mathbb{E}_{a\sim p^{\pi_t}(\cdot|s)} \left[\hat{A}^{\pi_t}(s, a) \, \log(p^\pi(a|s))\right]$$

Similarly, simplifying $\sum_a \left[p^{\pi_t}(a|s) \, \hat{A}^{\pi_t}(s, a) \, z_t(s, a)\right]$

$$\sum_a \left[p^{\pi_t}(a|s) \, \hat{A}^{\pi_t}(s, a) \, z_t(s, a)\right] = E_{a\sim p^{\pi_t}(\cdot|s)} \left[\hat{A}^{\pi_t}(s, a) \, \log(p^{\pi_t}(a|s))\right]$$

$$\implies \sum_a \left[p^{\pi_t}(a|s) \, \hat{A}^{\pi_t}(s, a) \, [z(s, a) - z_t(s, a)]\right] = \mathbb{E}_{a\sim p^{\pi_t}(\cdot|s)} \left[\hat{A}^{\pi_t}(s, a) \, \log\left(\frac{p^\pi(a|s)}{p^{\pi_t}(a|s)}\right)\right]$$

Using the above relations,

$$J(\pi) \geq J(\pi_t) + \sum_s d^{\pi_t}(s) \, \mathbb{E}_{a \sim p^{\pi_t}(\cdot|s)} \left[ \hat{A}^{\pi_t}(s,a) \log\left(\frac{p^\pi(a|s)}{p^{\pi_t}(a|s)}\right) - \left(\frac{1}{\eta} + \frac{1}{c}\right) \log\left(\frac{p^{\pi_t}(a|s)}{p^\pi(a|s)}\right) \right]$$

$$- \frac{1}{c} \mathbb{E}_{s \sim d^{\pi_t}} \mathbb{E}_{a \sim p^{\pi_t}(\cdot|s)} \left[ \left(1 - c\left[A^{\pi_t}(s,a) - \hat{A}^{\pi_t}(s,a)\right]\right) \log\left(1 - c\left[A^{\pi_t}(s,a) - \hat{A}^{\pi_t}(s,a)\right]\right) \right]$$

$$= J(\pi_t) + \mathbb{E}_{s \sim d^{\pi_t}} \mathbb{E}_{a \sim p^{\pi_t}(\cdot|s)} \left[ \left(\hat{A}^{\pi_t}(s,a) + \frac{1}{\eta} + \frac{1}{c}\right) \log\left(\frac{p^\pi(a|s)}{p^{\pi_t}(a|s)}\right) \right]$$

$$- \frac{1}{c} \mathbb{E}_{s \sim d^{\pi_t}} \mathbb{E}_{a \sim p^{\pi_t}(\cdot|s)} \left[ \left(1 - c\left[A^{\pi_t}(s,a) - \hat{A}^{\pi_t}(s,a)\right]\right) \log\left(1 - c\left[A^{\pi_t}(s,a) - \hat{A}^{\pi_t}(s,a)\right]\right) \right].$$

$\square$

**Proposition 8.** For the stochastic value gradient representation and Euclidean mirror map, $c > 0$, $\eta$ such that $J + \frac{1}{\eta}\Phi$ is convex in $\pi$.

$$J(\pi) - J(\pi_t) \geq C + \mathbb{E}_{s \sim d^{\pi_t}} \mathbb{E}_{\varepsilon \sim \chi} \left[ \widehat{\nabla_a Q^{\pi_t}}(s,a)\Big|_{a = \pi_t(s,\varepsilon)} \pi(s,\varepsilon) - \frac{1}{2}\left(\frac{1}{\eta} + \frac{1}{c}\right) [\pi_t(s,\epsilon) - \pi(s,\epsilon)]^2 \right]$$

$$- \frac{c}{2} \mathbb{E}_{s \sim d^{\pi_t}} \mathbb{E}_{\varepsilon \sim \chi} \left[ \nabla_a Q^{\pi_t}(s,a)\Big|_{a = \pi_t(s,\varepsilon)} - \widehat{\nabla_a Q^{\pi_t}}(s,a)\Big|_{a = \pi_t(s,\varepsilon)} \right]^2$$

where $C$ is a constant and $\widehat{\nabla_a Q^{\pi_t}}(s,a)\Big|_{a = \pi_t(s,\varepsilon)}$ is the estimate of the action-value gradients for policy $\pi$ at state $s$ and $a = \pi_t(s,\epsilon)$.

*Proof.* For stochastic value gradients with a fixed $\varepsilon$, $\dfrac{\partial J(\pi)}{\partial \pi(s,\epsilon)} = d^\pi(s)\nabla_a Q^\pi(s,a)\Big|_{a = \pi(s,\epsilon)}$. We choose $\hat{g}(\pi)$ such that $[\hat{g}(\pi)]_{s,a} = d^\pi(s)\widehat{\nabla_a Q^\pi}(s,a)\Big|_{a = \pi(s,\epsilon)}$. Define $\delta_t^s \in \mathbb{R}^A$ such that $\delta_t^s[a] := \nabla_a Q^{\pi_t}(s,a)\Big|_{a = \pi_t(s,\epsilon)} - \widehat{\nabla_a Q^{\pi_t}}(s,a)\Big|_{a = \pi_t(s,\epsilon)}$.

Using Prop. 12 with $c_s = c$ for all $s$,

$$J(\pi) \geq J(\pi_t) + \mathbb{E}_{\varepsilon \sim \chi}\left[ \sum_s d^{\pi_t}(s) \widehat{\nabla_a Q^{\pi_t}}(s,a)\Big|_{a = \pi_t(s,\epsilon)} [\pi(s,\varepsilon) - \pi_t(s,\varepsilon)] - \sum_s d^{\pi_t}(s)\left(\frac{1}{\eta} + \frac{1}{c}\right) D_\phi(\pi^s, \pi_t{}^s) \right.$$

$$\left. - \sum_s \frac{d^{\pi_t}(s) D_{\phi^*}\left(\nabla\phi(\pi_t{}^s) - c\,\delta_t^s, \nabla\phi(\pi_t{}^s)\right)}{c} \right]$$

For a fixed $\varepsilon$, since $\phi(\pi^s) = \phi(\pi(s,\epsilon)) = \frac{1}{2}[\pi(s,\epsilon)]^2$, $D_\phi(\pi^s, \pi_t{}^s) = \frac{1}{2}[\pi(s,\epsilon) - \pi_t(s,\epsilon)]^2$. Hence,

$$J(\pi) \geq J(\pi_t) + \mathbb{E}_{\varepsilon \sim \chi}\left[ \sum_s d^{\pi_t}(s) \widehat{\nabla_a Q^{\pi_t}}(s,a)\Big|_{a = \pi_t(s,\epsilon)} [\pi(s,\varepsilon) - \pi_t(s,\varepsilon)] - \frac{1}{2}\left(\frac{1}{\eta} + \frac{1}{c}\right) \sum_s d^{\pi_t}(s)[\pi(s,\epsilon) - \pi_t(s,\epsilon)]^2 \right.$$

$$\left. - \sum_s \frac{d^{\pi_t}(s) D_{\phi^*}\left(\nabla\phi(\pi_t{}^s) - c\,\delta_t^s, \nabla\phi(\pi_t{}^s)\right)}{c} \right]$$

Simplifying the last term, since $\phi(\pi(s,\epsilon)) = \frac{1}{2}[\pi(s,\epsilon)]^2$,

$$\frac{D_{\phi^*}\left(\nabla\phi(\pi_t{}^s) - c\,\delta_t^s, \nabla\phi(\pi_t{}^s)\right)}{c} = \frac{c}{2}[\delta_t^s]^2 = \frac{c}{2}\left[\nabla_a Q^{\pi_t}(s,a)\Big|_{a = \pi_t(s,\epsilon)} - \widehat{\nabla_a Q^{\pi_t}}(s,a)\Big|_{a = \pi_t(s,\epsilon)}\right]^2$$

Putting everything together,

$$J(\pi) \geq J(\pi_t) + \mathbb{E}_{\varepsilon \sim \chi}\left[\sum_s d^{\pi_t}(s)\, \widehat{\nabla_a Q^{\pi_t}}(s,a)\big|_{a=\pi_t(s,\epsilon)}\, \pi(s,\varepsilon) - \underbrace{\sum_s d^{\pi_t}(s)\, \nabla_a Q^{\pi_t}(s,a)\big|_{a=\pi_t(s,\epsilon)}\, \pi_t(s,\varepsilon)]}_{:=-C}\right.$$

$$\left. - \mathbb{E}_{\varepsilon \sim \chi}\left[\frac{1}{2}\left(\frac{1}{\eta}+\frac{1}{c}\right)\sum_s d^{\pi_t}(s)\,[\pi(s,\epsilon)-\pi_t(s,\epsilon)]^2 - \frac{c}{2}\sum_s d^{\pi_t}(s)\left[\nabla_a Q^{\pi_t}(s,a)\big|_{a=\pi_t(s,\epsilon)} - \widehat{\nabla_a Q^{\pi_t}}(s,a)\big|_{a=\pi_t(s,\epsilon)}\right]^2\right]\right]$$

$$J(\pi) \geq J(\pi_t) + C + \mathbb{E}_{\varepsilon \sim \chi}\left[\mathbb{E}_{s \sim d^{\pi_t}}\left[\widehat{\nabla_a Q^{\pi_t}}(s,a)\big|_{a=\pi_t(s,\epsilon)}\, \pi(s,\varepsilon) - \frac{1}{2}\left(\frac{1}{\eta}+\frac{1}{c}\right)[\pi(s,\epsilon)-\pi_t(s,\epsilon)]^2\right]\right]$$

$$- \frac{c}{2}\mathbb{E}_{\varepsilon \sim \chi}\left[\mathbb{E}_{s \sim d^{\pi_t}}\left[\nabla_a Q^{\pi_t}(s,a)\big|_{a=\pi_t(s,\epsilon)} - \widehat{\nabla_a Q^{\pi_t}}(s,a)\big|_{a=\pi_t(s,\epsilon)}\right]^2\right]$$

$\square$

**Proposition 13.** For the softmax representation and Euclidean mirror map, $c>0$, $\eta \leq \frac{(1-\gamma)^3}{8}$ then

$$J(\pi) \geq J(\pi_t) + C + \mathbb{E}_{s \sim d^{\pi_t}(s)}\left[\mathbb{E}_{a \sim p^{\pi_t}(.|s)}\left[\hat{A}^{\pi_t}(s,a)\, z(s,a)\right] - \frac{1}{2}\left(\frac{1}{\eta}+\frac{1}{c}\right)||z(s,\cdot)-z_t(s,\cdot)||^2\right]$$

$$- \frac{c}{2}\mathbb{E}_{s \sim d^{\pi_t}}\mathbb{E}_{a \sim p^{\pi_t}(.|s)}\left[A^{\pi_t}(s,a) - \hat{A}^{\pi_t}(s,a)\right]^2$$

where $C$ is a constant and $\hat{A}^\pi$ is the estimate of advantage function for policy $\pi_t$.

*Proof.* For the softmax representation, $\pi_{s,a} = z(s,a)$ s.t. $p^\pi(a|s) = \frac{\exp(z(s,a))}{\sum_{a'}\exp(z(s,a'))}$. Using the policy gradient theorem, $[\nabla_\pi J(\pi)]_{s,a} = d^\pi(s)\, p^\pi(a|s)\, A^\pi(s,a)$. We choose $\hat{g}(\pi)$ such that $[\hat{g}(\pi)]_{s,a} = d^\pi(s)\, p^\pi(a|s)\, \hat{A}^\pi(s,a)$ as the estimated gradient. Define $\delta_s \in \mathbb{R}^A$ such that $\delta_t^s[a] := \nabla_{\pi^s} J(\pi_t) - \hat{g}^s(\pi_t) = p^{\pi_t}(a|s)\,[A^{\pi_t}(s,a) - \hat{A}^{\pi_t}(s,a)]$. Using Mei et al. [37, Lemma 7], $J + \frac{1}{\eta}\Phi$ is convex for $\eta \leq 1-\gamma$. Using Prop. 12 with $c_s = c$ for all $s$,

$$J(\pi) \geq J(\pi_t) + \langle\hat{g}(\pi_t), \pi - \pi_t\rangle - \left(\frac{1}{\eta}+\frac{1}{c}\right)\sum_s d^{\pi_t}(s)\, D_\phi(\pi^s, \pi_t{}^s) - \sum_s \frac{d^{\pi_t}(s)\, D_{\phi^*}\left(\nabla\phi(\pi_t{}^s) - c\,\delta_t^s, \nabla\phi(\pi_t{}^s)\right)}{c}$$

Since $\phi(\pi^s) = \phi(z(s,\cdot)) = \frac{1}{2}\sum_a[z_{s,a}]^2$, $D_\phi(\pi^s, \pi_t{}^s) = \frac{1}{2}\|z(s,\cdot)-z_t(s,\cdot)\|_2^2$. Hence,

$$J(\pi) \geq J(\pi_t) + \sum_s d^{\pi_t}(s)\sum_a \hat{A}^{\pi_t}(s,a)\, p^{\pi_t}(a|s)\,[z(s,a)-z_t(s,a)] - \frac{1}{2}\left(\frac{1}{\eta}+\frac{1}{c}\right)\sum_s d^{\pi_t}(s)\,\|z(s,\cdot)-z_t(s,\cdot)\|_2^2$$

$$- \sum_s \frac{d^{\pi_t}(s)\, D_{\phi^*}\left(\nabla\phi(\pi_t{}^s) - c\,\delta_t^s, \nabla\phi(\pi_t{}^s)\right)}{c}$$

Simplifying the last term, since $\phi(z(\cdot,a)) = \frac{1}{2}[z(s,a)]^2$,

$$\frac{\sum_s d^{\pi_t}(s)\, D_{\phi^*}\left(\nabla\phi(\pi_t{}^s) - c\,\delta_t^s, \nabla\phi(\pi_t{}^s)\right)}{c} = \frac{c}{2}\sum_s d^{\pi_t}(s)\sum_a[\delta_t^s(a)]^2$$

$$= \frac{c}{2}\sum_s d^{\pi_t}(s)\sum_a p^{\pi_t}(a|s)^2\,[A^{\pi_t}(s,a) - \hat{A}^{\pi_t}(s,a)]^2$$

$$\leq \frac{c}{2}\sum_s d^{\pi_t}(s)\sum_a p^{\pi_t}(a|s)\left[A^{\pi_t}(s,a) - \hat{A}^{\pi_t}(s,a)\right]^2 \qquad \text{(Since } p^{\pi_t}(a|s) \leq 1\text{)}$$

Putting everything together,

$$J(\pi) \geq J(\pi_t) + \sum_s d^{\pi_t}(s) \sum_a \hat{A}^{\pi_t}(s,a) \, p^{\pi_t}(a|s) \, [z(s,a) - z_t(s,a)] - \frac{1}{2}\left(\frac{1}{\eta} + \frac{1}{c}\right) \sum_s d^{\pi_t}(s) \, \|z(s,\cdot) - z_t(s,\cdot)\|_2^2$$

$$- \frac{c}{2} \sum_s d^{\pi_t}(s) \sum_a p^{\pi_t}(a|s) \left[A^{\pi_t}(s,a) - \hat{A}^{\pi_t}(s,a)\right]^2$$

$$= J(\pi_t) - \underbrace{\sum_s d^{\pi_t}(s) \sum_a \hat{A}^{\pi_t}(s,a) \, p^{\pi_t}(a|s) \, z_t(s,a)}_{:=-C}$$

$$+ \sum_s d^{\pi_t}(s) \left[\sum_a \hat{A}^{\pi_t}(s,a) \, p^{\pi_t}(a|s) \, z(s,a) - \frac{1}{2}\left(\frac{1}{\eta} + \frac{1}{c}\right) \|z(s,\cdot) - z_t(s,\cdot)\|_2^2\right]$$

$$- \frac{c}{2} \sum_s d^{\pi_t}(s) \sum_a p^{\pi_t}(a|s) \left[A^{\pi_t}(s,a) - \hat{A}^{\pi_t}(s,a)\right]^2$$

$$= J(\pi_t) + C + \mathbb{E}_{s \sim d^{\pi_t}} \left[\mathbb{E}_{a \sim p^{\pi_t}(\cdot|s)} \left[\hat{A}^{\pi_t}(s,a) \, z(s,a)\right] - \frac{1}{2}\left(\frac{1}{\eta} + \frac{1}{c}\right) \|z(s,\cdot) - z_t(s,\cdot)\|_2^2\right]$$

$$- \frac{c}{2} \mathbb{E}_{s \sim d^{\pi_t}} \mathbb{E}_{a \sim p^{\pi_t}(\cdot|s)} \left[A^{\pi_t}(s,a) - \hat{A}^{\pi_t}(s,a)\right]^2$$

$\square$

**Proposition 14.** For both the direct (with the negative-entropy mirror map) and softmax representations (with the log-sum-exp mirror map), for a fixed state $s$, if $\delta \in \mathbb{R}^A := \nabla_{\pi^s} J(\pi_t) - \hat{g}^s(\pi_t)$, the second-order Taylor expansion of $f(c) = D_{\phi^*}(\nabla\phi(\pi_t^s) - c\delta, \nabla\phi(\pi_t^s))$ around $c = 0$ is equal to

$$f(c) \approx \frac{c^2}{2} \sum_a p^{\pi_t}(a|s)[A(s,a) - \hat{A}(s,a)]^2 \, .$$

*Proof.*

$$f(c) = D_{\phi^*}(\nabla\phi(\pi_t^s) - c\delta, \nabla\phi(\pi_t^s)) \implies f(0) = D_\phi^*(\nabla\phi(\pi_t^s), \nabla\phi(\pi_t^s)) = 0$$

$$f(c) = D_{\phi^*}(\nabla\phi(\pi_t^s) - c\delta, \nabla\phi(\pi_t^s)) = \phi^*(\nabla\phi(\pi_t^s) - c\delta) - \phi^*(\nabla\phi(\pi_t^s)) - \langle\nabla\phi^*(\nabla\phi(\pi_t^s)), \nabla\phi(\pi_t^s) - c\delta - \nabla\phi(\pi_t^s)\rangle$$

$$\implies f'(c) = \langle\nabla\phi^*(\nabla\phi(\pi_t^s) - c\delta), -\delta\rangle + \langle\pi_t^s, \delta\rangle \implies f'(0) = \langle\pi_t^s, -\delta\rangle + \langle\pi_t^s, \delta\rangle = 0$$

$$f''(c) = \langle\delta, \nabla^2\phi^*(\nabla\phi(\pi_t^s) - c\delta)\delta\rangle \implies f''(0) = \langle\delta, \nabla^2\phi^*(\nabla\phi(\pi_t^s))\delta\rangle.$$

By the second-order Taylor series expansion of $f(c)$ around $c = 0$,

$$f(c) \approx f(0) + f'(0)(c - 0) + \frac{f''(0)\,(c-0)^2}{2} = \frac{c^2}{2}\langle\delta, \nabla^2\phi^*(\nabla\phi(\pi_t^s))\delta\rangle$$

Let us first consider the softmax case with the log-sum-exp mirror map, where $\pi^s = z(s,\cdot)$ and $\phi(z(s,\cdot)) = \log(\sum_a \exp(z(s,a)))$, $\phi^*(p^\pi(\cdot|s)) = \sum_a p^\pi(a|s)\log(p^\pi(a|s))$. Since the negative entropy and log-sum-exp are Fenchel conjugates (see Lemma 9), $\nabla\phi(z_t(s,\cdot)) = p^{\pi_t}(\cdot|s)$. Hence, we need to compute $\nabla^2\phi^*(p^{\pi_t}(\cdot|s))$.

$$\nabla\phi^*(p^{\pi_t}(\cdot|s)) = 1 + \log(p^{\pi_t}(\cdot|s)) \quad ; \quad \nabla^2\phi^*(p^{\pi_t}(\cdot|s)) = \text{diag}\left(1/p^{\pi_t}(\cdot|s)\right)$$

For the softmax representation, using the policy gradient theorem, $[\delta]_a = p^{\pi_t}(a|s)[A(s,a) - \hat{A}(s,a)]$ and hence,

$$\langle\delta, \nabla^2\phi^*(\nabla\phi(\pi_t))\delta\rangle = \sum_a p^{\pi_t}(a|s)[A(s,a) - \hat{A}(s,a)]^2 \, .$$

Hence, for the softmax representation, the second-order Taylor series expansion around $c = 0$ is equal to,

$$f(c) \approx \frac{c^2}{2} \sum_a p^{\pi_t}(a|s)[A(s,a) - \hat{A}(s,a)]^2 \, .$$

Now let us consider the direct case, where $\pi^s = p^\pi(\cdot|s)$, $\phi(p^\pi(\cdot|s)) = \sum_a p^\pi(a|s)\log(p^\pi(a|s))$, $\phi^*(z(s,\cdot)) = \log(\sum_a \exp(z(s,a)))$. Since the negative entropy and log-sum-exp are Fenchel conjugates (see Lemma 9), $\nabla\phi(p^{\pi_t}(\cdot|s)) =$

$z_t(s, \cdot)$. Hence, we need to compute $\nabla^2 \phi^*(z_t(s, \cdot))$.

$$[\nabla \phi^*(z_t(s, \cdot))]_a = \frac{\exp z_t(s, a)}{\sum_{a'} \exp(z_t(s, a'))} = p^{\pi_t}(a|s) \quad ; \quad [\nabla^2 \phi^*(z_t(s, \cdot))]_{a,a} = p^{\pi_t}(a|s) - [p^{\pi_t}(a|s)]^2$$

$$[\nabla^2 \phi^*(z_t(s, \cdot))]_{a,a'} = -p^{\pi_t}(a|s) \, p^{\pi_t}(a'|s) \implies \nabla^2 \phi^*(z_t(s, \cdot)) = \mathrm{diag}(p^{\pi_t}(\cdot|s)) - p^{\pi_t}(\cdot|s) \, [p^{\pi_t}(\cdot|s)]^\mathsf{T}.$$

For the direct representation, using the policy gradient theorem, $[\delta]_a = Q^{\pi_t}(s, a) - \hat{Q}^{\pi_t}(s, a)$ and hence,

$$\langle \delta, \nabla^2 \phi^*(\nabla \phi(\pi_t)) \, \delta \rangle = [Q^{\pi_t}(s, \cdot) - \hat{Q}^{\pi_t}(s, \cdot)]^\mathsf{T} \, [\mathrm{diag}(p^{\pi_t}(\cdot|s)) - p^{\pi_t}(\cdot|s) \, [p^{\pi_t}(\cdot|s)]^\mathsf{T}] \, [Q^{\pi_t}(s, \cdot) - \hat{Q}^{\pi_t}(s, \cdot)]$$

$$= \sum_a p^{\pi_t}(a|s) \, [Q^{\pi_t}(s, a) - \hat{Q}^{\pi_t}(s, a)]^2 - \left[ \langle p^{\pi_t}(a|s), Q^{\pi_t}(s, \cdot) - \hat{Q}^{\pi_t}(s, \cdot) \rangle \right]^2$$

$$= \sum_a p^{\pi_t}(a|s) \, [Q^{\pi_t}(s, a) - \hat{Q}^{\pi_t}(s, a)]^2 - \left[ J_s(\pi_t) - \hat{J}_s(\pi_t) \right]^2$$

(where $\hat{J}_s(\pi_t)$ is the estimated value function for starting state $s$)

Hence, for the direct representation, the second-order Taylor series expansion around $c = 0$ is equal to,

$$f(c) \approx \frac{c^2}{2} \left[ \sum_a p^{\pi_t}(a|s) \, [Q^{\pi_t}(s, a) - \hat{Q}^{\pi_t}(s, a)]^2 - \left[ J_s(\pi_t) - \hat{J}_s(\pi_t) \right]^2 \right]$$

$$= \frac{c^2}{2} \left[ \sum_a p^{\pi_t}(a|s) \, [Q^{\pi_t}(s, a) - \hat{Q}^{\pi_t}(s, a)]^2 - 2 \left[ J_s(\pi_t) - \hat{J}_s(\pi_t) \right]^2 \sum_a p^{\pi_t}(a|s) + \left[ J_s(\pi_t) - \hat{J}_s(\pi_t) \right]^2 \sum_a p^{\pi_t}(a|s) \right]$$

$$= \frac{c^2}{2} \left[ \sum_a p^{\pi_t}(a|s) \, [Q^{\pi_t}(s, a) - \hat{Q}^{\pi_t}(s, a)]^2 - 2 \left[ J_s(\pi_t) - \hat{J}_s(\pi_t) \right] \sum_a p^{\pi_t}(a|s) [Q^{\pi_t}(s, a) - \hat{Q}^{\pi_t}(s, a)] \right.$$

$$\left. + \left[ J_s(\pi_t) - \hat{J}_s(\pi_t) \right]^2 \sum_a p^{\pi_t}(a|s) \right]$$

$$= \frac{c^2}{2} \left[ \sum_a p^{\pi_t}(a|s) \left[ [Q^{\pi_t}(s, a) - \hat{Q}^{\pi_t}(s, a)]^2 - 2 \left[ J_s(\pi_t) - \hat{J}_s(\pi_t) \right] [Q^{\pi_t}(s, a) - \hat{Q}^{\pi_t}(s, a)] + \left[ J_s(\pi_t) - \hat{J}_s(\pi_t) \right]^2 \right] \right]$$

$$= \frac{c^2}{2} \left[ \sum_a p^{\pi_t}(a|s) \left( [Q^{\pi_t}(s, a) - \hat{Q}^{\pi_t}(s, a)] - [J_s(\pi_t) - \hat{J}_s(\pi_t)] \right)^2 \right]$$

$$= \frac{c^2}{2} \left[ \sum_a p^{\pi_t}(a|s) \left[ A^{\pi_t}(s, a) - \hat{A}^{\pi_t}(s, a) \right]^2 \right]$$

$\square$

### E.1 Bandit examples to demonstrate the benefit of the decision-aware loss

**Proposition 15** (Detailed version of Prop. 5)**.** Consider a two-armed bandit example with deterministic rewards where arm 1 is optimal and has a reward $r_1 = Q_1 = 2$ whereas arm 2 has reward $r_2 = Q_2 = 1$. Using a linear parameterization for the critic, $Q$ function is estimated as: $\hat{Q} = x\,\omega$ where $\omega$ is the parameter to be learned and $x$ is the feature of the corresponding arm. Let $x_1 = -2$ and $x_2 = 1$ implying that $\hat{Q}_1(\omega) = -2\omega$ and $\hat{Q}_2(\omega) = \omega$. Let $p_t$ be the probability of pulling the optimal arm at iteration $t$, and consider minimizing two alternative objectives to estimate $\omega$:

(1) Squared loss: $\omega_t^{(1)} := \arg\min \left\{ \frac{p_t}{2}\, [\hat{Q}_1(\omega) - Q_1]^2 + \frac{1-p_t}{2}\, [\hat{Q}_2(\omega) - Q_2]^2 \right\}$.

(2) Decision-aware critic loss: $\omega_t^{(2)} := \arg\min \mathcal{L}_t(\omega) := p_t\,[Q_1 - \hat{Q}_1(\omega)] + (1 - p_t)\,[Q_2 - \hat{Q}_2(\omega)] + \frac{1}{c} \log\left( p_t \exp\left(-c\,[Q_1 - \hat{Q}_1(\omega)]\right) + (1 - p_t) \exp\left(-c\,[Q_2 - \hat{Q}_2(\omega)]\right) \right)$.

Using the tabular parameterization for the actor, the policy update at iteration $t$ is given by: $p_{t+1} = \frac{p_t \exp(\eta \hat{Q}_1)}{p_t \exp(\eta \hat{Q}_1) + (1 - p_t)\ \exp(\eta \hat{Q}_2)}$, where $\eta$ is the functional step-size for the actor. For $p_0 < \frac{2}{5}$, minimizing the squared loss results in convergence to the sub-optimal action, while minimizing the decision-aware loss (for $c, p_0 > 0$) results in convergence to the optimal action.

*Proof.* Note that $\hat{Q}_1(\omega) - Q_1 = -2(\omega + 1)$ and $\hat{Q}_2(\omega) - Q_2 = \omega - 1$. Calculating $\omega^{(1)}$ for a general policy s.t. the probability of pulling the optimal arm equal to $p$,

$$\text{MSE}(\omega) = \frac{p}{2}\,[\hat{Q}_1(\omega) - Q_1]^2 + \frac{1-p}{2}\,[\hat{Q}_2(\omega) - Q_2]^2 = \frac{1}{2}\left[4p\,(\omega + 1)^2 + (1 - p)\,(\omega - 1)^2\right]$$
$$\implies \nabla_\omega \text{MSE}(\omega) = 4p\,(\omega + 1) + (1 - p)\,(\omega - 1)$$

Setting the gradient to zero,

$$\implies \omega^{(1)} = \frac{1 - 5p}{3p + 1}$$

Calculating $\omega^{(2)}$ for a general policy s.t. the probability of pulling the optimal arm equal to $p$,

$$L_t(\omega) = 2p\,(\omega + 1) - (1 - p)\,(\omega - 1) + \frac{1}{c} \log(p \exp(-2c\,(\omega + 1)) + (1 - p)\,\exp(c\,(\omega - 1)))$$
$$\implies \nabla_\omega L_t(\omega) = (3p - 1) + \frac{1}{c} \nabla_\omega \left[\log(p \exp(-2c\,(\omega + 1)) + (1 - p)\,\exp(c\,(\omega - 1)))\right]$$

Setting the gradient to zero,

$$\implies \nabla_\omega \left[\log(p \exp(-2c\,(\omega + 1)) + (1 - p)\,\exp(c\,(\omega - 1)))\right] = (1 - 3p)\,c$$

Define $A := \exp(-2c\,(\omega + 1))$ and $B := \exp(c\,(\omega - 1)))$

$$\implies \frac{-2p\,c\,A + (1 - p)\,c\,B}{p\,A + (1 - p)\,B} = (1 - 3p)\,c \implies \frac{A}{p\,A + (1 - p)\,B} = 1 \implies \omega^{(2)} = \frac{-1}{3}.$$

Now, let us consider the actor update,

$$p_{t+1} = \frac{p_t \exp(\eta \hat{Q}_1)}{p_t \exp(\eta \hat{Q}_1) + (1 - p_t)\ \exp(\eta \hat{Q}_2)} \implies \frac{p_{t+1}}{p_t} = \frac{1}{p_t + (1 - p_t)\ \exp(\eta\,(\hat{Q}_2 - \hat{Q}_1))}$$

Since arm 1 is optimal, if $\frac{p_{t+1}}{p_t} < 1$ for all $t$, the algorithm will converge to the sub-optimal arm. This happens when $\frac{1}{p_t + (1 - p_t)\ \exp(\eta\,(\hat{Q}_2 - \hat{Q}_1))} < 1 \implies \hat{Q}_2 - \hat{Q}_1 > 0 \implies \omega > 0$. Hence, for any $\eta$ and any iteration $t$, if $\omega_t > 0$, $p_{t+1} < p_t$. For the decision-aware critic loss, $\omega_t^{(2)} = -\frac{1}{3}$ for all $t$, implying that $p_{t+1} > p_t$ and hence the algorithm will converge to the optimal policy for any $\eta$ and any initialization $p_0 > 0$. However, for the squared MSE loss, $\omega_t^{(2)} = \frac{1 - 5p_t}{3p_t + 1}$, $\omega_t^{(2)} > 0$ if $p_t < \frac{1}{5}$. Hence, if $p_0 < \frac{1}{5}$, $p_1 < p_0 < \frac{1}{5}$. Using the same reasoning, $p_2 < p_1 < p_0 < 1/5$, and hence the policy will converge to the sub-optimal arm. $\qquad\square$

**Proposition 16.** Consider two-armed bandit problem with deterministic rewards - arm 1 has a reward $r_1 = Q_1$ whereas arm 2 has a reward $r_2 = Q_2$ such that arm 1 is the optimal, i.e. $Q_1 \geq Q_2$. Using a linear parameterization for the critic, $Q$ function is estimated as: $\hat{Q} = x\,\omega$ where $\omega$ is the parameter to be learned and $x$ is the feature of the corresponding arm. Let $p_t$ be the probability of pulling the optimal arm at iteration $t$, and consider minimizing the decision-aware critic loss to estimate $\omega$: $\omega_t := \arg\min \mathcal{L}_t(\omega) := p_t\,[Q_1 - \hat{Q}_1(\omega)] + (1 - p_t)\,[Q_2 - \hat{Q}_2(\omega)] + \frac{1}{c}\log\left(p_t\,\exp\left(-c\,[Q_1 - \hat{Q}_1(\omega)]\right) + (1 - p_t)\,\exp\left(-c\,[Q_2 - \hat{Q}_2(\omega)]\right)\right)$.
Using the tabular parameterization for the actor, the policy update at iteration $t$ is given by: $p_{t+1} = \frac{p_t\,\exp(\eta\hat{Q}_1)}{p_t\,\exp(\eta\hat{Q}_1) + (1 - p_t)\,\exp(\eta\hat{Q}_2)}$, where $\eta$ is the functional step-size for the actor. For the above problem, minimizing the decision-aware loss (for $c, p_0 > 0$) results in convergence to the optimal action, and $\mathcal{L}_t(\omega_t) = 0$ for any iteration $t$.

*Proof.* Define $A := \exp(-c[Q_1 - \hat{Q}_1(\omega)])$ and $B := \exp(-c[Q_2 - \hat{Q}_2(\omega)])$. Calculating the gradient of $\mathcal{L}_t$ w.r.t $\omega$ and setting it to zero,

$$\nabla_\omega \mathcal{L}_t(\omega) = p_t\,x_1 + (1 - p_t)\,x_2 - \frac{p_t\,x_1\,A + (1 - p_t)\,x_2\,B}{p_t\,A + (1 - p_t)\,B} = 0$$
$$\implies p_t\,(1 - p_t)\,A\,(x_1 - x_2) = p_t\,(1 - p_t)\,B\,(x_1 - x_2)$$
$$\implies Q_1 - x_1\,\omega_t = Q_2 - x_2\,\omega_t \implies \omega_t = \frac{Q_1 - Q_2}{x_1 - x_2}.$$

Observe that $Q_1 - \hat{Q}_1(\omega_t) = Q_2 - \hat{Q}_2(\omega_t)$ and thus $\mathcal{L}_t(\omega_t) = 0$ for all $t$. Writing the actor update,

$$p_{t+1} = \frac{p_t\,\exp(\eta\,x_1\,\omega_t)}{p_t\,\exp(\eta\,x_1\,\omega_t) + (1 - p_t)\,\exp(\eta\,x_2\,\omega_t)}$$
$$\implies \frac{p_{t+1}}{p_t} = \frac{1}{p_t + (1 - p_t)\exp(\eta\,(x_2 - x_1)\omega_t)} = \frac{1}{p_t + (1 - p_t)\exp(\eta\,(Q_2 - Q_1))} \geq 1$$

$\square$

**Proposition 17** (Detailed version of Prop. 7). Consider a two-armed bandit example and define $p \in [0, 1]$ as the probability of pulling arm 1. Given $p$, let the advantage of arm 1 be equal to $A_1 := \frac{1}{2} > 0$, while that of arm 2 is $A_2 := -\frac{p}{2(1-p)} < 0$ implying that arm 1 is optimal. For the critic, consider approximating the advantage of the two arms using a discrete hypothesis class with two hypotheses that depend on $p$ for: $\mathcal{H}_0 : \hat{A}_1 = \frac{1}{2} + \varepsilon, \hat{A}_2 = -\frac{p}{1-p}\left(\frac{1}{2} + \varepsilon\right)$ and $\mathcal{H}_1 : \hat{A}_1 = \frac{1}{2} - \varepsilon\,\mathrm{sgn}\left(\frac{1}{2} - p\right), \hat{A}_2 = -\frac{p}{1-p}\left(\frac{1}{2} - \varepsilon\,\mathrm{sgn}\left(\frac{1}{2} - p\right)\right)$ where sgn is the signum function and $\varepsilon \in \left(\frac{1}{2}, 1\right)$. If $p_t$ is the probability of pulling arm 1 at iteration $t$, consider minimizing two alternative loss functions to choose the hypothesis $\mathcal{H}_t$:
(1) Squared **(MSE)** loss: $\mathcal{H}_t = \arg\min_{\{\mathcal{H}_0, \mathcal{H}_1\}}\left\{\frac{p_t}{2}\,[A_1 - \hat{A}_1]^2 + \frac{1 - p_t}{2}\,[A_2 - \hat{A}_2]^2\right\}$.
(2) Decision-aware critic loss **(DA)** with $c = 1$: $\mathcal{H}_t = \arg\min_{\{\mathcal{H}_0, \mathcal{H}_1\}}$
$\left\{p_t\,(1 - [A_1 - \hat{A}_1])\,\log(1 - [A_1 - \hat{A}_1]) + (1 - p_t)\,(1 - [A_2 - \hat{A}_2])\,\log(1 - [A_2 - \hat{A}_2])\right\}$.

Using the tabular parameterization for the actor, the policy update at iteration $t$ is given by: $p_{t+1} = \frac{p_t\,(1 + \eta\,\hat{A}_1)}{p_t\,(1 + \eta\,\hat{A}_1) + (1 - p_t)\,(1 + \eta\,\hat{A}_2)}$.
For $p_0 \leq \frac{1}{2}$, the squared loss cannot distinguish between $\mathcal{H}_0$ and $\mathcal{H}_1$, and depending on how ties are broken, minimizing it can result in convergence to the sub-optimal action. On the other hand, minimizing the divergence loss (for any $p_0 > 0$) results in convergence to the optimal arm.

*Proof.* First note that when $p > \frac{1}{2}$, $\mathcal{H}_0$ and $\mathcal{H}_1$ are identical, ensure that $\hat{A}_1 > \hat{A}_2$ and the algorithm will converge to the optimal arm no matter which hypothesis is chosen. The regime of interest is therefore when $p \leq \frac{1}{2}$ and we focus on this case. Let us calculate the MSE and decision-aware (DA) losses for $\mathcal{H}_0$.

$$A_1 - \hat{A}_1 = \frac{1}{2} - \left(\frac{1}{2} + \varepsilon\right) = -\varepsilon \quad ; \quad A_2 - \hat{A}_2 = -\frac{p}{1-p}\,[1 - (1 + \varepsilon)] = \frac{p}{1-p}\,\varepsilon$$

$$\mathrm{MSE}(\hat{A}_1, \hat{A}_2) = p\,\varepsilon^2 + (1 - p)\left(\frac{p}{1-p}\,\varepsilon\right)^2 = p\,\varepsilon^2 + \frac{\varepsilon^2\,p^2}{1-p}$$

$$\mathrm{DA}(\hat{A}_1, \hat{A}_2) = p\,(1 + \varepsilon)\,\log(1 + \varepsilon) + (1 - p)\left(1 - \frac{\varepsilon\,p}{1-p}\right)\log\left(1 - \frac{\varepsilon\,p}{1-p}\right)$$

Similarly, we can calculate the MSE and decision-aware losses for $\mathcal{H}_1$.

$$A_1 - \hat{A}_1 = \frac{1}{2} - \left(\frac{1}{2} - \varepsilon\right) = \varepsilon \quad ; \quad A_2 - \hat{A}_2 = -\frac{p}{1-p}\left[\frac{1}{2} - \left(\frac{1}{2} - \varepsilon\right)\right] = -\frac{p}{1-p}\varepsilon$$

$$\text{MSE}(\hat{A}_1, \hat{A}_2) = p\,\varepsilon^2 + (1-p)\left(\frac{p}{1-p}\varepsilon\right)^2 = p\varepsilon^2 + \frac{\varepsilon^2\,p^2}{1-p}$$

$$\text{DA}(\hat{A}_1, \hat{A}_2) = p\,(1-\varepsilon)\log(1-\varepsilon) + (1-p)\left(1 + \frac{\varepsilon\,p}{1-p}\right)\log\left(1 + \frac{\varepsilon\,p}{1-p}\right)$$

For both $\mathcal{H}_0$ and $\mathcal{H}_1$, the MSE loss is equal to $p\varepsilon^2 + \frac{\varepsilon^2 p^2}{1-p}$ and hence it cannot distinguish between the two hypotheses. Writing the actor update,

$$p_{t+1} = \frac{p_t\,(1 + \eta\,\hat{A}_1)}{p_t\,(1 + \eta\,\hat{A}_1) + (1 - p_t)\,(1 + \eta\,\hat{A}_2)} \implies \frac{p_{t+1}}{p_t} = \frac{1}{p_t + (1 - p_t)\frac{1+\eta\hat{A}_2}{1+\eta\hat{A}_1}}$$

Hence, in order to ensure that $p_{t+1} > p_t$ and eventual convergence to the optimal arm, we want that $\hat{A}_2 < \hat{A}_1$. For $\varepsilon \in \left(\frac{1}{2}, 1\right)$, for $\mathcal{H}_0$, $\hat{A}_1 > 0$ while $\hat{A}_2 < 0$. On the other hand, for $\mathcal{H}_1$, $\hat{A}_1 < 0$ and $\hat{A}_2 > 0$. This implies that the algorithm should choose $\mathcal{H}_0$ in order to approximate the advantage. Since the MSE loss is the same for both hypotheses, convergence to the optimal arm depends on how the algorithm breaks ties. Next, we prove that for the decision-aware loss and any iteration such that $p_t < 0.5$, the loss for $\mathcal{H}_0$ is smaller than that for $\mathcal{H}_1$, and hence the algorithm chooses the correct hypothesis and pulls the optimal arm. For this, we define $f(p)$ as follows,

$$f(p) := \left[p\,(1+\varepsilon)\log(1+\varepsilon) + (1-p)\left(1 - \frac{\varepsilon\,p}{1-p}\right)\log\left(1 - \frac{\varepsilon\,p}{1-p}\right)\right]$$
$$- \left[p\,(1-\varepsilon)\log(1-\varepsilon) + (1-p)\left(1 + \frac{\varepsilon\,p}{1-p}\right)\log\left(1 + \frac{\varepsilon\,p}{1-p}\right)\right]$$

For $f(p)$ to be well-defined, we want that, $1 - \epsilon > 0 \implies \epsilon < 1$ and $1 - \frac{\epsilon p}{1-p} > 0 \implies p < \frac{1}{1+\epsilon}$. Since $\epsilon \in (1/2, 1)$, $p < \frac{1}{2}$. In order to prove that the algorithm will always choose $\mathcal{H}_0$, we will show that $f(p) \le 0$ for all $p \in [0, 1/2]$ next. First note that,

$$f(0) = 0 \quad ; \quad f(1/2) = \frac{1+\varepsilon}{2}\log(1+\varepsilon) + \frac{(1-\varepsilon)}{2}\log(1-\varepsilon) - \frac{1-\varepsilon}{2}\log(1-\varepsilon) - \frac{1+\varepsilon}{2}\log(1+\varepsilon) = 0$$

Next, we will prove that $f(p)$ is convex. This combined with the fact $f(0) = f(1/2) = 0$ implies that $f(p) < 0$ for all $p \in (0, 1/2)$. For this, we write $f(p) = g(p) + h_1(p) - h_2(p)$ where,

$$g(p) = p\,(1+\varepsilon)\log(1+\varepsilon) - p\,(1-\varepsilon)\log(1-\varepsilon)$$

$$h_1(p) = (1-p)\left(1 - \frac{\varepsilon\,p}{1-p}\right)\log\left(1 - \frac{\varepsilon\,p}{1-p}\right) = (1 - \epsilon'p)\log\left(\frac{1 - \epsilon'\,p}{1-p}\right) \qquad (\epsilon' = 1 + \epsilon)$$

$$h_2(p) = (1-p)\left(1 + \frac{\varepsilon\,p}{1-p}\right)\log\left(1 + \frac{\varepsilon\,p}{1-p}\right) = (1 - \epsilon''p)\log\left(\frac{1 - \epsilon''\,p}{1-p}\right) \qquad (\epsilon'' = 1 - \epsilon)$$

Differentiating the above terms,

$$g'(p) = (1+\varepsilon)\log(1+\varepsilon) - (1-\varepsilon)\log(1-\varepsilon) \quad ; \quad g''(p) = 0$$

$$h_1'(p) = -\epsilon'\log\left(\frac{1 - \epsilon'\,p}{1-p}\right) + \frac{1 - \epsilon'}{1-p}$$

$$h_1''(p) = -\frac{\epsilon'}{1 - \epsilon'p}\frac{1 - \epsilon'}{1-p} - \frac{(1-\epsilon')^2}{(1-p)^2} = \frac{(\epsilon' - 1)\,p\left[(\epsilon' - 1)^2 + \left(\frac{1}{p} - 1\right)\right]}{(1 - \epsilon'p)\,(1-p)^2} > 0$$

Similarly,

$$h_2''(p) = \frac{(\epsilon'' - 1)\,p\left[(\epsilon'' - 1)^2 + \left(\frac{1}{p} - 1\right)\right]}{(1 - \epsilon''p)\,(1-p)^2} < 0$$

Combining the above terms, $f''(p) = g''(p) + h_1''(p) - h_2''(p) > 0$ for all $p \in (0, 1/2)$ and hence $f(p)$ is convex. Hence, for all $p < \frac{1}{2}$, minimizing the divergence loss results in choosing $\mathcal{H}_0$ and the actor pulling the optimal arm. Once the probability of

pulling the optimal arm is larger than $0.5$, both hypotheses are identical and the algorithm will converge to the optimal arm regardless of the hypothesis chosen. $\square$

### E.2 Lemmas

**Lemma 7.** *For a probability distribution $p \in \mathbb{R}^A$, the negative entropy mirror map $\phi(p) = \sum_i p_i \log(p_i)$, $\delta \in \mathbb{R}^A$, $c > 0$,*

$$D_{\phi^*}\left(\nabla\phi(p) - c\,\delta, \nabla\phi(p)\right) = c\langle p, \delta \rangle + \log\left(\sum_j p_j \exp(-c\delta_j)\right).$$

*Proof.* In this case, $[\nabla\phi(p)]_i = 1 + \log(p_i)$. Hence, we need to compute $D_{\phi^*}(z', z)$ where $z'_i := 1 + \log(p_i) - c\delta_i$ and $z_i := 1 + \log(p_i)$. If $\phi(p) = \sum_i p_i \log(p_i)$, using Lemma 9, $\phi^*(z) = \log\left(\sum_i \exp(z_i)\right)$ where $z_i - \log(\sum_i \exp(z_i)) = \log(p_i)$. Define distribution $q$ such that $q_i := \frac{\exp(1+\log(p_i)-c\delta_i)}{\sum_j \exp(1+\log(p_j)-c\delta_j)}$. Using Lemma 11,

$$D_{\phi^*}(z', z) = \mathrm{KL}(p||q) = \sum_i p_i \log\left(\frac{p_i}{q_i}\right)$$

Simplifying $q$,

$$q_i = \frac{\exp(1 + \log(p_i) - c\delta_i)}{\exp(\sum_j(1 + \log(p_j) - c\delta_j))} = \frac{p_i \exp(-c\delta_i)}{\sum_j p_j \exp(-c\delta_j)}$$

$$\implies D_{\phi^*}(z', z) = \sum_i p_i \log\left(\frac{p_i \sum_j p_j \exp(-c\delta_j)}{p_i \exp(-c\delta_i)}\right) = \sum_i p_i \log\left(\exp(c\delta_i) \sum_j p_j \exp(-c\delta_j)\right)$$

$$= c\sum_i p_i \delta_i + \sum_i p_i \log\left(\sum_j p_j \exp(-c\delta_j)\right) = c\langle p, \delta\rangle + \log\left(\sum_j p_j \exp(-c\delta_j)\right)$$

$\square$

**Lemma 8.** *For $z \in \mathbb{R}^A$, the log-sum-exp mirror map $\phi(z) = \log(\sum_i \exp(z_i))$, $\delta \in \mathbb{R}^A$ s.t. $\sum_i \delta_i = 0$, $c > 0$,*

$$D_{\phi^*}\left(\nabla\phi(z) - c\,\delta, \nabla\phi(z)\right) = \sum_i (p_i - c\delta_i) \log\left(\frac{p_i - c\delta_i}{p_i}\right),$$

*where $p_i = \frac{\exp(z_i)}{\sum_j \exp(z_j)}$.*

*Proof.* In this case, $[\nabla\phi(z)]_i = \frac{\exp(z_i)}{\sum_j \exp(z_j)} = p_i$. Define distribution $q$ s.t. $q_i := p_i - c\delta_i$. Note that since $\sum_i \delta_i = 0$, $\sum_i q_i = \sum_i p_i = 1$ and hence, $q$ is a valid distribution. We thus need to compute $D_{\phi^*}(q, p)$. Using Lemma 9, $\phi^*(p) = \sum_i p_i \log(p_i)$ where $p_i = \frac{\exp(z_i)}{\sum_j \exp(z_j)}$. Using Lemma 10,

$$D_{\phi^*}(q, p) = \mathrm{KL}(q||p) = \sum_i (p_i - c\delta_i) \log\left(\frac{p_i - c\delta_i}{p_i}\right)$$

$\square$

**Lemma 9.** *The log-sum-exp mirror map on the logits and the negative entropy mirror map on the corresponding probability distribution are Fenchel duals. In particular for $z \in \mathbb{R}^d$, if $\phi(z) := \log\left(\sum_i \exp(z_i)\right)$, then $\phi^*(p) = \sum_i p_i \log(p_i)$ where $p_i = \frac{\exp(z_i)}{\sum_j \exp(z_j)}$. Similarly, if $\phi(p) = \sum_i p_i \log(p_i)$, then $\phi^*(z) = \log\left(\sum_i \exp(z_i)\right)$ where $z_i - \log(\sum_i \exp(z_i)) = \log(p_i)$.*

*Proof.* If $\phi(z) := \log\left(\sum_i \exp(z_i)\right)$,

$$\phi^*(p) := \sup_z \left[\langle p, z \rangle - \phi(z)\right] = \sup_z \left[\sum_i p_i z_i - \log\left(\sum_i \exp(z_i)\right)\right]$$

Setting the gradient to zero, we get that $p_i = \frac{\exp(z_i^*)}{\sum_j \exp(z_j^*)}$ for $z^* \in \mathcal{Z}^*$ where $\mathcal{Z}^*$ is the set of maxima related by a shift (i.e. if $z^* \in \mathcal{Z}^*$, $z^* + C \in \mathcal{Z}^*$ for a constant $C$). Using the optimality condition, we know that $\sum_i p_i = 1$ and

$$\log(p_i) = z_i^* - \log\left(\sum_j \exp(z_j^*)\right) \implies z_i^* = \log(p_i) + \phi(z^*)$$

Using this relation,

$$\phi^*(p) = \left[\sum_i p_i z_i^* - \log\left(\sum_i \exp(z_i^*)\right)\right] = \left[\sum_i p_i \log(p_i) + \phi(z^*)\sum_i p_i - \phi(z^*)\right]$$

$$\implies \phi^*(p) = \sum_i p_i \log(p_i)$$

The second statement follows since the $\phi^*(\phi^*) = \phi$. $\qquad\square$

**Lemma 10.** *For probability distributions, $p$ and $p'$, if $\phi(p) = \sum_i p\log(p_i)$, then $D_\phi(p,p') = KL(p||p')$.*

*Proof.* Note that $[\nabla\phi(p)]_i = 1 + \log(p_i)$. Using the definition of the Bregman divergence,

$$D_\phi(p,p') := \phi(p) - \phi(p') - \langle \nabla\phi(p'), p - p'\rangle$$
$$= \sum_i [p_i \log(p_i) - p_i' \log(p_i') - (1 + \log(p_i'))(p_i - p_i')]$$
$$= \sum_i \left[p_i \log\left(\frac{p_i}{p_i'}\right)\right] - \sum_i p_i + \sum_i p_i'$$

Since $p$ and $p'$ are valid probability distributions, $\sum_i p_i = \sum_i p_i' = 1$, and hence, $D_\phi(p,p') = KL(p||p')$. $\qquad\square$

**Lemma 11.** *If $\phi(z) = \log(\sum_i \exp(z_i))$, then $D_\phi(z,z') = KL(p'||p)$, where $p_i := \frac{\exp(z_i)}{\sum_j \exp(z_j)}$ and $p_i' := \frac{\exp(z_i')}{\sum_j \exp(z_j')}$.*

*Proof.* Note that $[\nabla\phi(z)]_i = \frac{\exp(z_i)}{\sum_j \exp(z_j)} = p_i$ where $p_i := \frac{\exp(z_i)}{\sum_j \exp(z_j)}$. Using the definition of the Bregman divergence,

$$D_\phi(z,z') := \phi(z) - \phi(z') - \langle \nabla\phi(z'), z - z'\rangle$$
$$= \log\left(\sum_j \exp(z_j)\right) - \log\left(\sum_j \exp(z_j')\right) - \sum_i \left[\frac{\exp(z_i')}{\sum_j \exp(z_j')}(z_i - z_i')\right]$$
$$= \sum_i p_i'\left[\log\left(\sum_j \exp(z_j)\right) - \log\left(\sum_j \exp(z_j')\right) - z_i + z_i'\right] \qquad \text{(Since } \sum_i p_i' = 1\text{)}$$
$$= \sum_i p_i'\left[\log\left(\sum_j \exp(z_j)\right) - \log\left(\sum_j \exp(z_j')\right) - \log(\exp(z_i)) + \log(\exp(z_i'))\right]$$
$$= \sum_i p_i'\left[\log\left(\frac{\exp(z_i')}{\sum_j \exp(z_j')}\right) - \log\left(\frac{\exp(z_i)}{\sum_j \exp(z_j)}\right)\right]$$
$$= \sum_i p_i'[\log(p_i') - \log(p_i')] = \sum_i p_i'\left[\log\left(\frac{p_i'}{p_i}\right)\right] = KL(p'||p)$$

$\qquad\square$

# F Implementation Details

## F.1 Heuristic to estimate $c$

We estimate $c$ to maximize the lower-bound on $J(\pi)$. In particular, using Prop. 1,

$$J(\pi) \geq J(\pi_t) + \hat{g}(\pi_t)^\top (\pi - \pi_t) - \left(\frac{1}{\eta} + \frac{1}{c}\right) D_\Phi(\pi, \pi_t) - \frac{1}{c} D_{\Phi^*}\left(\nabla\Phi(\pi_t) - c[\nabla J(\pi_t) - \hat{g}(\pi_t)], \nabla\Phi(\pi_t)\right)$$

For a fixed $\hat{g}(\pi_t)$, we need to maximize the RHS w.r.t $\pi$ and $c$, i.e.

$$\max_{c>0} \max_{\pi\in\Pi} J(\pi_t) + \hat{g}(\pi_t)^\top (\pi - \pi_t) - \left(\frac{1}{\eta} + \frac{1}{c}\right) D_\Phi(\pi, \pi_t) - \frac{1}{c} D_{\Phi^*}\left(\nabla\Phi(\pi_t) - c[\nabla J(\pi_t) - \hat{g}(\pi_t)], \nabla\Phi(\pi_t)\right) \quad (6)$$

Instead of maximizing w.r.t $\pi$ and $c$, we will next aim to find an upper-bound on the RHS that is independent of $\pi$ and aim to maximize it w.r.t $c$. Using Lemma 1 with $y' = \pi$, $y = \pi_t$, $x = -\hat{g}(\pi_t)$ and define $c'$ such that $\frac{1}{c'} = \frac{1}{\eta} + \frac{1}{c}$.

$$\langle -\hat{g}(\pi_t), \pi - \pi_t \rangle \geq -\frac{1}{c'}\left[D_\Phi(\pi, \pi_t) + D_\Phi^*(\nabla\Phi(\pi_t) + c'\hat{g}(\pi_t), \nabla\Phi(\pi_t))\right]$$

$$\implies J(\pi_t) + \langle \hat{g}(\pi_t), \pi - \pi_t \rangle - \left(\frac{1}{\eta'} + \frac{1}{c}\right) D_\Phi(\pi, \pi_t) \leq J(\pi_t) + \frac{1}{c'} D_\Phi^*(\nabla\Phi(\pi_t) + c'\hat{g}(\pi_t), \nabla\Phi(\pi_t))$$

Using the above upper-bound in Eq. (6),

$$\max_{c>0} \left[ J(\pi_t) + \frac{1}{c'} D_\Phi^*(\nabla\Phi(\pi_t) + c'\hat{g}(\pi_t), \nabla\Phi(\pi_t)) - \frac{1}{c} D_{\Phi^*}\left(\nabla\Phi(\pi_t) - c[\nabla J(\pi_t) - \hat{g}(\pi_t)], \nabla\Phi(\pi_t)\right)\right]$$

This implies that the estimate $\hat{c}$ can be calculated as:

$$\hat{c} = \arg\max_{c>0} \left\{ \left(\frac{1}{\eta} + \frac{1}{c}\right) D_\Phi^*\left(\nabla\Phi(\pi_t) + \frac{1}{\left(\frac{1}{\eta} + \frac{1}{c}\right)}\hat{g}(\pi_t), \nabla\Phi(\pi_t)\right) - \frac{1}{c} D_{\Phi^*}\left(\nabla\Phi(\pi_t) - c[\nabla J(\pi_t) - \hat{g}(\pi_t)], \nabla\Phi(\pi_t)\right)\right\}$$

In order to gain some intuition, let us consider the case where $D_\Phi(u, v) = \frac{1}{2}\|u - v\|_2^2$. In this case,

$$\hat{c} = \arg\max_{c>0} \left\{ \frac{\|\hat{g}(\pi_t)\|_2^2}{2\left(\frac{1}{\eta} + \frac{1}{c}\right)} - \frac{c}{2}\|\hat{g}(\pi_t) - \nabla J(\pi_t)\|_2^2\right\}$$

If $\|\hat{g}(\pi_t) - \nabla J(\pi_t)\|_2^2 \to 0$,

$$\hat{c} = \arg\max_{c>0} \left\{ \frac{\|\hat{g}(\pi_t)\|_2^2}{2\left(\frac{1}{\eta} + \frac{1}{c}\right)}\right\} \implies c \to \infty$$

If $\|\hat{g}(\pi_t) - \nabla J(\pi_t)\|_2^2 \to \infty$,

$$\hat{c} = \arg\max_{c>0} \left\{ -\frac{c}{2}\right\} \implies c \to 0$$

## F.2 Environments and constructing features

**Cliff World**: We consider a modified version of the CliffWorld environment [53, Example 6.6]. The environment is deterministic and consists of 21 states and 4 actions. The objective is to reach the Goal state as quickly as possible. If the agent falls into a Cliff, it yields reward of $-100$, and is then returned to the Start state. Reaching the Goal state yields a reward of $+1$, and the agent will stay in this terminal state. All other transitions are associated with a zero reward, and the discount factor is set to $\gamma = 0.9$.

**Frozen Lake**: We Consider the Frozen Lake v.1 environment from gym framework [6]. The environment is stochastic and consists of 16 states and 4 actions. The agent starts from the Start state and according to the next action (chosen by the policy) and the stochastic dynamics moves to the next state and yields a reward. The objective is to reach the Goal state as quickly as possible without entering the Hole States. All the Hole states and the Goal are terminal states. Reaching the goal state yields $+1$ reward and all other rewards are zero, and the discount factor is set to $\gamma = 0.9$.

**Sampling**: We employ the Monte-Carlo method to sample from both environments and we use the expected return to estimate the action-value function $Q$. Specifically, we iteratively start from a randomly chosen state-action pair $(s, a)$, run a roll-out with a specified length starting from that pair, and collect the expected return to estimate $Q(s, a)$.

**Constructing features**: Also, in order to use function approximation on the above environments, we use tile-coded features [53]. Specifically, tilde-coded featurization needs three parameters to be set: $(i)$ hash table size (equivalent to the feature dimension) $d$, $(ii)$ number of tiles $N$ and $(iii)$ size of tiles $s$. For Cliff world environment, we consider following pairs to construct features: $\{(d = 40, N = 5, s = 1), (d = 50, N = 6, s = 1), (d = 60, N = 4, s = 3), (d = 80, N = 5, s = 3), (d = 100, N = 6, s = 3)\}$. This means whenever we use $d = 40$, the number of tiles is $N = 5$ and the tiling size is $s = 1$. The reported number of tiles and tiling size parameters are tuned and have achieved the best performance for all algorithms. Similarly for Frozen Lake environment, we use the following pairs to construct features: $\{(d = 40, N = 3, s = 3), (d = 50, N = 4, s = 13), (d = 60, N = 5, s = 3), (d = 100, N = 8, s = 3)\}$.

### F.3 Critic optimization

We explain implementation of `MSE`, `Adv-MSE` and decision-aware critic loss functions. We use tile-coded features $\mathbf{X}(s, a)$ and linear function approximation to estimate action-value function $Q$, implying that $\hat{Q}(s, a) = \omega^T \mathbf{X}(s, a)$ where $\omega, \mathbf{X}(s, a) \in \mathbb{R}^d$.

**Baselines:** For policy $\pi$, the MSE objective is to return the $\omega$ that minimizes the squared norm error of the action-value function $Q^\pi$ across all state-actions weighted by the state-action occupancy measure $\mu^\pi(s, a)$.

$$\omega^{\text{MSE}} = \arg\min_{\omega \in \mathbb{R}^d} \mathbb{E}_{(s,a) \sim \mu^\pi(s,a)} [Q^\pi(s, a) - \omega^T \mathbf{X}(s, a)]^2$$

Taking the derivative with respect to $\omega$ and setting it to zero:

$$\mathbb{E}_{(s,a) \sim \mu^\pi(s,a)} \left[ \left( Q^\pi(s, a) - \omega^T \mathbf{X}(s, a) \right) \mathbf{X}(s, a)^T \right] = 0$$

$$\implies \underbrace{\sum_{s,a} \mu^\pi(s, a) Q^\pi(s, a) \mathbf{X}(s, a)^T}_{:=y} = \underbrace{\left[ \sum_{s,a} \mu^\pi(s, a) \mathbf{X}(s, a) \mathbf{X}(s, a)^T \right]}_{:=K} \omega$$

Given features $\mathbf{X}$, the true action-value function $Q^\pi$ and state-action occupancy measure $\mu^\pi$, we can compute $K$, $y$ and solve $\omega^{\text{MSE}} = K^{-1} y$.

Similarly for policy $\pi$, the advantage-MSE objective is to return $\omega$ that minimizes the squared error of the advantage function $A^\pi$ across all state-actions weighted by the state-action occupancy measure $\mu^\pi$.

$$\omega^{\text{Adv-MSE}} = \arg\min_{\omega \in \mathbb{R}^d} \mathbb{E}_{(s,a) \sim \mu^\pi(s,a)} \left[ A^\pi(s, a) - \omega^T \left( \mathbf{X}(s, a) - \sum_{a'} \mathbf{X}(s, a') \right) \right]^2$$

Taking the derivative with respect to $\omega$ and setting it to zero:

$$\mathbb{E}_{(s,a) \sim \mu^\pi(s,a)} \left[ \left[ A^\pi(s, a) - \omega^T \left( \mathbf{X}(s, a) - \sum_{a'} \mathbf{X}(s, a') \right) \right] \left[ \mathbf{X}(s, a) - \sum_{a'} \mathbf{X}(s, a') \right]^T \right] = 0$$

$$\implies \underbrace{\sum_{s,a} \mu^\pi(s, a) A^\pi(s, a) \left[ \mathbf{X}(s, a) - \sum_{a'} \mathbf{X}(s, a') \right]^T}_{:=y} = \underbrace{\sum_{s,a} \mu^\pi(s, a) \left[ \mathbf{X}(s, a) - \sum_{a'} \mathbf{X}(s, a') \right] \left[ \mathbf{X}(s, a) - \sum_{a'} \mathbf{X}(s, a') \right]^T}_{:=K} \omega$$

Given features $\mathbf{X}$, the true advantage function $A^\pi$ and state-action occupancy measure $\mu^\pi$, we ca compute $K$ and $y$ and solve $w^{\text{Adv-MSE}} = K^{-1} y$.

**Decision-aware critic in direct representation:** Recall that for policy $\pi$, the decision-aware critic loss in direct representation is the blue term in Prop. 4, which after linear parameterization on $\hat{Q}^\pi$ would be as follows:

$$\mathbb{E}_{s \sim d^\pi} \left[ \mathbb{E}_{a \sim p^\pi(\cdot|s)} [Q^\pi(s, a) - \omega^T \mathbf{X}(s, a)] + \frac{1}{c} \log \left( \mathbb{E}_{a \sim p^\pi(\cdot|s)} \left[ \exp \left( -c [Q^\pi(s, a) - \omega^T \mathbf{X}(s, a)] \right) \right] \right) \right]$$

The above term is a convex function of $\omega$ for any $c > 0$. We minimize the term using gradient descent, where the gradient with respect to $\omega$ is:

$$-\mathbb{E}_{s\sim d^\pi}\left[\mathbb{E}_{a\sim p^\pi(\cdot|s)}\mathbf{X}(s,a) - \frac{\mathbb{E}_{a\sim p^\pi(\cdot|s)}\left[\exp\left(-c\left[Q^\pi(s,a) - \omega^T\mathbf{X}(s,a)\right]\right)\mathbf{X}(s,a)\right]}{\mathbb{E}_{a\sim p^\pi(\cdot|s)}\left[\exp\left(-c\left[Q^\pi(s,a) - \omega^T\mathbf{X}(s,a)\right]\right)\right]}\right]$$

The step-size of gradient ascent is determined using Armijo line-search [3] where the maximum step size is set to 1000 and it decays with the rate $\beta = 0.9$. The number of iteration for critic inner-loop, $m_c$ in Algorithm 1, is set to 10000, and if the gradient norm becomes smaller than $10^{-6}$ we terminate the inner loop.

**Decision-aware critic in softmax representation:** Recall that for policy $\pi$, the decision-aware critic loss in softmax representation is the blue term in Prop. 6, which after linear parameterization on $\hat{Q}^\pi$ and substituting $\hat{A}^\pi(s,a)$ with $\omega^T\left(\mathbf{X}(s,a) - \sum_{a'}\mathbf{X}(s,a')\right)$ would be as follows:

$$\frac{1}{c}\mathbb{E}_{s\sim d^{\pi_t}}\mathbb{E}_{a\sim p^{\pi_t}(\cdot|s)}\left[\left(1 - c\left[A^{\pi_t}(s,a) - \omega^T(\mathbf{X}(s,a) - \sum_{a'}\mathbf{X}(s,a'))\right]\right)\log\left(1 - c\left[A^{\pi_t}(s,a) - \omega^T(\mathbf{X}(s,a) - \sum_{a'}\mathbf{X}(s,a'))\right]\right)\right]$$

Similarly, the above term is convex with respect to $\omega$ and we minimize it using gradient descent. The step-size is determined using Armijo line-search with the same parameters as mentioned in direct case. The number of iterations in inner loop is set to 10000 and we terminate the loop if the gradient norm becomes smaller than $10^{-8}$. The gradient with respect to $\omega$:

$$\mathbb{E}_{s\sim d^{\pi_t}}\mathbb{E}_{a\sim p^{\pi_t}(\cdot|s)}\left[\left(1 + \log\left(1 - c\left[A^{\pi_t}(s,a) - \omega^T(\mathbf{X}(s,a) - \sum_{a'}\mathbf{X}(s,a'))\right]\right)\right)\left(\mathbf{X}(s,a) - \sum_{a'}\mathbf{X}(s,a')\right)\right]$$

### F.4  Actor optimization

**Direct representation**: For all actor-critic algorithms, we maximize the green term in Prop. 4 known as MDPO [56].

$$\mathbb{E}_{s\sim d^{\pi_t}}\left[\mathbb{E}_{a\sim p^{\pi_t}(\cdot|s)}\left[\frac{p^\pi(a|s)}{p^{\pi_t}(a|s)}\left(\hat{Q}^{\pi_t}(s,a) - \left(\frac{1}{\eta} + \frac{1}{c}\right)\log\left(\frac{p^\pi(a|s)}{p^{\pi_t}(a|s)}\right)\right)\right]\right]$$

In tabular parameterization of the actor, $\theta_{s,a} = p^\pi(s,a)$, the actor update is exactly natural policy gradient [25] and can be solved in closed-form. We refer the reader to Appendix F.2 of [57] for explicit derivation. At iteration $t$, given policy $\pi_t$, the estimated action-value function from the critic $\hat{Q}^{\pi_t}$ and $\eta$ as the functional step-size, the update at iteration $t$ is:

$$p^{\pi_{t+1}}(a|s) = \frac{p^{\pi_t}(a|s)\exp\left(\eta\hat{Q}^{\pi_t}(s,a)\right)}{\sum_{a'}p^{\pi_t}(a'|s)\exp\left(\eta\hat{Q}^{\pi_t}(s,a')\right)} \implies \theta_{s,a} = \frac{\theta_{s,a}\exp\left(\eta\hat{Q}^{\pi_t}(s,a)\right)}{\sum_{a'}\theta_{s,a'}\exp\left(\eta\hat{Q}^{\pi_t}(s,a')\right)}$$

When we linearly parameterize the policy, implying that for policy $\pi$, $p^\pi(a|s) = \frac{\exp(\theta^T\mathbf{X}(s,a))}{\sum_{a'}\exp(\theta^T\mathbf{X}(s,a'))}$ where $\theta, \mathbf{X}(s,a) \in \mathbb{R}^n$ and $n$ is the actor expressivity, we use the off-policy update loop (Lines 10-13 in Algorithm 1) and we iteratively update the parameters using gradient ascent. The MDPO objective with linear parameterization will be:

$$\mathbb{E}_{s\sim d^{\pi_t}}\left[\mathbb{E}_{a\sim p^{\pi_t}(\cdot|s)}\left[\frac{\exp(\theta^T\mathbf{X}(s,a))}{p^{\pi_t}(a|s)\sum_{a'}\exp(\theta^T\mathbf{X}(s,a'))}\left(\hat{Q}^{\pi_t}(s,a) - \left(\frac{1}{\eta} + \frac{1}{c}\right)\log\left(\frac{\exp(\theta^T\mathbf{X}(s,a))}{p^{\pi_t}(a|s)\sum_{a'}\exp(\theta^T\mathbf{X}(s,a'))}\right)\right)\right]\right]$$

And the gradient of objective with respect to $\theta$ is:

$$\mathbb{E}_{s\sim d^{\pi_t}}\left[\mathbb{E}_{a\sim p^{\pi_t}(\cdot|s)}\left[\frac{p^\pi(a|s)}{p^{\pi_t}(a|s)}\left(\mathbf{X}(s,a) - \frac{\sum_{a'}\exp(\theta^T\mathbf{X}(s,a))\mathbf{X}(s,a')}{\sum_{a'}\exp(\theta^T\mathbf{X}(s,a'))}\right)\left(\hat{Q}^{\pi_t}(s,a) - (\frac{1}{\eta} + \frac{1}{c})(1 + \log(\frac{p^\pi(a|s)}{p^{\pi_t}(a|s)}))\right)\right]\right]$$

**Softmax representation**: For all actor-critic algorithms, we maximize the green term in Prop. 6 known as sMDPO [57].

$$\mathbb{E}_{s\sim d^{\pi_t}}\mathbb{E}_{a\sim p^{\pi_t}(\cdot|s)}\left[\left(\hat{A}^{\pi_t}(s,a) + \frac{1}{\eta} + \frac{1}{c}\right)\log\left(\frac{p^\pi(a|s)}{p^{\pi_t}(a|s)}\right)\right]$$

In tabular parameterization of the actor, $\theta_{s,a} = p^\pi(s,a)$, at iteration $t$ given the policy $\pi_t$, the estimated advantage function from the critic $\hat{A}^{\pi_t}(s,a) = \hat{Q}^{\pi_t}(s,a) - \sum_{a'}p^{\pi_t}(a|s)\hat{Q}^{\pi_t}(s,a')$, and functional step-size $\eta$, the actor update can be solved in closed-form and is as follows:

$$p^{\pi_{t+1}}(a|s) = \frac{p^{\pi_t}(a|s)\max(1+\eta A^{\pi_t}(s,a),0)}{\sum_{a'} p^{\pi_t}(a'|s)\max(1+\eta A^{\pi_t}(s,a'),0)} \implies \theta_{s,a} = \frac{\theta_{s,a}\max(1+\eta A^{\pi_t}(s,a),0)}{\sum_{a'} \theta_{s,a'}\max(1+\eta A^{\pi_t}(s,a'),0)}$$

We refer the reader to Appendix F.1 of [57] for explicit derivation. When we linearly parameterize the policy, implying that for policy $\pi$, $p^\pi(a|s) = \frac{\exp(\theta^T \mathbf{X}(s,a))}{\sum_{a'}\exp(\theta^T \mathbf{X}(s,a'))}$, we need to maximize the following with respect to $\theta$:

$$\mathbb{E}_{s\sim d^{\pi_t}} \mathbb{E}_{a\sim p^{\pi_t}(\cdot|s)} \left[ \left( \hat{A}^{\pi_t}(s,a) + \frac{1}{\eta} + \frac{1}{c} \right) \log \left( \frac{\exp(\theta^T \mathbf{X}(s,a))}{p^{\pi_t}(a|s)\sum_{a'}\exp(\theta^T \mathbf{X}(s,a'))} \right) \right]$$

Similar to direct representation, we use the off-policy update loop and we iteratively update the parameters using gradient ascent. The gradient with respect to $\theta$ is:

$$\mathbb{E}_{s\sim d^{\pi_t}} \mathbb{E}_{a\sim p^{\pi_t}(\cdot|s)} \left[ \left( \hat{A}^{\pi_t}(s,a) + \frac{1}{\eta} + \frac{1}{c} \right) \left( \mathbf{X}(s,a) - \frac{\sum_{a'}\exp(\theta^T\mathbf{X}(s,a))\mathbf{X}(s,a')}{\sum_{a'}\exp(\theta^T\mathbf{X}(s,a))} \right) \right]$$

### F.5 Parameter Tuning

| | Parameter | Value/Range |
|---|---|---|
| **Sampling** | # of samples | $\{1000, 5000\}$ |
| | length of episode | $\{20, 50\}$ |
| **Actor** | Gradient termination criterion | $\{10^{-3}, 10^{-4}\}$ |
| | $m_a$ | $\{1000, 10000\}$ |
| | Armijo max step-size | 1000 |
| | Armijo step-size decay $\beta$ | 0.9 |
| | Policy initialization (linear) | $\mathcal{N}(0, 0.1)$ |
| | Policy initialization (tabular) | Random |
| **Linear Critic** | Gradient termination criterion (direct) | $\{10^{-6}, 10^{-8}\}$ |
| | Gradient termination criterion (softmax) | $\{10^{-8}, 10^{-10}\}$ |
| | $m_c$ | $\{1000, 10000\}$ |
| | Armijo max step-size | 1000 |
| | Armijo step-size decay $\beta$ | 0.9 |
| **Others** | $\eta$ in direct | $\{0.001, 0.005, 0.01, 0.1, 1\}$ |
| | $c$ in direct | $\{0.001, 0.01, 0.1, 1\}$ |
| | $\eta$ in softmax | $\{0.001, 0.005, 0.01, 0.1, 1\}$ |
| | $c$ in softmax | $\{0.001, 0.01, 0.1\}$ |
| | $d$ | $\{40, 50, 60, 80, 100\}$ |

Table 1: Parameters for the Cliff World environment

| | Parameter | Value/Range |
|---|---|---|
| **Sampling** | # of samples | $\{1000, 10000\}$ |
| | length of episode | $\{20, 50\}$ |
| **Actor** | Gradient termination criterion | $\{10^{-4}, 10^{-5}\}$ |
| | $m_a$ | $\{100, 1000\}$ |
| | Armijo max step-size | 1000 |
| | Armijo step-size decay $\beta$ | 0.9 |
| | Policy initialization (linear) | $\mathcal{N}(0, 0.1)$ |
| | Policy initialization (tabular) | Random |
| **Linear Critic** | Gradient termination criterion (direct) | $\{10^{-6}, 10^{-8}\}$ |
| | Gradient termination criterion (softmax) | $\{10^{-6}, 10^{-8}\}$ |
| | $m_c$ | $\{10000, 1000000\}$ |
| | Armijo max step-size | 1000 |
| | Armijo step-size decay $\beta$ | 0.9 |
| **Others** | $\eta$ in direct | $\{0.01, 0.1, 1, 10\}$ |
| | $c$ in direct | $\{0.01, 0.1, 1\}$ |
| | $\eta$ in softmax | $\{0.01, 0.1, 1, 10\}$ |
| | $c$ in softmax | $\{0.01, 0.1\}$ |
| | $d$ | $\{40, 50, 60, 100\}$ |

Table 2: Parameters for the Frozen Lake environment

# G   Additional Experiments

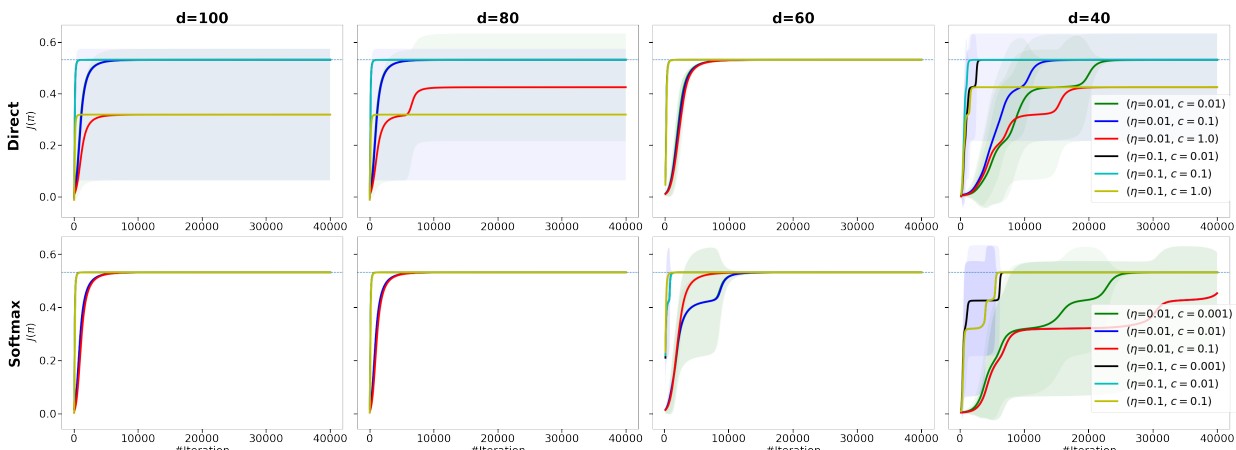

Figure 2: **Cliff World – Linear actor and Linear critic with exact** $Q$ **computation** Assessing the impact of $c$ (trade-off parameter in decision-aware framework) on the performance. We perform the experiment on the same setting as Fig. 1, linear actor and linear (with four different dimensions) critic with known MDP on Cliff World environment. We consider two values of functional step-size $\eta \in \{0.01, 0.1\}$ and three values of $c \in \{0.01, 0.1, 1\}$ for direct and $c \in \{0.001, 0.01, 0.1\}$ for softmax representations, and compare the performance of 6 combinations. Overall, among different critic capacities and step-sizes, the value of $c = 0.01$ demonstrates superior performance in both policy representations.

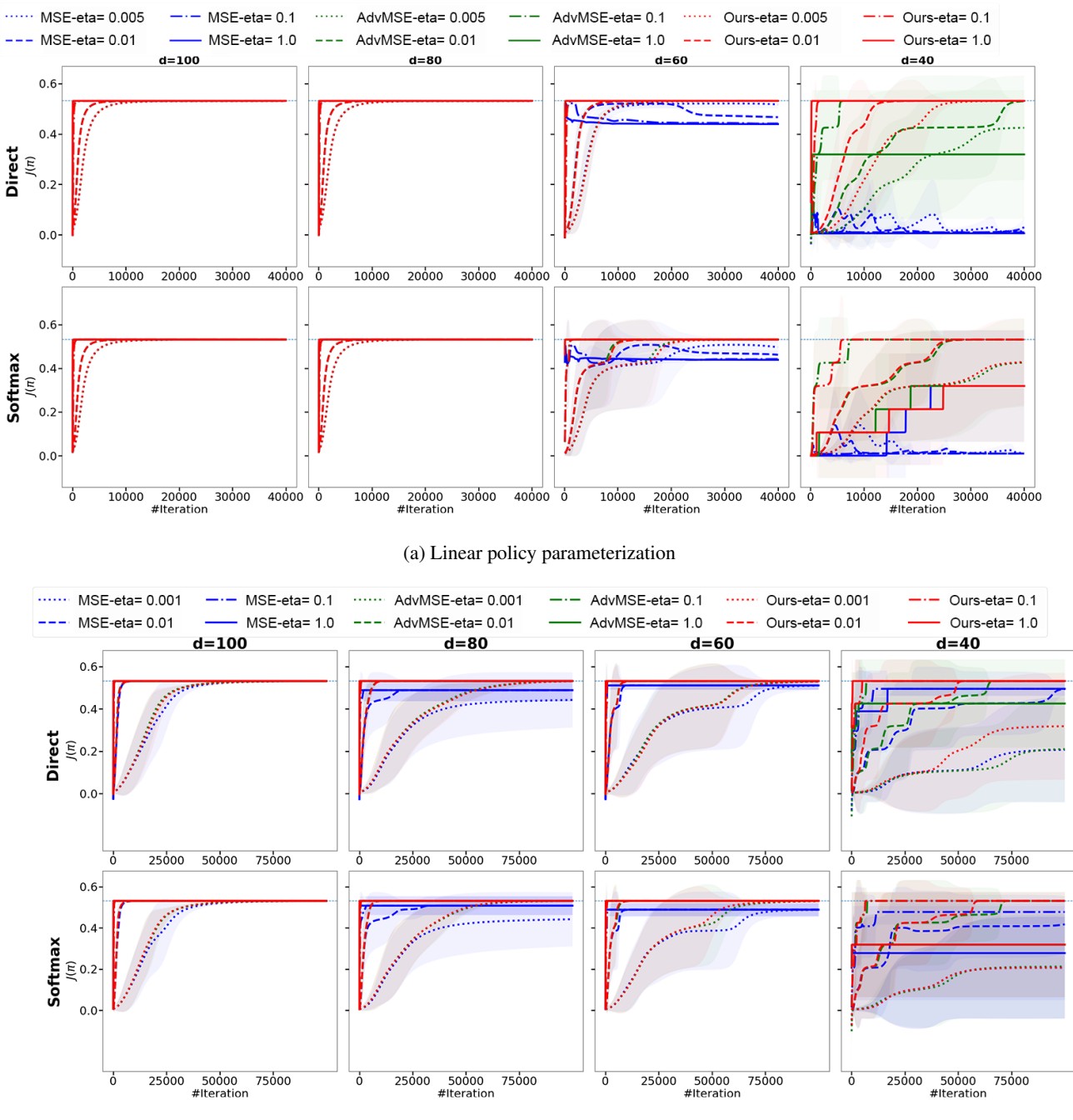

(a) Linear policy parameterization

(b) Tabular policy parameterization

Figure 3: **Cliff World – Linear/Tabular actor and Linear critic with exact $Q$ computation**: Comparison of decision-aware, `Adv-MSE`, and `MSE` loss functions using a linear actor Fig. 3a and Fig. 3b coupled with a linear (with four different dimensions) critic in the Cliff World environment for direct and softmax policy representations with known MDP. For $d = 100$ (corresponding to an expressive critic) in both actor parameterizations and $d = 80$ in linear parameterization, all algorithms have almost the same performance. In other scenarios, minimizing `MSE` loss function with any functional step-size leads to a sub-optimal policy. In contrast, minimizing `Adv-MSE` and decision-aware loss functions always result in reaching the optimal policy even in the less expressive critic $d = 40$. Additionally, decision-aware convergence is faster than `Adv-MSE` particularly when the critic has limited capacity (e.g. In $d = 40$ for direct and softmax representations and for both actor parameterizations, decision-aware reaches the optimal policy faster.)

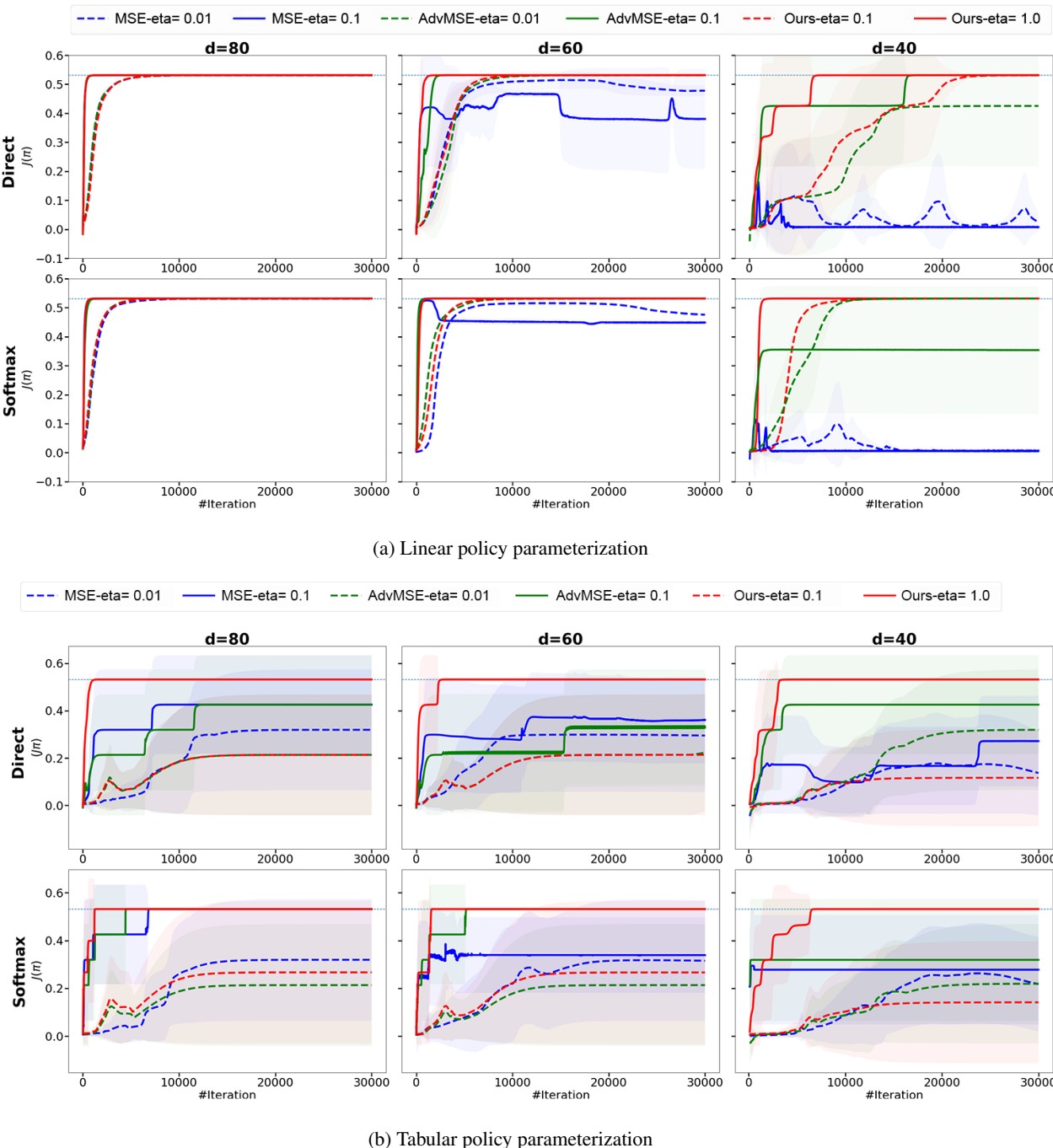

(a) Linear policy parameterization

(b) Tabular policy parameterization

Figure 4: **Cliff World – Linear/Tabular actor and Linear critic with estimated** $Q$: Comparison of decision-aware, `Adv-MSE`, and `MSE` loss functions using a linear actor Fig. 4a and Fig. 4b coupled with a linear (with three different dimensions) critic in the Cliff World environment for direct and softmax policy representations with Monte-Carlo sampling. When employing a linear actor alongside an expressive critic ($d = 80$), all algorithms have nearly identical performance. However, minimizing the `MSE` loss with a linear actor and a less expressive critic ($d = 40, 60$) leads to a loss of monotonic policy improvement and converging towards a sub-optimal policy in both representations. Conversely, minimizing the decision-aware and `Adv-MSE` losses enables reaching the optimal policy. Notably, decision-aware demonstrates a faster rate of convergence when the critic has limited capacity (e.g., $d = 40$) in both policy representations. The disparity among algorithms becomes more apparent when using tabular parameterization. In this case, the decision-aware loss either achieves a faster convergence rate (in $d = 80$ and $d = 60$), or it alone reaches the optimal policy ($d = 40$).

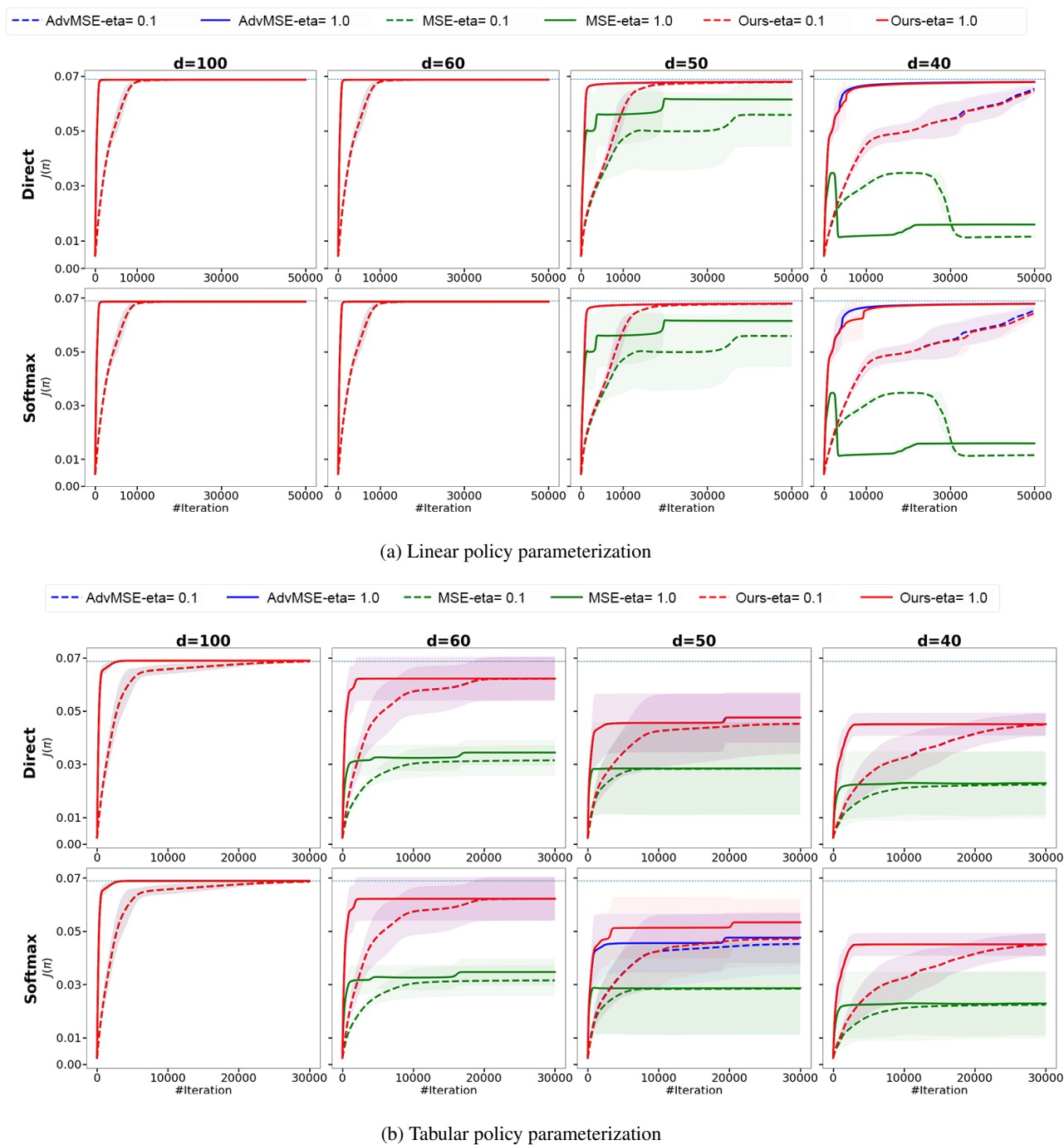

(a) Linear policy parameterization

(b) Tabular policy parameterization

Figure 5: **Frozen Lake – Linear/Tabular actor and Linear critic with exact** $Q$ **computation**: Comparison of decision-aware, `Adv-MSE`, and `MSE` loss functions using a linear actor Fig. 5a and Fig. 5b coupled with a linear (with four different dimensions) critic in the Frozen Lake environment for direct and softmax policy representations with known MDP. For $d = 100$ (corresponding to an expressive critic) in both actor paramterizations and $d = 60$ in linear paramterization, all algorithms have the same performance. In other scenarios, minimizing `MSE` loss functions leads to worse performance than decision-aware and `Adv-MSE` loss functions and for $d = 40$ in linear parameterization `MSE` does not have monotonic improvement. `Adv-MSE` and decision-aware almost have a similar performance for all scenarios except $d = 50$ with tabular actor where decision-aware reaches a better sub-optimal policy.

**Frozen Lake – Linear/Tabular actor and Linear critic with estimated** $Q$: For the Frozen Lake environment, when estimating the $Q$ functions using Monte Carlo sampling (all other choices being the same as in Fig. 5), we found that the variance resulting from Monte Carlo sampling (even with $\geq 1000$ samples) dominates the bias. As a result, the effect of the critic loss is minimal, and all algorithms result in similar performance.

