# OpenReview forum: "Decision-Aware Actor-Critic with Function Approximation and Theoretical Guarantees"
_NeurIPS.cc/2023/Conference — NeurIPS 2023 poster_

### Official Review · Reviewer_uLAb · 2023-06-25

**Soundness:** 2 fair
**Presentation:** 3 good
**Contribution:** 2 fair
**Rating:** 3
**Confidence:** 4

**Summary:**

The authors present a decision aware actor-critic algorithm and then try to analyse the same. Subsequently they also provide some numerical results.

**Strengths:**

The authors try and provide an analysis of an actor-critic algorithm. However, there are a number of questions that are unclear to me that I write in detail below.

**Weaknesses:**

I am writing down the entire list of comments on the paper below. One can call these weaknesses as these points have not been explained well or there are issues in the explanations provided.

1. The authors say in the introduction that the objective of the critic is to minimize the value estimation error across all states and actions. This is incorrect since a critic's job is to estimate the expected TD error for any given policy provided by the actor and the actor's job is to find the policy that minimizes this expected TD error over policies. The authors seem to mention this at multiple places.

2. Weird notation: J_s(\pi) is well defined but what is J_s(\rho)? Note that \rho is the initial state distribution and then you say J(\pi) = E[J_s(\rho)]!

3. Note that d^\pi and \mu^\pi are not valid probability distributions. How do you then value sample states or state-action tuples from these?

4. What does it mean to say that since p^\pi(s,a) is a probability, you can write it in the Boltzmann form as you have written. Is it always possible to express any probability distribution as some Boltzmann distribution? Later you introduce a parameterization \theta to this distributional form. Why not directly give a parameterized Boltzmann probability if that is what you want for the ramdomized policy instead of doing this in such a round about manner.

5. In the definition of \pi_{t+1} in Section 3.1, obviously \eta plays a major role. What values can it take?

6. For the blue term on page 4, you mention that as c decreases, critic error decreases. I don't see how? As c decreases, the first term inside D_{\Phi^*}(.,.) will decrease as c decreases but that term is multiplied by 1/c which will blow up as c decreases.

7. Algorithm 1: How do you compute $\nabla_v L_t (v_k) and \hat{g}_t?

8. Step-sizes and stability of the algorithm: You use \alpha_c and \alpha_a as the step-sizes for the critic and actor respectively. I don't see any conditions written on these step-sizes. The reason is that actor-critic algorithms are meant to track policy iteration. This requires that the critic recursion moves on a faster timescale as compared to the actor recursion. When diminishing step-sizes are used, one requires that $\alpha_a go to zero at a rate faster than \alpha_c in order to get this effect of asymptotic convergence. Furthermore, the authors do not mention anything about the stability of their procedure. How does one ensure that this algorithm is stable - a precondition to ensuring convergence.

9. Since you do not assume anything about mixing times of the underlying Markov process under any policy, how do you ensure that Markov noise does not create any problems in convergence of the scheme, see for instance,Sajad Khodadadian, Zaiwei Chen, Siva Theja Maguluri, "Finite-Sample Analysis of Off-Policy Natural Actor-Critic Algorithm," ICML 2021.

10. In Proposition 2, what are the terms \tilde{H}_t^\dagger and [\hat{g}_t]_{s,a}?

11. How does one interpret Proposition 3? Not clear at what rate the second term decreases and whether it even does so? In the absence of a proof of decrease and the rate at which it happens, the result is meaningless.

12. In Section 4, you give a monotonic policy improvement result. TRPO also has a similar result. Which one is better - the improvement provided by your algorithm or TRPO?

13. Mistakes: Section 5.1 - you say \nabla_\pi J(\pi) = d^\pi(s) Q^\pi(s,a). What is s and what is (s,a) on the RHS since the LHS does not have any s or (s,a) tuple? Also, in Section 5.2, you say \nabla_\pi(J(\pi)) = d^\pi(s) A^\pi(s,a) p^\pi(a|s)? What are s, a on RHS. There is clearly a mistake since LHS does not depend on these quantities. Also, how do you reconcile these two definitions for the same object?

14. The bandit proposition 5 and also 7 suddenly come in and appear crude with such assumptions as deterministic rewards, deterministic Q-value updates etc. It is not clear why these results have been provided.

**Questions:**

I have given my questions in detail above.

**Limitations:**

I think the analysis of convergence is flawed since there are no assumptions or results  regarding (a) Lipschitz continuity of the objectives, (b) step-sizes used in the actor-critic scheme, (c) stability of the two coupled recursions, (d) fast mixing nature of the Markov noise, etc., have been made or shown. Moreover, from my comments mentioned above, the paper lacks on several fronts and will require significant revision and re-review in order to be publishable.

---

> ### Author Rebuttal · Authors · 2023-08-08
>
> **Part 2 of the rebuttal (please refer to the global rebuttal for Part 1)**
>
> [4] *...don't see any conditions written on the step sizes...*
>
> Both step-sizes $\alpha_c$ and $\alpha_a$ can be set according to the smoothness of the critic ($L_t(\omega)$) and actor ($\ell_t(\theta)$) objectives. Using such step-sizes is enough for Proposition 3 to hold. Setting these step-sizes is orthogonal to the main message of the paper, and we relegated these details to the Appendix (see Lines 705-707) where we directly make use of the result and associated step sizes in [26]. In practice, we set both step-sizes adaptively using an Armijo line-search which, by definition, guarantees ascent (descent) for the actor (critic) objectives respectively.
>
> [5] *How does one interpret Proposition 3?*
>
> Proposition 3 proves an $O(1/T)$ convergence of the policy to the *neighborhood* of a stationary point of $J$. Such results proving convergence to a neighborhood that depends on the critic error are common in the literature (see [2,23,60,61] in our references). As we explain in Lines 245-255, the second term is the critic error, which in turn, depends on the optimization error and the bias due to the function approximation. When using a large value of $m_c$ and a sufficiently expressive critic parameterization, the critic error in the second term can be driven to zero.
>
> [6] *...Is it always possible to express any probability distribution as some Boltzmann distribution? Later you introduce a parameterization $\theta$ to this distributional form. Why not directly give a parameterized Boltzmann probability...?*
>
> Indeed, any distribution can be represented using a softmax transformation such that $p^\pi(a|s) \propto \exp(z^\pi(s, a))$. As explained in Section 2 (Functional representation vs Policy parameterization), we explicitly distinguish between the functional representation of a policy and its parameterization. Specifically, we can use any function approximation to parameterize either the distribution $p^\pi(a|s)$ (the direct representation of a policy) or its logits $z^\pi(s, a)$ (the softmax representation of a policy). This abstraction is important because it enables us to prove monotonic policy improvement results for *any* actor/critic parameterization (Proposition 2).
>
> [7] *What values can $\eta$ take in the definition of $\pi_{t+1}$*
>
> Section 3 considers a general functional representation of the policy. For Proposition 1 to hold, $\eta$ needs to be set such that $J + \frac{1}{\eta} \Phi$ is convex (please see the Proposition 1 statement). In Section 5, we instantiate the generic framework and this gives more concrete requirements on $\eta$. For example, for the softmax representation in Proposition 6, $\eta$ needs to be set to $1 - \gamma$.
>
> [8] *For the blue term on page 4, you mention that as c decreases, critic error decreases. I don't see how?*
>
> As mentioned in Lines 194-195, in order to gain some intuition about the effect of $c$, we use a second-order Taylor series expansion (around $c = 0$) which shows that the $D_{\Phi^*}$ term is proportional to $c$. As a simple example, choose $\Phi$ to be the Euclidean mirror map, in which case the $D_{\Phi^*}$ term becomes equal to $\frac{c}{2} || \nabla J(\pi_t) - \hat{g}_t ||^{2}$.
>
> [9] *Algorithm 1: How do you compute $\nabla_v L_t (v_k)$ and $g_t$?*
>
> Algorithm 1 is a *generic* algorithm and the computation of $\nabla_v L_t (v_k)$ and $g_{t}$ depends on the choice of the functional representation. We instantiate both these quantities for the direct and softmax functional representations in Section 5. For example, for the direct representation (Line 278), $[g_t]_{s, a} = d^{\pi_t}(s)  \hat{Q}^{\pi_t}(s, a)$. $L_t (\omega)$ is defined in Proposition 4 and corresponds to the blue term. In practice, we use a sample-based approximation of $L_t(\omega)$ and compute its gradient using automatic differentiation.
>
> [10] *The bandit proposition 5 and also 7 suddenly come in...*
>
> As is explained in Lines 104-105, 283-284, we aim to demonstrate the effectiveness of using the proposed decision-aware critic loss over the squared critic loss typically used in practice. Consequently, we consider bandit examples to demonstrate that even for extremely simple examples (2-armed bandit with deterministic rewards, a special case of the multi-armed bandit, and hence RL problem), the **typical actor-critic algorithm that relies on minimizing the squared loss can fail to converge to the optimal policy, whereas the proposed decision-aware actor-critic algorithm converges to the optimal policy**.
>
> [11] *Weird notation: $J_s(\pi)$ is well defined but what is $J_s(\rho)$?*
>
> This is a typo, and it should be $J(\pi) = E_{s \sim \rho}[J_s(\pi)]$.
>
> [12] *Note that $d^\pi$ and $\mu^\pi$ are not valid probability distributions.*
>
> The definition of $d^\pi$ has a missing normalization factor of $1 - \gamma$, which makes $d^\pi$ a valid probability distribution. With this correction, $\mu^\pi$ is also a valid distribution.
>
> [13] *In Proposition 2, what are the terms $\tilde{H}_t^\dagger$ and $g_t$?*
>
> As explained in Section 3, $g_t \in \mathbb{R}^{SA}$ is any gradient estimator used to approximate $\nabla J(\pi_t)$, and $[g_t]_{s,a}$ is the $(s,a)$ component of this vector. The matrix $\tilde{H}_t$ is defined in Proposition 2, and $\tilde{H}_t^\dagger$ denotes the matrix pseudo-inverse. We will clarify this in the final version of the paper.
>
> [14] *...you say $\nabla_\pi J(\pi) = d^\pi(s) Q^\pi(s,a)$. What is s and what is (s, a) on the RHS...*
>
> $\nabla J(\pi) \in \mathbb{R}^{SA}$ and the statement in Section 5.1 should be fixed to $[\nabla J(\pi)]_{s, a}$ $= d^\pi(s) Q^\pi(s, a)$.
>
> Similarly, in Section 5.2, $[\nabla J(\pi)]_{s, a}$ $= d^{\pi}(s) A^{\pi}(s, a) p^{\pi}(a|s)$ where the (s,a) subscript denotes the component corresponding to state $s$ and action $a$.

---

> > ### Comment · Reviewer_uLAb · 2023-08-15
> > **response to rebuttal to uLAb**
> >
> > I appreciate the authors responses. However, many of my comments have been ignored. The most important of these is of stability of the procedure?
> > 1. Since the resulting scheme is a stochastic approximation algorithm, proving stability of such a procedure is important and there is no result in the paper which talks about stability. In other words, how do you ensure that the resulting stochastic iterates remain uniformly bounded almost surely on any sample trajectory?
> > 2. Also, the response concerning the choice of step-size is not convincing enough from a theoretical point of view. I say this because in actor-critic algorithms such as those of Konda and Borkar, SIAM J. Control and Optimization (1999) or  Konda and Tsitsiklis, SIAM J. Control and Optimization (2003), it is assumed that the critic step-size converges to zero slower than the actor's.
> > 3. Such an argument is missing here as  I also don't see how the noise is being treated here? Is it a martingale difference sequence or is it also Markovian? It should be Markovian because you are looking at samples coming from a Markov process. The convergence of the noisy scheme is not clear from the arguments.
> >
> > These are important concerns that cannot simply be wished away. I am not convinced with the responses to my questions and so I shall retain my rating of 3.

---

> > > ### Author Response · Authors · 2023-08-16
> > > **Further clarifications**
> > >
> > > We thank the reviewer for engaging with the rebuttal. We believe that we have addressed all of the reviewer's comments, and that there is still some misunderstanding.
> > >
> > > [1] *...Since the resulting scheme is a stochastic approximation algorithm...*
> > >
> > > As we have explained in the rebuttal (see the global response to all reviewers), **Alg. 1 is not a stochastic approximation algorithm similar to [Konda and Borkar, 1999]**. In order to clarify the difference between Alg 1. and the two time-scale setting in [Konda and Borkar, 1999], [59] (in our references), we can think of actor-critic as solving a bi-level optimization problem. Since the actor uses the $Q^\pi, A^\pi$ estimates from the critic in order to compute its loss and update the policy, the outer-level objective corresponds to the actor loss, whereas the inner-level objective is the critic loss. Similar to [23,60, a,b], Alg 1. aims to solve the inner-level optimization problem *using multiple critic updates* (Lines 5-8). On the other hand, the two time-scale algorithm in [59] performs one step of gradient descent (critic update) on the inner-level objective, followed by one gradient ascent step (actor update) on the outer-level objective.
> > >
> > > [2] *...there is no result in the paper which talks about stability...*
> > >
> > > Even though Alg 1. involves multiple critic updates, it is not required to *exactly* minimize the critic loss at iteration $t$. Specifically, the advantage of using the proposed joint objective for the actor and critic is that Proposition 2 guarantees monotonic improvement in $J(\pi)$ as long as the critic loss is less than a certain threshold. This result only depends on the magnitude of the critic loss (after $m_c$ updates). The form of the critic update (including the step-size) required to achieve that loss is irrelevant. Importantly, **the actor and critic are coupled via the threshold that the critic loss needs to achieve in order to guarantee policy improvement. Specifically, this threshold depends on the norm of the functional policy gradient (see Line 218)**. Hence, if this threshold on the critic loss is satisfied, Alg 1. can result in convergence to a stationary point of  $J(\pi)$. On the other hand, if the critic loss does not satisfy such a threshold, Proposition 3 still guarantees the convergence of Alg 1. to the neighborhood of a stationary point i.e. for the special case of the Euclidean mirror map, we can obtain a policy $\bar{\pi}\_T$ s.t that $||\nabla J(\bar{\pi}\_T)||^2 \leq O(\frac{1}{T} + \epsilon_{\text{critic}})$ where $\epsilon_{\text{critic}}$ is the critic error.
> > >
> > > On the other hand, in order to prove guarantees, the analysis in [59] requires reasoning about both the form and step-size of the critic update. **Importantly, in [59], the actor and critic are coupled through the step-size. Specifically, the critic step-size converges to zero at a slower rate than the actor's, effectively enabling more critic updates**
> > >
> > > Unlike conventional actor-critic analyses including [Konda and Borkar, 1999], [59], our theoretical results hold for non-linear function approximation and can handle off-policy updates. **Hence, our theoretical guarantees are much stronger than the stability results that the reviewer is alluding to.**
> > >
> > > [3] *Also, the response concerning the choice of step-size is not convincing enough from a theoretical point of view... I also don't see how the noise is being treated here...*
> > >
> > > Again, we refer to the global response to all reviewers. We reiterate this: compared to the two time-scale updates in [59], the advantages of our approach are that (i) we *do not need* to explicitly reason about the relative step-sizes for the actor/critic updates. This makes the resulting algorithm more stable, while retaining the theoretical guarantees in Proposition 3. (ii) Since our analysis abstracts out how the critic loss is minimized (similar to [23,60]), we do not need to make assumptions about the noise.
> > >
> > > We hope that our response has clarified the reviewer's misunderstandings and better placed our algorithm in the context of existing analyses of actor-critic. We will explicitly include these comparisons and explanations in the final version of the paper.
> > >
> > > [a] Kumar et al,  On the sample complexity of actor-critic method for reinforcement learning with function approximation, 2019.
> > >
> > > [b] Qiu et al, On the finite-time convergence of actor-critic algorithm, 2019.

---

### Official Review · Reviewer_odRi · 2023-07-03

**Soundness:** 3 good
**Presentation:** 3 good
**Contribution:** 3 good
**Rating:** 7
**Confidence:** 2

**Summary:**

The paper addresses the issue of objective mismatch in Actor-Critic methods by designing a joint objective that enables training the actor and critic in a decision-aware manner. The proposed algorithm ensures monotonic policy improvement, irrespective of the chosen policy and critic parameterization.

**Strengths:**

The paper presents an algorithm that guarantees monotonic policy improvement and has solid theoretical foundations.

**Weaknesses:**

The experimentation in the paper is somewhat limited, which may raise concerns about the algorithm's performance in more complex environments.

**Questions:**

In complex real-world environments, $\Delta_\pi J(\pi)$could only be estimated, e.g. using MC sampling. And for large MDPs, function approximation error of Q is unavoidable for large MDPs. The article provides experiments on two simple environments, Cliff World and Frozen Lake, using linear/tabular action and Linear cirtic. Thus, my questions are:

(1) Can the algorithm be applied to more challenging environments, such as the standard benchmark tasks in MuJoCo? If so, how well does it perform in those environments? If there are limitations or challenges in applying the algorithm to more difficult tasks, it would be helpful to elaborate on those limitations.

(2) Does the algorithm introduce significant extra time overhead compared to other methods? It would be useful to report the additional time required by the algorithm on the Cliff World and Frozen Lake environments.

**Limitations:**

Potential Challenges in Complex Environments: The applicability of the proposed algorithm to more complex environments, such as those found in the MuJoCo benchmark tasks, remains uncertain.

---

> ### Author Rebuttal · Authors · 2023-08-08
>
> We thank the reviewer for their positive feedback and address their questions below.
>
> [1] *Can the algorithm be applied to more challenging environments, such as the standard benchmark tasks in MuJoCo? If so, how well does it perform in those environments? If there are limitations or challenges in applying the algorithm to more difficult tasks, it would be helpful to elaborate on those limitations.*
>
> We emphasize that the aim of this paper is to lay down the theoretical foundations of decision-aware actor-critic algorithms. Our experiments are designed to isolate and study the effect of the critic loss, without non-convexity or optimization issues acting as confounders. While we intend to evaluate the proposed algorithm on an extensive deep RL benchmark in the future, these experiments are beyond the scope of the current work.
>
> However, we note that the algorithm can be directly used for more challenging environments. In order to do so, we would require more complex actor/critic parameterization in order to better generalize across states/actions. Theoretically, the monotonic policy improvement guarantees in Proposition 2 would still hold.
>
> From a practical perspective, Alg. 1 can be directly used with any actor/critic parameterization. With respect to hyper-parameter tuning, $\eta$ is only dependent on the smoothness of $J$ w.r.t $\pi$ and does not depend on the actor/critic parameterization. On the other hand, $\alpha_{a}$ and $\alpha_{c}$ are set adaptively using an Armijo line search. The line-search procedure only requires the smoothness of the actor/critic objectives and does not rely on these objectives being convex. Finally, since our actor-critic framework does not involve a two time-scale algorithm, the relative scales of $m_a$ and $m_c$ (the number of actor/critic updates) do not influence our results, and the performance improves with increasing $m_a$ and/or $m_c$ (see the discussion in lines 253-255 following Proposition 3).
>
> One limitation of the proposed algorithm is setting the hyper-parameter $c$. While we have a heuristic to set $c$ (see Appendix F.1), its effect on the performance for challenging environments is unclear. However, we do note that such hyper-parameter tuning is a major issue even for the standard algorithms in practice.
>
> [2] *Does the algorithm introduce significant extra time overhead compared to other methods? It would be useful to report the additional time required by the algorithm on the Cliff World and Frozen Lake environments.*
>
> No, compared to the other methods, the proposed algorithm involves minimizing a different loss function with respect to the critic and does not introduce significantly extra time overhead.

---

> ### Comment · Reviewer_odRi · 2023-08-10
>
> Thank the authors for their rebuttal. I don’t carefully review your theorems, as such, my perspective on that particular aspect is limited.
>
> However, theory and experimentation are not conflicting endeavors. Therefore, I recommend the authors to include more compelling experiments to enhance the quality of the work. This is not beyond the scope of your work.
>
> I respond quickly because I hope that perhaps during this discussion period, you could attempt some experiments using MuJoCo, as continuous control problems are crucial application scenarios. I'm not asking for results in all scenarios of MuJoCo; providing results in one or two scenarios would suffice.
>
> Furthermore, regarding the time complexity here, I'd like to know the wall time clock data if possible. Because using a different loss function might come with additional costs.
>
> Of course, those mentioned above are just some suggestions on my part. If you're unable to provide them, it's not a significant issue

---

> > ### Author Response · Authors · 2023-08-12
> > **Experimental evaluation**
> >
> > [1] *I hope that perhaps during this discussion period, you could attempt some experiments using MuJoCo*
> >
> > We thank the reviewer for engaging with the rebuttal. We agree that theory and experimentation are not conflicting endeavors, and consequently, we did provide experimental evidence that validates our theoretical results. As briefly alluded to in the rebuttal, there are several reasons why we did not consider Mujoco experiments in the current paper:
> >
> > 1. Mujoco experiments typically require over-parameterized deep neural networks for the critic. Since these models are highly expressive and can interpolate the data [a], it is possible to estimate all state-action values for simpler environments in the Mujoco suite. In this case, the choice of the critic loss becomes irrelevant. This effect can already be seen in our experiments -- In Figure 1, for $d = 80$ (corresponding to using a highly expressive model), the choice of the critic loss does not matter and all methods have similar performance.
> >
> > 2. For more complex environments, the experimental results for Mujoco are significantly affected by the choice of hyper-parameters [b], and secondary implementation-level factors have a major impact on the algorithm performance [c]. Unless we do careful ablations controlling for these factors, these experiments will not effectively test the paper's contribution. If the reviewer can suggest an experimental protocol that could help isolate the effect of the critic loss, we would be grateful and could test it.
> >
> > 3. Finally, our work is in line with similar theoretical papers that provide the necessary framework for developing algorithms with performance guarantees. Aware of the massive amount of engineering and finetuning required to develop state-of-the-art RL methods from these theoretical frameworks, we felt that it would be more honest to restrict our experiments to evaluate the specific advantages offered by our approach, highlighting the issues of standard approaches, without claiming to have developed a full-fledged RL algorithm.
> >
> > That being said, your point is well taken. We agree that more experiments will indeed enhance the quality of our work. Consequently, we are currently running experiments on Cart Pole, one of the simpler continuous control environments. Specifically, we are considering a linear critic with tile-coded features (following the experimental protocol in [d]). By varying the dimension of these features, we hope to study the effect of the critic loss. Because of time and computational constraints, we are not sure if we will be able to finish this set of experiments (with proper ablations) before the discussion period finishes. We will definitely add these experiments to the final version of the paper.
> >
> >     [a]. Zhang et al, "Understanding deep learning requires rethinking generalization", 2016
> >     [b]. Henderson et al, Deep reinforcement learning that matters, 2018
> >     [c]. Engstrom et al, "Implementation matters in deep policy gradients: a case study on PPO and TRPO", 2020
> >     [d]. Jain et al, "Towards Painless Policy Optimization for Constrained MDPs", 2022
> >
> > 2. *Furthermore, regarding the time complexity here, I'd like to know the wall time clock data if possible. Because using a different loss function might come with additional costs.*
> >
> > All methods have similar wall clock times. In particular, we report the average (across $1000$ outer iterations of Alg 1) time taken to minimize the critic loss corresponding to each method (TD, Adv-TD, and Ours) for the linear critic parameterization (with $d = 60$). Specifically, we use gradient descent with Armijo line-search (to automatically set the step-size $\alpha_c$) and terminate the algorithm when the gradient norm decreases below $10^{-6}$.
> >
> >     a. On the Cliff World environment, the wall clock times for the Decision-aware, TD, and Adv-TD methods are $0.0550$, $0.0779$, and $0.0427$ respectively.
> >     b. On the Frozen Lake environment, the wall clock times for the Decision-aware, TD, and Adv-TD methods are $0.0020$, $0.0051$, and $0.0017$ respectively.

---

> > > ### Comment · Reviewer_odRi · 2023-08-12
> > > **Suggestions on implementations.**
> > >
> > > Thank you so much! I think conducting experiments on CartPole is a fantastic initiative. I agree that the critical loss might attenuate the impact of your design, and my suggestion is that perhaps instead of comparing it directly with baselines, you could consider conducting an ablation study. Another suggestion is that you might explore implementing your critic choice concept directly on SAC, which could be a potential avenue for future work. I don't have any further questions, and once again, I truly appreciate your efforts.

---

### Official Review · Reviewer_os4C · 2023-07-05

**Soundness:** 4 excellent
**Presentation:** 4 excellent
**Contribution:** 3 good
**Rating:** 7
**Confidence:** 3

**Summary:**

The authors develop a generic decision-award AC algorithm where both the actor and the critic take steps iteratively to optimize some “policy improvement lower bound” under the FMAPG framework. In essence, each step takes the gradient estimation error as the critic error, and characterizes the policy improvement in terms of the estimated gradient $\hat g$. This result seems to come from the Taylor expansions of the Bregman divergence. The authors argue that since both the actor and the critic act in a cooperative manner, the critic therefore only focuses on parts of the state-action pairs that have the largest impact on the actor’s performance improvement. The authors provide the conditions for monotonic policy improvement and show that the algorithms converge to some stable points. Examples and experiments show that the proposed algorithm outperforms TD/AdvTD methods on certain tasks.

**Strengths:**

The algorithm is interesting and well-motivated. Although the idea is not new to the RL community, the theoretical results in this work are sound and in general support the authors’ claims. Moreover, the paper is well written and I find most of the statements clearly presented.

**Weaknesses:**

In order to perform GD/SGD, you at least need to rollout trajectories collected under $d^{\pi_t}$. I didn’t find evidence supporting the sample efficiency of the proposed algorithm as you have claimed. Questions include:
1. How many samples are used for each GD step in your experiment?
2. How large is the variance of the proposed algorithm compared to TD/AdvTD when the sampling budget is limited, especially when the samples do not have sufficient coverage?
4. How do the relative scales of $m_c$ and $m_a$ influence your results?


**Questions:**

1. In practice, buffers containing trajectories collected under history policies are commonly used. Is there any evidence showing that your algorithm can handle the distribution shift issue?

2. Can you perform gradient steps for the actor/critic simultaneously? If so, what are the relative step sizes for the inner and the outer loops?

**Limitations:**

No concerns here.

---

> ### Author Rebuttal · Authors · 2023-08-08
>
> We thank the reviewer for their positive feedback and address their questions below.
>
> [1] *How many samples are used for each GD step in your experiment?*
>
> We varied the number of samples -- $\{1000, 5000\}$ for Cliff World and $\{1000, 10000\}$ for Frozen Lake in order to estimate the $Q^\pi, A^\pi$ functions for a linear critic. These $Q^\pi, A^\pi$ estimates were then used to update the parameters of the linear actor. All details about the number of samples, step-sizes, and number of inner-loops are presented in Appendix F.
>
> [2] *How large is the variance of the proposed algorithm compared to TD/AdvTD when the sampling budget is limited, especially when the samples do not have sufficient coverage?*
>
> Our preliminary experiments reveal that the variance in the gradient estimator $\hat{g}_t$ (and hence in the $Q^\pi, A^\pi$ functions) affects the performance of all the compared methods in a similar manner. While it is important to control the variance and evaluate the methods when the samples do not have sufficient coverage in practice, our experiments are designed to study and isolate the effect of the critic loss. The form of the critic loss becomes important when the bias (because of using function approximation with limited capacity) dominates the variance. Hence, we did not evaluate the methods in the small sample regime, but rather used the same number of samples for all methods, and compared their relative performance. We will clearly explain this in the final version of the paper.
>
> [3] *How do the relative scales of $m_a$ and $m_c$ influence your results?*
>
> Note that **we do not update the actor and critic in a two time-scale setting (one environment interaction and update to the critic followed by an actor update)**. Hence, the *relative* scales of $m_a$ and $m_c$ do not influence our results.
>
> The proposed actor-critic algorithm is similar to the protocol in [2,60] in our references. Specifically, the critic interacts with the environment in *batch* and uses these interactions to form $\nabla \hat{J}(\pi)$ and form the decision-aware critic loss (Line 4 in Alg. 1). The critic is then trained (using $m_c$ inner-loops) to minimize the critic loss and form $\hat{Q}^\pi, \hat{A}^\pi$ estimates (Lines 5-8 in Alg. 1). These estimates are used to train the actor (using $m_a$ inner-loops) to maximize the surrogate function and update the policy. The performance of Alg. 1 monotonically improves with increasing $m_c$ and $m_a$ -- increasing $m_c$ ensures smaller critic error while increasing $m_a$ ensures better surrogate optimization for the actor. This effect is captured in theory by Proposition 3 (please see lines 253-255)
>
> [4] *In practice, buffers containing trajectories collected under history policies are commonly used. Is there any evidence showing that your algorithm can handle the distribution shift issue?*
>
> For the current paper, our experiments are designed to isolate and study the effect of the critic loss, without non-convexity, optimization, and distribution shift issues acting as confounders. Consequently, we did not perform large-scale experiments that require the use of replay buffers.
>
> However, we note that the proposed framework can systematically incorporate the use of a replay buffer. In particular, in our framework, the policy at which the gradient is computed can be any policy, not just the current one. Hence, a buffer containing trajectories from multiple history policies can be used to construct the surrogate $\ell_t(\theta)$ around that mixture of policies, which is perfectly valid. In order to incorporate a replay buffer with our framework, we can also make use of alternative algorithms such as dual-averaging in the policy space. This would enable us to utilize the gradient estimates from history policies and make use of the replay buffer. We aim to pursue this interesting direction in the future.
>
> [5] *Can you perform gradient steps for the actor/critic simultaneously? If so, what are the relative step sizes for the inner and the outer loops?*
>
> As explained above, we do not perform gradient steps for the actor/critic simultaneously in a two time-scale setting. Hence, their relative step-sizes do not influence our results. In practice, we set $\alpha_a$ and $\alpha_c$ using an Armijo line search, and vary $\eta$, the outer step-size used to instantiate the surrogate function for the actor. Please refer to Appendix F for further details.

---

> > ### Comment · Reviewer_os4C · 2023-08-15
> >
> > Thank you for addressing my concerns. While your explanation on the relative stepsize helps clarify things, I still have some reservations. I understand your intention to isolate the effect of the critic loss, and the monotonic increase in $m_c$ and $m_a$ does make sense for enhanced performance. However, it would be greatly beneficial if you could provide additional support for the theoretical analysis in Proposition 3, e.g., by showcasing empirical evidence of your algorithm's superior performance compared to its counterparts in scenarios where there's insufficient critic update, such as with small $m_c$ or larger stochastic error, you could offer a more robust validation of your approach. My evaluation of the work still remain unchanged.

---

> > > ### Author Response · Authors · 2023-08-16
> > > **Experimental evaluation**
> > >
> > > Thank you for engaging with the rebuttal, and for your suggestion. There are three sources of error for an insufficient critic -- the bias (due to the limited capacity of the critic), the optimization error (due to small $m_c$) and the variance (the stochastic error due to insufficient samples). Our experiments demonstrate that when the bias dominates, using a decision-aware critic loss does indeed result in better performance. For the final version of the paper, we will include ablation studies varying $m_c$ and the number of samples, and hence compare the different methods when the optimization error or variance dominates.

---

### Official Review · Reviewer_NBGk · 2023-07-05

**Soundness:** 3 good
**Presentation:** 4 excellent
**Contribution:** 4 excellent
**Rating:** 7
**Confidence:** 3

**Summary:**

This research addresses the mismatched objectives in actor-critic (AC) methods used in reinforcement learning (RL). By introducing a joint objective for training the actor and critic in a decision-aware fashion, a generic AC algorithm is developed. The algorithm ensures monotonic policy improvement regardless of the policy and critic parameterization. The proposed approach offers advantages over traditional methods, as demonstrated through rigorous analysis and empirical evaluations.

**Strengths:**

I think this work really pushes the RL community research efforts further by answering:

> can we design a generic actor critic algorithm with joint objective?

The main contribution of generic actor critic algorithm with joint objective is a really nice idea worthy for publication at NeurIPS.

**Weaknesses:**

I have only have one minor weakness for this work as follows:

> Using linear representations as "general function approximation" is a bit weak. I presume this is the reason why only simple RL problems have been demonstrated in this work.

I am open to discussions with the authors and reviewers to increase my score. All the best for future decisions!

**Questions:**

na

---

> ### Author Rebuttal · Authors · 2023-08-08
>
> We thank the reviewer for their positive feedback and address their questions below.
>
> [1] *Using linear representations as general function approximation is a bit weak. I presume this is the reason why only simple RL problems have been demonstrated in this work*
>
> We emphasize that the main contribution of this work is to develop a theoretically principled framework for jointly training the actor and critic while being able to handle general function approximation. Consequently, our main theoretical results (Propositions 1-3) hold for function approximation schemes beyond linear models. For example, Proposition 2 holds for *any* actor or critic parameterization and demonstrates monotonic policy improvement for general function approximation. This is in contrast with existing work that focuses on tabular or linear parameterization for the actor and/or critic.
>
> From an experimental perspective, we chose to use simple RL problems to demonstrate the importance of being decision-aware. These simple experiments enable us to isolate and study the effect of the critic loss (the standard squared TD loss vs the decision-aware loss instantiated in Prop 4,6), without non-convexity or optimization issues acting as confounders.

---

> ### Comment · Area_Chair_nNFt · 2023-08-16
> **Are you satisfied by the answers?**
>
> Dear reviewer,
>
> Would you please indicate whether the authors' response is satisfactory for you? If not, please engage with the authors, so we can get a better assessment of this work.
>
> Thank you,
> Area Chair

---

> > ### Comment · Area_Chair_nNFt · 2023-08-18
> >
> > Hi, I am following up on this!
> >
> > Thank you,
> > Area Chair

---

> > > ### Comment · Reviewer_NBGk · 2023-08-20
> > >
> > > The rebuttal addressed my concerns. I have adjusted my rating considering the rebuttal and other reviewers’ concerns.

---

### Author Rebuttal · Authors · 2023-08-08

**We respond to the major comments in the review of Rev. uLAb here. We believe that it would be helpful for all the reviewers to go through this response as it highlights the paper's key contributions, addressing possible misunderstandings**

We thank the Rev. uLAb for their feedback. However, we note that **the weaknesses highlighted by the reviewer stem from a misunderstanding of the paper's setting, or consist of notational typos that do not affect the results. In this part, we address their major misunderstanding about the actor-critic setting studied in our paper. Due to constraints on the rebuttal length, we address the notational typos and other questions in Part 2 (the direct rebuttal to Rev. uLAb).**

[1] *... a critic's job is to estimate the expected TD error for any given policy provided by the actor and the actor's job is to find the policy that minimizes this expected TD error over policies...*

The reviewer's understanding is incorrect -- the critic's job is to estimate/learn a policy's value, whereas TD methods are one way to do so. For example, please refer to [Konda et al, Actor-Critic Algorithms], the original actor-critic paper which clearly says this as *The critic uses an approximation architecture and simulation to learn a value function*

[2] *I think the analysis of convergence is flawed since there are no assumptions or results regarding (a) Lipschitz continuity of the objectives, (b) step-sizes used in the actor-critic scheme, (c) stability of the two coupled recursions, (d) fast mixing nature of the Markov noise, etc., have been made or shown.*

First, note that **Alg.1 does not update the actor and critic in a two time-scale setting (one environment interaction and update to the critic followed by an actor update)**. The proposed actor-critic algorithm is similar to the protocol in [2,60] in our references. Specifically, the critic interacts with the environment in *batch* and uses these interactions to form $\nabla \hat{J}(\pi)$ and form the decision-aware critic loss (Line 4 in Alg. 1). The critic is trained (using $m_c$ inner-loops) to minimize the critic loss and form $\hat{Q}^\pi, \hat{A}^\pi$ estimates (Lines 5-8 in Alg. 1). These estimates are used to train the actor (using $m_a$ inner-loops) to maximize the surrogate function and update the policy. **Since we are not in the two time-scale setting, the relative scales of the actor/critic step-sizes or the number of inner loops do not influence the performance of Alg 1**.  We now compare our results to the two time-scale analyses (for example, in [59] in our references) of actor-critic.
1. Since we do not update or analyze the actor and critic in a two time-scale setting, we do not need to consider coupled recursions, nor do we need to set the two step-sizes at different scales.
2. We do not require Lipschitz continuity of either the actor or critic objectives but assume that the actor objective is smooth for our theoretical results in Proposition 3. Proposition 2 does not require any such smoothness assumption for either the actor/critic.
3. **Both Propositions 2 and 3 are independent of how the critic loss is minimized** i.e. we could use TD or any other policy evaluation method in order to estimate the value function. Proposition 2 provides necessary and sufficient conditions on the magnitude of the critic error in order to ensure monotonic policy improvement (and hence convergence to a stationary point). On the other hand, Proposition 3 guarantees convergence to a *neighborhood* of a stationary point, where the neighborhood depends on the critic error. This result holds for *any* critic error and is similar to the results obtained in [2,23,60] (see Lines 259-262 for a discussion). Since these results are agnostic to how the critic error is minimized, we do not need to model the mixing of the Markov chain, and consequently make no assumptions.
4. **Compared to the existing actor-critic analyses (for example [59,61]), our theoretical results require fewer assumptions on the function approximation.** For example, the monotonic policy improvement result in Proposition 2 holds for *any* actor or critic parameterization (including complex neural networks). In contrast, the typical analyses of actor-critic that utilize a two time-scale update only work with tabular or linear function approximation for the actor/critic.
5. Since we do not explicitly model how the critic error is minimized, we can only prove convergence to the neighborhood of a stationary point. This is in contrast to the existing two time-scale analyses that jointly analyze the actor and critic, and show convergence to a stationary point (see [59] for example). Hence, we prove weaker results for a more general class of function approximation. We briefly explain this in Lines 255-265 and will add a more thorough comparison in the final version of the paper.
6. Finally, **our analysis supports off-policy updates i.e. the actor can re-use the value estimates (from the critic) to update the policy multiple times (corresponding to Lines 10-13 in Alg 1)**. This is possible because we explicitly distinguish between a policy's functional representation and its parameterization. This is in contrast to existing analyses of actor-critic methods that require interacting with the environment and gathering new data after each policy update.

[3] *You give a monotonic policy improvement result. Which one is better - the improvement provided by your algorithm or TRPO?*

**The monotonic policy improvement result for TRPO only holds for the tabular setting, does not handle function approximation nor does it handle the critic error**. As explained in Lines 225, Proposition 2 presents a condition to ensure monotonic policy improvement *regardless of the policy representation and parameterization of the policy or critic*. Compared to TRPO, our result can thus handle the critic error and *any* function approximation for the actor/critic.

---

### Decision · Program_Chairs · 2023-09-21

**Decision:**

Accept (poster)

**Comment:**

The actor-critic algorithms are often developed by designing a "good" actor and a "good" critic. It is not, however, obvious what it means for an actor or critic to be good. Ideally the critic should be designed in order to be useful for the actor. This paper approaches the problem of finding good actor and critic by formulating it as a Mirror Descent optimization problem where the effects of both actor and critic jointly appear. The choice of mirror map determines the loss function for both actor and critic. This leads to perhaps unexpected result that the right loss function for critic should not necessarily be the squared error between the action-value function estimate and the true value function of the policy.

We have three Accepts and one Reject. We had some discussions between the authors and reviewers, but the negative reviewer (uLAb) was not convinced about some technical detail of the paper. Reviewer uLAb's main issue is that the they consider the algorithm as a two time-scale stochastic approximation procedure, and demanded guarantees for stability.

I read the paper myself. I believe this is an important work for the RL community, as it lays a new optimization-based foundation for how we approach designing actor-critic algorithms.

I also see why Reviewer uLAb, who raised several good questions, had some concerns about the paper that were not eventually resolved. My short summary of the reason is that some of the results of the paper, especially those in Section 4 (Theoretical Guarantees), are a bit opaque and need more discussions and explanations. For example, it is difficult to clearly see the effect of choice of the inner optimization of the critic and actor, specified by the choice of m_c and m_a, on the final results. Proposition 3 has a O(exp(-m_a)) dependency, without specifying the constants. And m_c does not appear, as far as I can see, except indirectly through delta_t.

Or the LHS of Proposition 3 is difficult to interpret. It shows how far one iteration of the exact MD vs. Inexact MD (with a critic error) performs. The consequence of this is only briefly, and not very concretely, discussed in the paragraph afterwards. I would love more explanation, discussions, and examples there.

Overall, I believe this a good paper that has important results and has some room to be improved by trying to address reviewers comments and concerns, especially those that are based on a misunderstanding.

Other minor comments:
- Please increase the number of runs from 5 to a much larger value, especially given that the environments are very simple and the overlap between some curves is substantial.

- Calling the standard squared loss on the Q function "TD" is inaccurate, as Q is either computed exactly or through Monte Carlo estimate. There is no bootstrapping here to warrant calling it TD. This is simply MSE error.